# Brain-wide correspondence of neuronal epigenomics and distant projections

Jingtian Zhou[1,2,18], Zhuzhu Zhang[1,3,18], May Wu[1,4], Hanqing Liu[1], Yan Pang[5], Anna Bartlett[1], Zihao Peng[6,7], Wubin Ding[1], Angeline Rivkin[1], Will N. Lagos[5], Elora Williams[8], Cheng-Ta Lee[9], Paula Assakura Miyazaki[5], Andrew Aldridge[1], Qiurui Zeng[1,4], J. L. Angelo Salinda[5], Naomi Claffey[10], Michelle Liem[10], Conor Fitzpatrick[10], Lara Boggeman[10], Zizhen Yao[11], Kimberly A. Smith[11], Bosiljka Tasic[11], Jordan Altshul[1], Mia A. Kenworthy[1], Cynthia Valadon[1], Joseph R. Nery[1], Rosa G. Castanon[1], Neelakshi S. Patne[5], Minh Vu[5], Mohammad Rashid[5], Matthew Jacobs[5], Tony Ito[5], Julia Osteen[12], Nora Emerson[12], Jasper Lee[12], Silvia Cho[12], Jon Rink[12], Hsiang-Hsuan Huang[8], António Pinto-Duartec[12], Bertha Dominguez[9], Jared B. Smith[8], Carolyn O'Connor[10], Hongkui Zeng[11], Shengbo Chen[7,13], Kuo-Fen Lee[9], Eran A. Mukamel[14], Xin Jin[15,16], M. Margarita Behrens[12], Joseph R. Ecker[1,17]✉ & Edward M. Callaway[4,5]✉

Single-cell analyses parse the brain's billions of neurons into thousands of 'cell-type' clusters residing in different brain structures[1]. Many cell types mediate their functions through targeted long-distance projections allowing interactions between specific cell types. Here we used epi-retro-seq[2] to link single-cell epigenomes and cell types to long-distance projections for 33,034 neurons dissected from 32 different regions projecting to 24 different targets (225 source-to-target combinations) across the whole mouse brain. We highlight uses of these data for interrogating principles relating projection types to transcriptomics and epigenomics, and for addressing hypotheses about cell types and connections related to genetics. We provide an overall synthesis with 926 statistical comparisons of discriminability of neurons projecting to each target for every source. We integrate this dataset into the larger BRAIN Initiative Cell Census Network atlas, composed of millions of neurons, to link projection cell types to consensus clusters. Integration with spatial transcriptomics further assigns projection-enriched clusters to smaller source regions than the original dissections. We exemplify this by presenting in-depth analyses of projection neurons from the hypothalamus, thalamus, hindbrain, amygdala and midbrain to provide insights into properties of those cell types, including differentially expressed genes, their associated *cis*-regulatory elements and transcription-factor-binding motifs, and neurotransmitter use.

In any given brain, each neuron contributes uniquely to brain function. Nevertheless, neurons can be grouped into types on the basis of similarities and differences across several dimensions, including epigenetic state, gene expression, anatomy and physiology. Single-cell genomic technologies have been particularly impactful for cell-type classification owing to their high throughput (millions of cells assayed) and dimensionality (thousands of genes and even more genetic loci) leading to the identification of large numbers of transcriptomic and epigenomic clusters corresponding to possible cell types across the entire mouse brain.

A prominent and distinguishing anatomical feature of many brain neuron types is their long-distance axonal projections. Long-distance projections can be directly related to single-neuron gene expression or epigenomes by use of powerful linking technologies, including Barcoded Anatomy Resolved by Sequencing (BARseq)[3,4], retro-seq[5,6] and epi-retro-seq[2]. Previous studies have used retro-seq and epi-retro-seq to link mouse neocortical[2,5,6], hypothalamic[7] and thalamic projection cell types[8] to their genetic and epigenetic clusters, revealing complex but predictable relationships. For example, cortical neurons projecting

[1]Genomic Analysis Laboratory, The Salk Institute for Biological Studies, La Jolla, CA, USA. [2]Bioinformatics and Systems Biology Program, University of California San Diego, La Jolla, CA, USA. [3]Department of Human Genetics, The University of Chicago, Chicago, IL, USA. [4]Division of Biological Sciences, University of California San Diego, La Jolla, CA, USA. [5]Systems Neurobiology Laboratories, The Salk Institute for Biological Studies, La Jolla, CA, USA. [6]School of Mathematics and Computer Science, Nanchang University, Nanchang, China. [7]Henan Engineering Research Center of Intelligent Technology and Application, Henan University, Kaifeng, China. [8]Molecular Neurobiology Laboratory, The Salk Institute for Biological Studies, La Jolla, CA, USA. [9]Peptide Biology Laboratories, The Salk Institute for Biological Studies, La Jolla, CA, USA. [10]Flow Cytometry Core Facility, The Salk Institute for Biological Studies, La Jolla, CA, USA. [11]Allen Institute for Brain Science, Seattle, WA, USA. [12]Computational Neurobiology Laboratory, The Salk Institute for Biological Studies, La Jolla, CA, USA. [13]School of Computer and Information Engineering, Henan University, Kaifeng, China. [14]Department of Cognitive Science, University of California San Diego, La Jolla, CA, USA. [15]Center for Motor Control and Disease, Key Laboratory of Brain Functional Genomics, East China Normal University, Shanghai, China. [16]NYU–ECNU Institute of Brain and Cognitive Science, New York University Shanghai, Shanghai, China. [17]Howard Hughes Medical Institute, The Salk Institute for Biological Studies, La Jolla, CA, USA. [18]These authors contributed equally: Jingtian Zhou, Zhuzhu Zhang. ✉e-mail: ecker@salk.edu; callaway@salk.edu

solely to intratelencephalic (IT) targets fall into different clusters compared with those that project to extratelencephalic (ET) targets. By contrast, cortical layer 2/3 IT neuron types projecting to different cortical areas typically co-cluster despite having quantifiable and predictable genetic and epigenetic differences across the populations[2,6]. In the face of this complexity, it is unclear how single-cell genetic and epigenetic assays can be used to inform the structure and function of brain cell types and how neuronal structure can predict genetics and epigenetics. Further, it is unclear whether the principles learned from more limited previous studies can be extended to the entire brain, or whether there are different principles linking projection status to epigenetics for different brain areas.

To address these questions, we used epi-retro-seq to assay 33,034 neurons from 225 source-to-target combinations across the entire mouse brain. This approach combines retrograde labelling with single-nucleus methylation sequencing (snmC-seq), which allows identification of potential gene regulatory elements and prediction of gene expression in the same neuron. Gene expression can be predicted because non-CG (CH; in which H represents A, T or C) methylation of gene bodies is inversely related to RNA expression[9,10], and epigenetic elements regulating expression can be identified using methylation at CG (mCG) dinucleotides[9]. It is also expected that epi-retro-seq can provide unique insight into developmental mechanisms that shape connectivity because CH methylation accumulates during and peaks at the end of the developmental critical period, and mCG is reconfigured during synaptic development[11].

## Epi-retro-seq of 225 projections

To link single-neuron epigenomes to their projection targets and cell body locations, we used epi-retro-seq[2]. A retrogradely infecting AAV vector expressing Cre-recombinase (AAV-retro-Cre[12]) was injected into the brains of Cre-dependent, nuclear-GFP-expressing reporter mice (INTACT-Cre[9]) at a target region of interest (Fig. 1a). Four mice (two male and two female) were injected for each of twenty-four different target brain areas, including targets in the isocortex (CTX), hippocampal formation, olfactory areas, amygdala (AMY), cerebral nuclei (CNU), interbrain (IB), midbrain (MB), hindbrain (HB) and cerebellum (Fig. 1a,b and Supplementary Table 1). After 2 weeks, mice were killed and the brain was hand dissected into 32 possible source regions[13] spanning the same major brain structures as the target injections (Fig. 1a,b and Extended Data Fig. 1). For any given mouse, dissected sources corresponding to locations with known projections to the target were selected for profiling. Nucleus preparations were made from dissected source tissue and subjected to fluorescence-activated nuclear sorting for GFP⁺NeuN⁺ retrogradely labelled neuronal nuclei that were then processed for snmC-seq[14–16] (Fig. 1a and Methods).

After basic quality control, we recovered 48,032 single-cell methylomes (Supplementary Table 2) that were mapped to an unbiased sample of snmC-seq data with 301,626 cells[17] to carry out cell-type classification, and for removal of potential doublets (Methods and Extended Data Fig. 2a–e). Each single neuron in the epi-retro-seq sample was assigned to 1 of the 2,304 level 4 clusters identified in our companion study[17]. We have previously described cortical neurons from the same eight cortical sources included here and projecting to four cortical and six subcortical targets (63 combinations)[2]. For cortical sources, we now incorporate data for an additional five cortical targets and two more subcortical targets, with quality control steps similar to those in our previous work to eliminate experiments with inadvertent spread of injected AAV-retro-Cre into source regions (Methods). In total, 33,034 single-nucleus methylomes were analysed from 225 source-to-target combinations for which the projection target could be confidently assigned (Supplementary Table 3). These neurons were mapped to the unbiased snmC-seq dataset to visualize the epigenetic similarity of projection neurons across cell subclasses, sources and targets (Fig. 1c).

## Data analysis approaches across the brain

Overarching questions that can be addressed by this large dataset include the distinguishability of neurons from a given source that project to different targets, and whether neurons in different sources that project to the same target combinations are more or less distinguishable. To provide a resource that can be used to address the distinguishability of neurons with different projection targets, we trained linear models to distinguish neurons projecting to pairs of different targets on the basis of DNA methylation, and quantify which projection types are more different than the others by computing the model performance through area under the curve of the receiver operating characteristic (AUROC) for each of the target pairs from every source region (926 pairwise comparisons in total; Fig. 1d, Methods, Extended Data Fig. 3 and Supplementary Table 4).

To facilitate further, comprehensive multimodal characterization of projection neuron types, we integrated the epi-retro-seq data with unbiased samples of snmC-seq described above[17], and single-cell RNA-seq (scRNA-seq) data for 2.6 million neurons from 87 micro-dissected brain regions[18] (Fig. 1e, Methods and Extended Data Fig. 4). Alignment of epi-retro-seq data to these larger and carefully annotated datasets allows for the confident assignment of our cells to consensus clusters and enables the use of consistent nomenclature to describe the correspondence between projection targets and cell types or clusters. We carried out co-clustering of the three datasets to identify the cell clusters enriched in each projection type (Fig. 1f, Methods and Extended Data Fig. 5). It should be noted that in addition to clusters that are identified as being enriched in projection neurons, there are also neurons in other clusters without statistically significant enrichment. The absence of statistically significant enrichment should not be interpreted as an absence of projections from neurons belonging to a particular cluster.

Although microdissections effectively separate fairly small structures, most dissected source regions contain still smaller known anatomical regions, as typically illustrated in mouse brain atlases[13]. To potentially link projection-enriched clusters from particular sources to more precise anatomical loci, we carried out further integration with multiplexed error robust fluorescence in situ hybridization (MERFISH) data, allowing examination of the spatial locations of the cells belonging to particular clusters (Fig. 1g and Extended Data Fig. 6). Joint atlasing of single-neuron transcriptomes and epigenomes further allowed analyses of both the signature genes in projection-enriched clusters based on RNA expression, and methylation profiles to identify differentially methylated regions (DMRs) as putative *cis*-regulatory elements (CREs) and transcription factors (TFs) whose binding motifs are enriched in these DMRs (Fig. 1h). On the basis of the motif enrichment and the correlation between gene expression and DNA methylation, we constructed gene regulatory networks (GRNs) with TFs, DMRs and target genes as nodes (Fig. 1i). The GRNs allowed us to identify the most consistent changes across different data modalities, which pinpoint candidate regulators of projection-enriched clusters.

Extended Data Figs. 3–6 allow visualization of the integrative analysis approaches described above (for example, Fig. 1d–i) for all source-to-target combinations in our dataset. These integrative analyses were facilitated by combining source regions from the whole-brain datasets into 12 larger 'region groups' that were common to all 3 data modalities, before integration (Extended Data Fig. 2f,g). The groups include CTX, retro hippocampal region (RHP), piriform area (PIR), hippocampal region (HIP), main olfactory bulb and anterior olfactory nucleus (MOB + AON), striatum (STR), pallidum (PAL), AMY, thalamus (TH), hypothalamus (HY), MB and HB.

Below, we focus on a subset of all possible analyses of this very large dataset to highlight the utility of the data and to provide examples of interest. To facilitate further analyses of the complete dataset, we have generated a data browser that incorporates functions to allow each of the types of analysis that we highlight below to be conducted for any

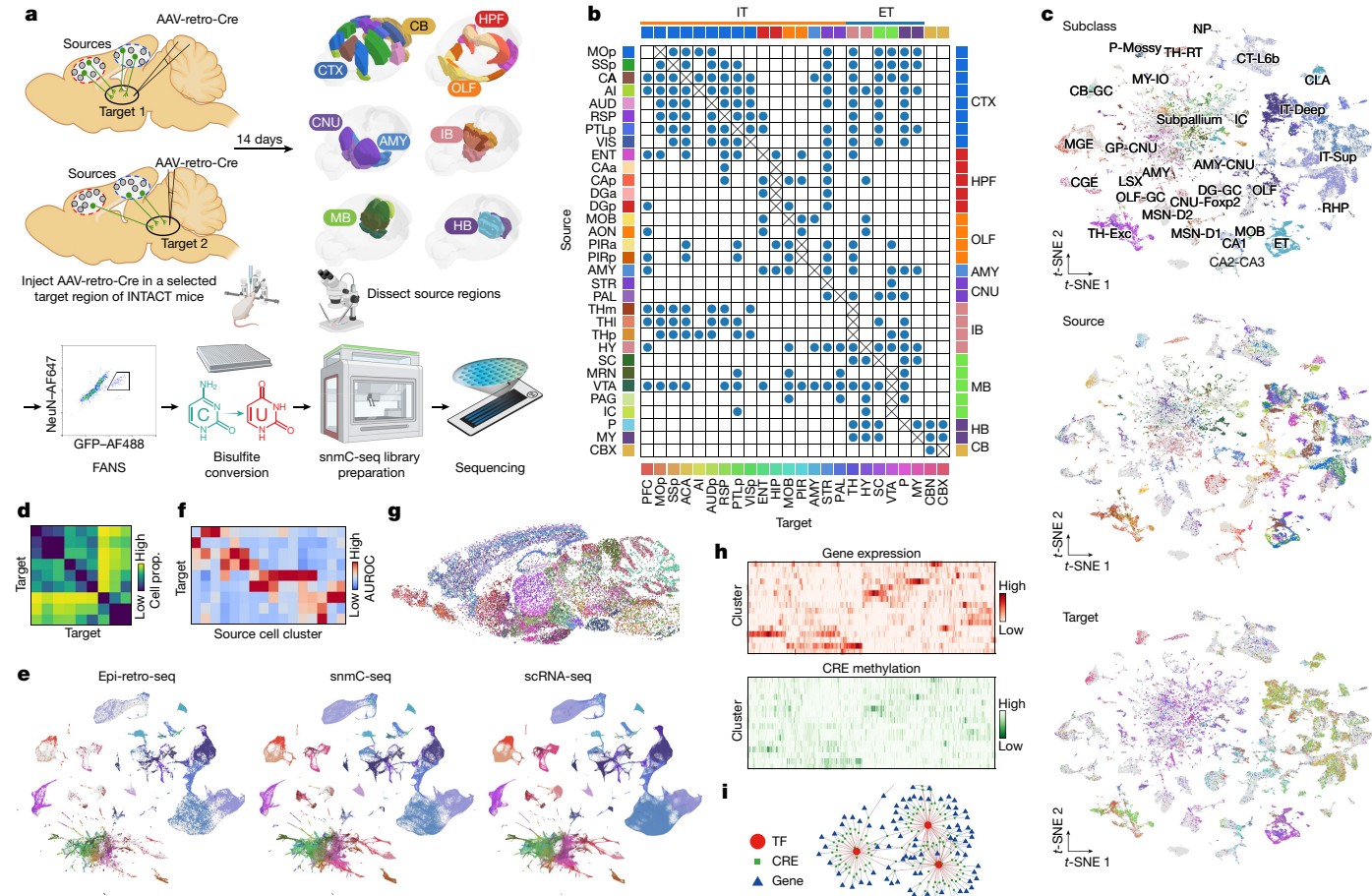

**Fig. 1 | The epigenomic landscape of brain-wide projection neurons.**
**a**, Schematics of the epi-retro-seq workflow for retrogradely labelling and epigenetically profiling single projection neurons. The three-dimensional brain contours of the source regions (top right) were derived from the Allen Mouse Brain Reference Atlas (http://atlas.brain-map.org, © 2017 Allen Institute for Brain Science, version 3 (2020)). The diagrams of the brain slices (top left) and the images of the surgical set up, microscope and library preparation apparatus in the workflow were created with BioRender.com. **b**, A total of 225 source–target combinations profiled using epi-retro-seq (blue dots) from 32 different source regions (rows) projecting to 24 different targets (columns) across the whole mouse brain. The colour palettes of sources (left), targets (bottom) and region groups (top and right) are labeled on the side and used across the article. **c**, Joint two-dimensional *t*-SNE of epi-retro-seq (*n* = 35,938) and unbiased snmC-seq (*n* = 276,187) neurons. snmC-seq neurons are shown in grey and epi-retro-seq neurons are coloured by cell subclass (top), the source regions of neurons (middle) or their projection targets (bottom; *n* = 33,034, after removing the experiments with less confident target assignment). **d**, As an example, AUROC for pairwise comparisons of AMY neurons projecting to nine targets. Higher AUROC scores suggest greater distinguishability between the compared projections based on their gene-body mCH levels. **e**, Joint *t*-SNE of whole-mouse-

brain neurons from epi-retro-seq (*n* = 35,743), unbiased snmC-seq (*n* = 266,740) and scRNA-seq (*n* = 2,434,472) coloured by cell subclass. **f**, As an illustration, the proportion (prop.) of neurons found in each AMY cell cluster (row) that projects to each target (column). Only clusters that were enriched for projection neurons are shown and values are *z*-score normalized across targets. **g**, A sagittal brain slice for MERFISH with all neurons coloured by their assigned subclasses. **h**, An illustration of joint analysis of single-cell transcriptomes and DNA methylomes that enables the characterization of gene expression patterns of DEGs between these projection-enriched clusters, as well as the mCG levels of DEG-associated putative CREs, as marked by DMRs. **i**, An illustration of GRN linking TFs, CRE and target genes. AI, agranular insular cortex; AUD, auditory cortex; AUDp, primary auditory cortex; CAa, anterior cornu ammonis; CAp, posterior cornu ammonis; CB, cerebellum; CBN, cerebellar nuclei; CGE, caudal ganglionic eminence; CT, corticothalamic; DGa, anterior dentate gyrus; DGp, posterior dentate gyrus; GC, granule cell; GP, globus pallidus; HPF, hippocampal formation; IC, inferior colliculus; IO, inferior olivary; LSX, lateral septal complex; MGE, medial ganglionic eminence; MSN, medium spiny neurons; NP, near projecting; OLF, olfactory areas; PAG, periaqueductal grey; PIRa, anterior piriform cortex; PIRp, posterior piriform cortex; THl, anterior lateral thalamus; THm, anterior medial thalamus; THp, posterior TH; VIS, visual cortex.

given source brain region and/or projection target that might be of particular interest (http://neomorph.salk.edu/epiretro).

## ET- versus IT-projecting neurons

In CTX, the most explicit correspondence between projection types and molecular types is observed for neurons that project to ET targets versus IT targets (for a breakdown of ET and IT target regions sampled, see Fig. 1b). To investigate whether such distinctions are shared with neurons from other sources, we explored the genetic distinguishability of neurons projecting to ET versus IT targets across source brain areas.

For the cortical source *t*-distributed stochastic neighbour embedding (*t*-SNE) plots, the ET-projecting neurons clearly separate into a distinct cluster (layer 5 ET) whereas the IT neurons are found distributed across the annotated IT clusters, as expected (Fig. 2a). ET and IT neurons are also well separated for the projection neurons in the entorhinal cortex (ENT; illustrated in the RHP plot) as well as TH (Fig. 2a), as expected from known projections of glutamatergic TH neurons to cortex versus GABAergic neurons to subcortical targets. ET versus IT neurons show varying levels of separation for the other sources. Generally, comparisons show some degree of separability for each of the source regions, but AUROC scores are higher for cortical sources than for subcortical

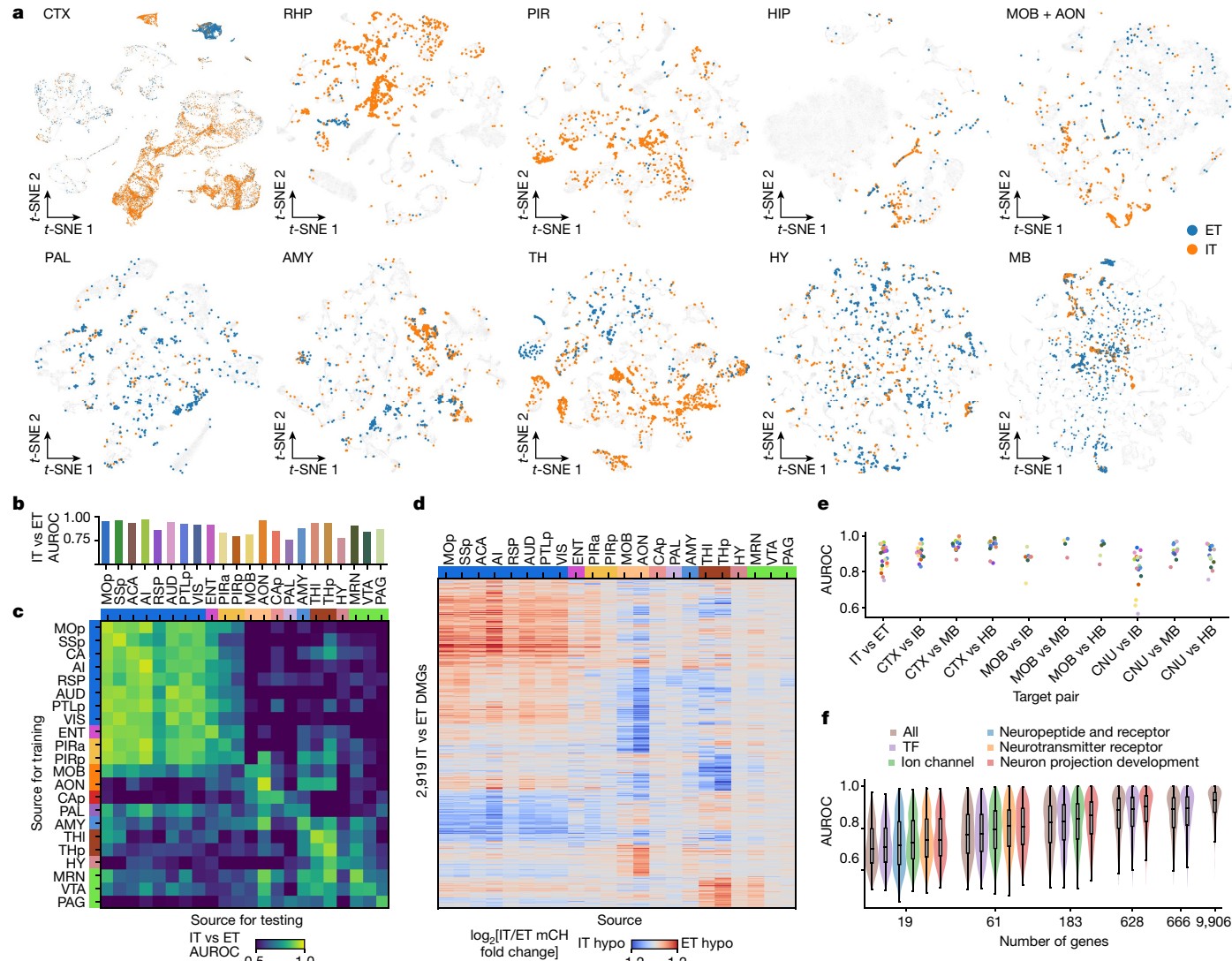

**Fig. 2 | Distinguishability of neurons projecting to different targets across the entire brain. a**, Joint *t*-SNEs of epi-retro-seq, unbiased snmC-seq and scRNA-seq data from ten region groups containing both IT- and ET-projecting neurons from the same source. Only the epi-retro-seq neurons projecting from the same source to ET (blue) and IT (orange) targets are coloured and the other cells are in grey. **b**, AUROC scores for comparisons between IT versus (vs) ET neurons from each of the 22 source regions. **c**, AUROC for IT versus ET neurons when the model was trained in one source region (row) and tested on another source region (column). A high AUROC indicates that the epigenetic differences between IT and ET neurons were similar between the training and testing sources. The values on the diagonal of **c** are the same as the values in **b**. **d**, The log$_2$ fold changes of mCH levels in each source at the IT versus ET DMGs. A total

of 2,919 DMGs are shown that were identified in at least one of the 22 source regions. The row and column colours represent the region groups in **b**–**d**. **e**, AUROC for comparisons of neurons projecting to each pair of target groups. Each dot represents a comparison in one source region (colour palette as in **b**). **f**, AUROC for the comparison of all target pairs from every source region (*n* = 926) with models using different sets of genes as features. Subsets or all of the 9,906 genes with high coverage in single cells were used. The larger gene sets were downsampled to the same number of genes as the smaller sets for comparison. All comparisons between gene sets are significant (false discovery rates (FDRs) in Supplementary Table 6; two-sided Wilcoxon signed-rank test, Benjamini–Hochberg procedure) except the ones between "neuron projection development" and "neurotransmitter receptor" with 19 genes.

courses (except TH and AON; Fig. 2b). These observations suggest that ET- versus IT-projecting neurons are not as genetically distinct for other source regions as they are for cortex and TH.

We next asked whether the epigenetic differences between ET- and IT-projecting neurons are shared across sources or alternatively whether different sources might have distinct molecular signatures that distinguish ET from IT neurons. We trained logistic regression models to distinguish ET- versus IT-projecting neurons in each 1 of the 22 sources, and tested whether each model could accurately separate ET and IT neurons from each of the other sources (Fig. 2c). We observed that the knowledge learned by the models could largely be transferred between isocortical sources and between isocortical and archicortical (ENT and PIR) areas, but not beyond the cortical regions. Other source

groups sharing similar ET versus IT differences include MOB and AON, as well as AMY, TH and midbrain reticular nucleus (MRN). To further evaluate these relationships, we identified the differentially methylated genes (DMGs) between ET and IT-projecting cells in each source, which merge into a combined set of 2,919 genes. Consistent with the AUROC results, the mCH levels of these DMGs show similar fold changes across isocortical and archicortical areas, MOB and AON, as well as different parts of TH and MB (Fig. 2d). These observations suggest that the mechanisms that give rise to relationships between projection targets and epigenetics are relatively conserved across cortical areas, across MOB and AON, and across AMY, TH and MRN, but differ between these sets of areas. We further assessed whether neurons projecting to more finely separated groups of targets might be more or less separable. We

separated the ET and IT targets into three finer groups (IT: CTX, MOB and CNU; ET: IB, MB and HB) and asked which pairs of target groups are less separable between ET and IT. Most of the target group pairs have better prediction results than ET versus IT, except that the CNU versus IB projecting cells are less distinguishable compared to ET versus IT on the basis of DNA methylomes with linear models (Fig. 2e).

To better understand what types of gene are contributing to the predictions of projection targets, we used genes in the following five categories as features to compute AUROC scores: neurotransmitter receptors; neuropeptides and receptors; ion channels; TFs; and neuron projection development (Methods and Supplementary Table 5). As different categories have different numbers of genes, and use of more genes increases prediction performance, we downsampled the larger gene categories into samples including the same numbers of genes as the smaller categories to facilitate comparisons in 5 different groups using from 19 to 666 genes and compared the AUROC scores. We observed that the neuron projection development genes have the strongest target prediction power, followed by neurotransmitter receptors, ion channels, neuropeptides and receptors, TFs, and randomly selected genes (Fig. 2f and Supplementary Table 6). Using all 628 genes in the neuron projection development category achieved an average AUROC of 0.88, which is slightly lower than that using all of the 9,906 genes as features (AUROC 0.91; Fig. 2f), suggesting that additional genes from other Gene Ontology (GO) categories also contribute to the target predictability. The greater predictive power of genes involved in neuron projection development aligns with the idea that similarities and differences between ET- and IT-projecting neurons (from various sources) are tied to developmental mechanisms specifying projection targets.

## Hypothalamic projection neurons

Analyses of gene expression and DNA methylation patterns have revealed the existence of numerous cell clusters within the HY, indicating a high level of cell-type diversity[18,19]. Additionally, the HY is composed of many distinct subregions and nuclei, each with unique functions and contributions to innate behaviours such as aggression, mating and feeding[20]. The HY therefore serves as an excellent use case for our dataset to further examine the relationships between neuronal cell types as defined by their transcriptional and epigenomic signatures, their projection patterns and their spatial organization.

We profiled hypothalamic neurons that project to ten distinct targets throughout the brain, including prefrontal cortex (PFC), MOB, STR, PAL, AMY, TH, superior colliculus (SC), ventral tegmental area (VTA) and substantia nigra (referred to later as VTA), pons (P) and medulla (MY). By integrating epi-retro-seq data with unbiased snmC-seq and scRNA-seq hypothalamic data, we identified a total of 94 neuronal cell clusters, of which 17 were enriched for the profiled HY projections (Fig. 3a,b). We annotated each co-cluster on the basis of the 302 neuronal cell subclasses identified in scRNA-seq across the whole brain[18]. Note that this annotation does not give a unique name to each of the finest cluster divisions. Therefore, clusters noted by different identifying cluster numbers (for example, 0 to 94) may share the same cluster name (for example, clusters 0 and 76 were both annotated as STN-PSTN Pitx2 Glut). Each of the projections to the ten targets was enriched in a unique subset of cell clusters (Fig. 3b–d and Supplementary Table 7). For example, HY-to-STR neurons were predominantly enriched in cluster 76, whereas HY-to-AMY neurons were uniquely enriched in cluster 64 (Fig. 3b), indicating distinct cell-type specificity of different HY projection neurons. HY-to-P and HY-to-MY show similar enrichment patterns across clusters, but only HY-to-P neurons were enriched in cluster 76 (Fig. 3b). Similarly, HY-to-PFC and HY-to-MOB neurons were both enriched in cluster 50 and 29, but HY-to-MOB neurons were uniquely enriched in cluster 17 (Fig. 3b). These results indicate that HY neurons projecting to structurally related targets may share some common molecular cell types but also exhibit some level of diversity.

These findings underscore the cell-type specificity and diversity of hypothalamic neurons projecting to different targets, shedding light on the potential functional roles of these cell clusters in various physiological and behavioural processes.

Next we examined the spatial distributions of projection-enriched HY neuron clusters. We carried out MERFISH on both sagittal and coronal brain slices to visualize the spatial location of neurons. By using the gene expression signatures of the projection-enriched clusters, we mapped them to MERFISH cells (Fig. 3e,f and Methods). Notably, most of the 17 projection-enriched clusters were located in different HY subregions, and the spatial distributions of cells from many clusters were distinguished by well-defined boundaries. For instance, clusters 0, 3 and 76 were located in separate 'stripes' in the dorsolateral HY, in regions corresponding to zona incerta or subthalamic nucleus (Fig. 3e). With respect to projection targets, some clusters that were enriched for particular projections were relatively confined to specific regions within the HY, whereas other projection-enriched clusters were distributed topographically across the HY. For example, HY-to-TH neurons were enriched in clusters 12, 32 and 3, all of which were located in well-delineated subregions of dorsal HY (Fig. 3b,f). By contrast, the clusters enriched for HB projection neurons were distributed along the anterior to posterior axis of the HY and also occupied locations across the dorsoventral and mediolateral axes (Fig. 3b,f). Overall, our findings underscore the fine-scale spatial organizations of these projection-enriched cell clusters within the HY and the varying degrees of topographical heterogeneity of the locations of projection-defined HY neuronal populations. These observations also highlight the utility of MERFISH data for linking projection cell types to locations that are much smaller than the regions that were dissected.

To gain insight into the molecular characteristics and gene regulation of the projection-enriched clusters, we further used the integrative analysis of epi-retro-seq, snmC-seq and scRNA-seq. We identified 1,163 differentially expressed genes (DEGs) across the 17 clusters in all pairwise comparisons (Fig. 3g). Each cluster has a different set of DEGs, even when there are several clusters enriched for projections to a particular target (note that this contrasts with results for TH below). Notably, many of the DEGs were found to be involved in neuronal function and connectivity, as exemplified by a few highlighted genes (Fig. 3g). mCH levels plotted with an inverted colour map are strikingly similar to the expression levels for the same genes indicating the anticorrelation between methylation and expression of these genes across clusters (Fig. 3h). To investigate the regulation of these DEGs, we identified 148,897 DMRs associated with the DEGs (Methods). The mCG levels of the DEG-associated DMRs exhibited differential methylation patterns consistent with the gene expression and gene-body mCH levels (Fig. 3i). To uncover the regulatory network of these DEGs, we further identified TFs whose binding motifs were enriched in CREs (Fig. 3j), and built a GRN of HY clusters connecting 389 TFs, 46,075 DMRs and 8,184 target genes (Methods). The network captures concordant variation of different data modalities across clusters. For example, Zic1, whose motif is enriched in hypo-CG-methylated DMRs (hypo-DMRs) of clusters 17 and 39, is also expressed at high levels in these clusters. These clusters are located at the anterior ventral part of the HY, and are enriched for neurons projecting to MOB. Another TF-encoding gene expressed at high levels in these clusters, Zic4, was predicted to be a potential target of ZIC1, with 15 DMRs at the flanking region (transcription start site ± 1 megabase) that have ZIC1-binding motifs, and the mCG levels are correlated with the expression of Zic1 and Zic4 (Fig. 3k). The analysis showed some shared sets of TFs between clusters enriched for some projections, such as HY-to-TH. By contrast, more varied sets of TFs were identified between clusters enriched for some other projections, such as HY-to-P or HY-to-MY (Fig. 3j). Additionally, distinct sets of TFs were observed between clusters that were enriched for different projections. Collectively, these findings underscore the existence of diverse GRNs that use distinct TFs and DMRs for different hypothalamic projections.

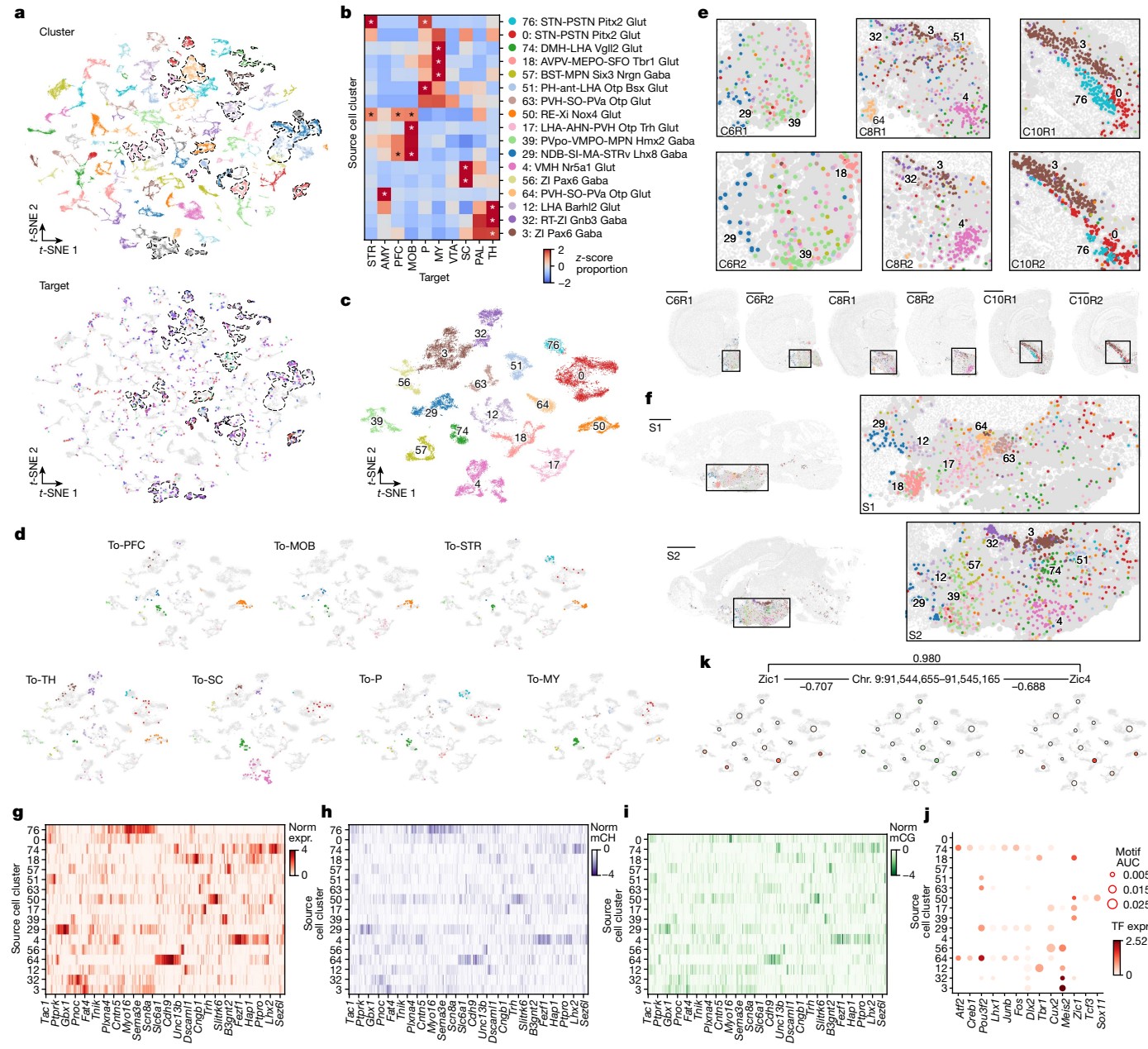

**Fig. 3 | The diversity of cell type, spatial location and gene regulation of hypothalamic projection neurons. a**, Joint *t*-SNE of epi-retro-seq (*n* = 1,572), unbiased snmC-seq (*n* = 11,554) and scRNA-seq (*n* = 148,840) data for hypothalamic neurons coloured by cell cluster (top) or projection target (bottom, same colour palette as left row colours in Fig. 1b). Seventeen clusters enriched for the profiled projection neurons are outlined. **b**, The proportion of each of the 10 projections in each of the 17 projection-enriched clusters, *z*-score normalized across targets. The asterisk denotes enrichment with FDR < 0.01 (one-sided Fisher exact test, Benjamini–Hochberg procedure; Methods). **c**,**d**, *t*-SNE of cells in the 17 projection-enriched clusters coloured by co-cluster. All cells are colored in **c**, whereas only cells projecting to one target are colored in each of the subplots in **d**. **e**,**f**, Projection-enriched HY clusters mapped to MERFISH data for six coronal slices (**e**; two biological replicates (R1 and R2) of slices from anterior to posterior (C6, C8 and C10)) and two sagittal slices (**f**; S1 and S2 from lateral to medial). Examples of clusters with specific spatial locations are labelled in the enlarged insets of each slice. Scale bars, 15 mm. The colour palette for clusters is the same in **b**–**f**. **g**–**i**, Normalized gene expression (norm expr., **g**) and gene-body mCH (**h**) levels of DEGs (*n* = 1,163) between the 17 projection-enriched clusters, and mCG levels of DEG-associated

DMRs (**i**; *n* = 148,897), in each cluster. The values are *z*-score normalized across clusters. The DEGs and cell clusters are arranged in the same orders in **g**–**i**. Only the DMRs with the highest anticorrelation with each DEG are shown in **i** to make the column orders consistent for **g**–**i**. Examples of DEGs with GO annotations related to neuronal function and connectivity are labelled on the *x* axis. **j**, Examples of TFs whose binding motifs were enriched in hypo-CG-methylated DMRs of each cluster are shown in the bubble plot. The size of each dot represents the enrichment level (area under the curve (AUC)). The colour of the dot indicates the expression (expr.) level of the TF. The clusters are arranged in the same order as in **g**–**i**. Cluster 76 was excluded in **i**,**j** as the number of cells in the cluster is too small (<30) to be included in the DMR analysis. **k**, Example triplet combination of TF–CRE–target gene in GRN of HY. Pearson correlation coefficients (PCCs) are shown on edges. The *t*-SNE plots share the same coordinates as **c**. Single cells are greyed and shown in the background. Large dots represent clusters, plotted at the mean coordinates of cells in the cluster. The size of each dot is proportional to the number of cells in the cluster. The dots are coloured by *Zic1* expression (left), mCG level of an example DMR (middle) and *Zic4* expression (right). Colour scales for gene expression are the same as in **g**, and the colour scale for mCG is the same as in **i**.

Furthermore, they offer valuable insights into the molecular mechanisms that govern the regulation of projection-enriched cell clusters and their associated genes in the HY.

In summary, our integrative analysis has revealed the relationships between hypothalamic neurons projecting to ten different targets and their methylation profiles, enrichment in molecular clusters and spatial locations of neurons belonging to those clusters. A previous study linked transcriptomic clusters and their spatial locations within the medial pre-optic area of the HY to specific behaviours[21], suggesting that those clusters might mediate their differential contributions to behaviour through differences in their projections. Another study directly linked transcriptomic clusters of neurons and their locations within the ventromedial HY to their projections to the medial pre-optic area or periaqueductal grey by combining retrograde labelling with scRNA-seq and sequential fluorescence in situ hybridization (seqFISH)[7]. Those experiments revealed projection-enriched clusters, as we have found for a different set of hypothalamic projection targets, but they did not observe clear relationships between transcriptomic clusters, behaviour-specific activation and projections to the periaqueductal grey or medial pre-optic area. We mapped the neurons from ref. 7 to our HY clusters and found that none of the behaviour-enriched clusters or projection-enriched clusters from ref. 7 correspond to any of our projection-enriched clusters in the entire HY (Methods and Extended Data Fig. 7). Our observations across the full spatial extent of HY and a large number of projection targets reveal strong correlations between clusters and projection targets, suggesting that cell types defined by their projections and genetics or epigenetics are also likely to make distinct contributions to hypothalamic function and related behaviours.

## Thalamic projection neurons

The TH is a primary hub in sensory and cortical information processing and also projects to subcortical structures. Like HY, TH consists of a large number of nuclei that are organized into many functional groups that are smaller than our dissected regions. The main, central regions of the TH are composed of exclusively excitatory regions (except for a few local GABAergic interneurons in the dorsal lateral geniculate nucleus (LGd)) that are reciprocally connected with cortical areas[22]. Other more ventral and lateral regions of the TH (such as the ventral lateral geniculate nucleus and reticular thalamic nucleus) contain GABAergic inhibitory neurons that are either reciprocally connected with thalamic excitatory neurons or project to subcortical structures such as the basal ganglia and brainstem[22]. In contrast to the HY, the TH had a lower degree of cell-type complexity as shown by the smaller number of cell clusters identified through gene expression analysis[18]. Despite both the TH and HY showing a high level of heterogeneity in their anatomical nuclei and projections, the differences in their cell-type complexity prompted us to investigate whether the relationships between cell types, their projections and their spatial locations in the TH differ from those observed in the HY, as discussed above.

We analysed thalamic neurons that project to 12 different targets, including 9 cortical areas (PFC, primary motor cortex (MOp), primary somatosensory cortex (SSp), anterior cingulate cortex (ACA), agranular insular cortex, primary auditory cortex, retrosplenial cortex (RSP), posterior parietal cortex (PTLp) and primary visual cortex (VISp)), SC, VTA and P. To gain a comprehensive understanding of these neurons, we combined epi-retro-seq data with unbiased snmC-seq and scRNA-seq data from the TH. Through this integration, we identified a total of 58 thalamic neuronal cell clusters (Fig. 4a), of which 33 clusters were enriched for epi-retro-seq neurons (Fig. 4b and Supplementary Table 7). It is worth noting that neurons dissected from different anatomical regions within the TH were located in distinct sets of clusters[19] (Fig. 4c), as expected from previous descriptions based on analysis of scRNA-seq data[8], suggesting that these molecularly defined cell clusters also have a spatial organization.

As observed in the HY, each population of thalamic projection neurons exhibited enrichment in distinct subsets of cell clusters, with each cluster showing enrichment for a specific set of projections, sometimes only one (Fig. 4b). Notably, the clusters enriched for TH-to-SC, TH-to-VTA, TH-to-P and TH-to-cortex were mostly mutually exclusive. Regarding cortical projections, TH-to-PTLp and TH-to-VISp neurons exhibited enrichment in a largely overlapping set of clusters, but with varying degrees of enrichment. TH-to-MOp and TH-to-SSp neurons also shared most of their enriched clusters, which differed from those enriched for TH-to-PTLp and TH-to-VISp (Fig. 4b). These results support the notion of a separation of thalamic cell types between the visual and motor pathways in the TH and highlight the heterogeneity of cell types within each pathway. Notably, TH-to-RSP neurons showed no overlap in enriched clusters with any other cortical projections, and were uniquely enriched in clusters 13, 26 and 47 (Fig. 4b). These clusters were annotated by their gene expression patterns as belonging to the anteroventral (AV) nucleus (clusters 13 and 26) and anterodorsal nucleus (cluster 47), which is consistent with TH-to-RSP projections originating from anterior thalamic nuclei[23]. In summary, TH neurons projecting to cortex versus subcortical targets were enriched in distinct sets of clusters. The enriched cell clusters for cortical projections were further segregated by different thalamic pathways, with several enriched cell clusters observed for each pathway or projection. These findings highlight the cell-type specificity as well as heterogeneity at the level of TH projections.

Such cell-type specificity and heterogeneity of TH projection neurons were also reported in transcriptomic analysis of single TH projection neurons (retro-seq)[8]. Retro-seq neurons of each projection were more similar to epi-retro-seq neurons for the corresponding projections than for any other projections (Extended Data Fig. 8). Small differences between the populations of retrogradely labelled TH neurons in the two datasets are probably due to the use of different injection coordinates for each cortical target (Supplementary Table 8).

As in our approach for the HY, we used the MERFISH data to map the spatial locations of the 33 TH projection-enriched clusters (Fig. 4d,e). Notably, almost all of these clusters exhibited a unique spatial pattern, many of them with distinct boundaries in the distributions of their cells (Fig. 4d,e). These boundaries often corresponded to specific thalamic nuclei, exemplified by clusters 25 and 45 that were enriched for P-projecting neurons and annotated as medial habenula cell types on the basis of their molecular signatures. When mapped to the MERFISH data, cells in these clusters demonstrated a clearly defined spatial location that corresponded to medial habenula (Fig. 4d). This illustrates the high resolution of our data and analysis, enabling the identification of specific medial habenula-to-P projection neurons among all thalamic neurons. Similarly, we were able to accurately map the molecularly annotated anterodorsal cluster 47 and anteroventral cluster 26 that were enriched for the TH-to-RSP projection to their corresponding locations in the dorsal and ventral anterior TH (Fig. 4e). This high resolution of our data also allowed us to investigate the molecular and spatial cellular heterogeneity within a projection. For instance, the visual input from the retina reaches VISp through LGd in TH. When mapped to MERFISH, clusters 34, 5, 4, 1 and 6 that were enriched for TH-to-VISp neurons collectively occupied the location that corresponds to LGd, with each cluster having a unique distribution within LGd (Fig. 4d). These findings underscore the heterogeneity of LGd-to-VISp neurons and provide valuable insights for future in-depth analysis of different types of LGd-to-VISp neurons.

Next, we investigated the gene regulation of thalamic neurons in these projection-enriched clusters (Fig. 4f). Joint analysis of scRNA-seq and scmC-seq data for TH identified a total of 2,348 DEGs (Fig. 4g,h) and 1,566,402 associated DMRs (Fig. 4i) across the 33 clusters. As expected, the expression levels of the DEGs (Fig. 4g) were anticorrelated with their mCH levels (Fig. 4h), and their associated DMRs also showed strong correspondence in terms of mCG levels (Fig. 4i). In contrast to HY, TH

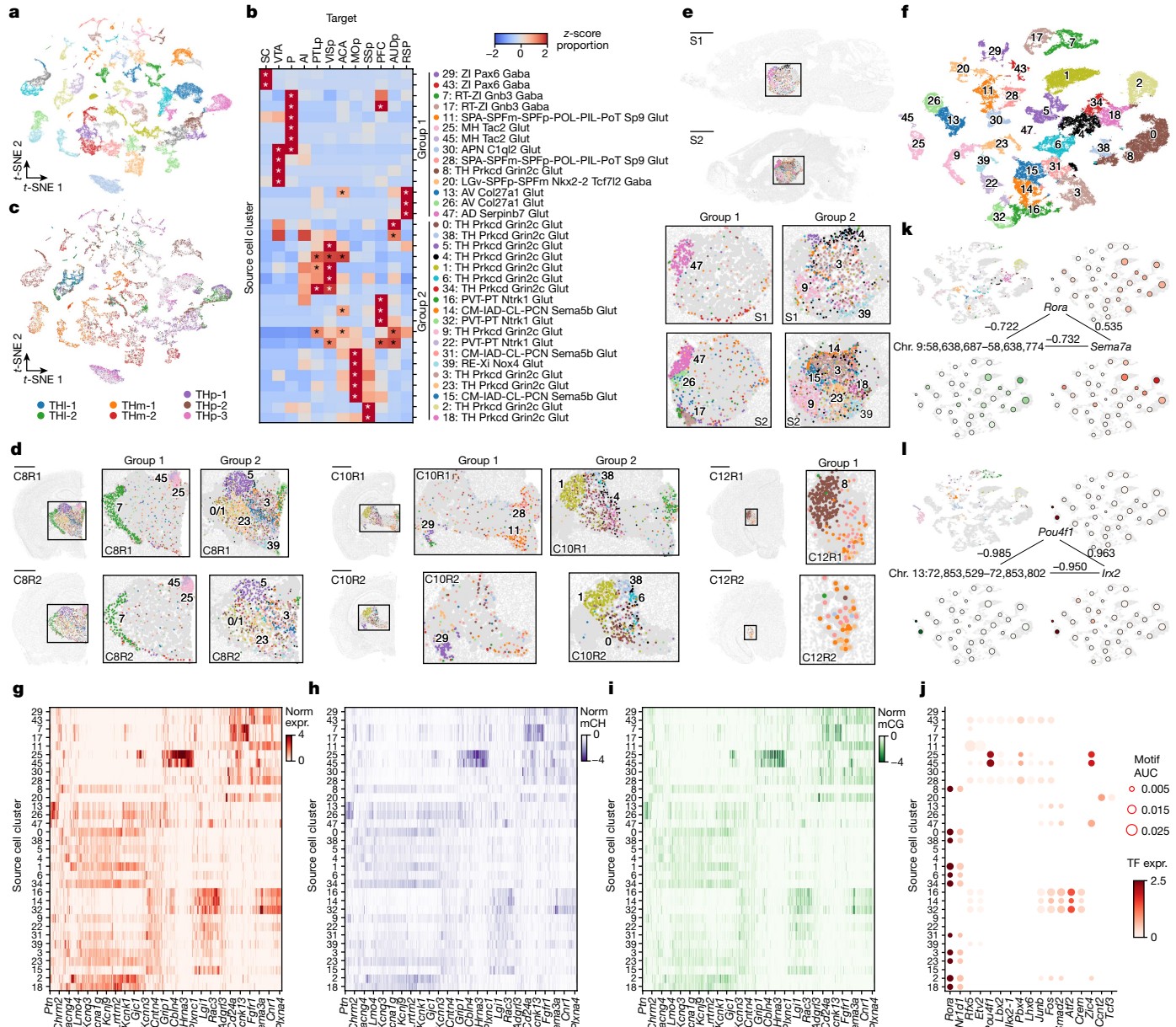

**Fig. 4 | The diversity of cell type, spatial location and gene regulation of thalamic projection neurons. a**, Joint *t*-SNE of epi-retro-seq (*n* = 2,606), unbiased snmC-seq (*n* = 16,943) and scRNA-seq (*n* = 162,795) data for thalamic neurons coloured by cell cluster. **b**, The proportion of each of the 12 assayed TH projections in each of the 33 projection-enriched clusters, *z*-score normalized across targets. The same set of colours is used for labelling clusters in both cluster group 1 (14 clusters) and cluster group 2 (19 clusters). The asterisk denotes enrichment with FDR < 0.01 (one-sided Fisher exact test, Benjamini–Hochberg procedure; Methods). **c**, TH neurons dissected from two anterior lateral regions (THl-1 and THl-2), two anterior medial regions (THm-1 and THm-2) and three posterior regions (THp-1, THp-2 and THp-3) coloured respectively in the *t*-SNE. See slices 7–10 in Extended Data Fig. 1 for details. **d**,**e**, Projection-enriched TH clusters mapped to MERFISH data for six coronal slices (**d**; two biological replicates (R1 and R2) of slices from anterior to posterior (C8, C10 and C12)) and two sagittal slices (**e**; S1 and S2 from lateral to medial). The colours of the clusters in the left (right) column of the insets are the same as for cluster group 1 (2) in **b**. Examples of clusters with specific spatial locations are labelled in the enlarged insets of each slice. C12R1 and C12R2 are not shown for cluster group 2. Scale bars, 15 mm. **f**, *t*-SNE of projection-enriched clusters coloured by co-cluster. The colour palette for clusters in **f** and the top left of **k**,**l** is the same as in **b**. **g**–**i**, Normalized gene expression (norm expr, **g**) and gene-body mCH (**h**)

levels of DEGs (*n* = 2,348) between the 33 projection-enriched clusters, and mCG levels of DEG-associated DMRs (**i**; *n* = 1,566,402), in each cluster. The values are *z*-score normalized across clusters. The DEGs and cell clusters are arranged in the same orders in **g**–**i**. Only the DMRs with the highest anticorrelation with each DEG are shown in **i** to make the column orders consistent for **g**–**i**. Examples of DEGs with GO annotations related to neuronal function and connectivity are labelled on the *x* axis. **j**, Examples of TFs whose binding motifs were enriched in hypo-CG-methylated DMRs of each cluster are shown in the bubble plot. The size of each dot represents the enrichment level (AUC). The colour of the dot indicates the expression level of the TF. The clusters are arranged in the same order as in **b**,**g**–**i**. **k**,**l**, Example triplet combinations of TF–CRE–target gene in GRN of TH. PCCs are shown on edges. The coordinates of the *t*-SNE plots are the same as in **f**. Top left, cells projecting to SSp (**k**) or P (**l**) are highlighted, coloured by co-clusters. In other plots, single cells are greyed and shown as background. Large dots represent clusters, plotted at the mean coordinates of the cells in the cluster. The size of each dot is proportional to the number of cells in the cluster. The dots are coloured by *Rora* (**k**) or *Pou4f1* (**l**) expression (top right), mCG level of an example DMR (bottom left) and *Sema7a* (**k**) or *Irx2* (**l**) expression (bottom right). The colour scales for gene expression are the same as in **g**, and the colour scale for mCG is the same as in **i**.

clusters enriched for the same projections exhibited similar expression patterns of DEGs and methylation patterns of the associated DMRs. Additionally, clusters enriched for the same projection had similar sets of TFs, whereas those enriched for different projections had more distinct sets of TFs (Fig. 4j), implying the existence of projection-specific GRNs. These relationships are in contrast to those observed in HY, where the organization of TF motifs is not closely related to projection targets.

We then constructed a GRN in TH, which consists of 10.9 million TF–DMR–target triplet combinations, involving 469 TFs, 375,279 DMRs and 13,283 target genes. These networks captured regulatory relationships reported in previous studies. For example, RORA has been identified as an essential factor for thalamocortical axon branching[24], and transcriptome analysis suggested that *Sema7a*, another essential regulator of thalamic cortical circuit maturation[25], could be a potential target of RORA. In our data, RORA motifs are enriched in many clusters that are enriched for neurons projecting to cortical targets. Similar expression patterns were observed for *Rora* and *Sema7a*, as both of these genes are also expressed at high levels in the cortical-projection-enriched clusters. A total of 43 DMRs that potentially mediate this regulation were identified at the flanking region of *Sema7a* (Fig. 4k and Methods). Our study also suggests new regulatory relationships in TH. POU4F1 has its binding motif enriched in DMRs hypo-methylated in clusters 25 and 45 that make projections to P. The network suggests that genes encoding prepattern TFs IRX1 and IRX2 (ref. 26) are candidate downstream targets of POU4F1, which is also specifically expressed in the same two clusters (Fig. 4l).

## Neurotransmitters in projection neurons

Recent brain-wide single-cell and spatial transcriptomic analyses have revealed remarkable heterogeneity and spatial specificity in neurotransmitter usage among different cell types across the mouse brain[18,19]. As described above and exemplified in TH and HY, our integrative analysis revealed high levels of cell-type and spatial specificity in neurons with different projections. These findings sparked a further investigation into the neurotransmitter usage of these distinct projection neurons that were in different brain regions and had different cell-type compositions. Insights into the neurotransmitter usage of different projection neurons may shed light on their functional properties and their potential role in behaviour, with broader implications for understanding neural circuits and the mechanisms underlying various brain functions and disorders.

To systematically examine the use of neurotransmitters by different projections, we quantified the levels of expression of nine canonical neurotransmitter transporter genes in each of the projection-enriched clusters within the 12 grouped brain regions described previously (Extended Data Fig. 5). These transporter genes included *Slc17a7* (*Vglut1*), *Slc17a6* (*Vglut2*) and *Slc17a8* (*Vglut3*) for glutamatergic neurons, *Slc32a1* (*Vgat*) for GABAergic neurons, *Slc6a2* (*Net*) for noradrenergic neurons, *Slc6a3* (*Dat*) for dopaminergic neurons, *Slc6a4* (*Sert*) for serotonergic neurons, *Slc6a5* (*Glyt2*) for glycinergic neurons, and *Slc18a3* (*Vacht*) for cholinergic neurons. In addition, we used histidine decarboxylase (*Hdc*) for histaminergic neurons. Our analysis revealed a diverse range of neurotransmitter usage across the projection-enriched clusters, particularly those in the MB and HB regions. Furthermore, a large proportion of the projection-enriched clusters expressed more than one neurotransmitter transporter gene. These findings indicate that there is a wide variation in neurotransmitter usage across different neural pathways and highlight the heterogeneity within some of these pathways. Below, we discuss a few notable cases in more detail, including projections from the HB regions of P and MY, AMY, and the MB region of VTA.

## Neurotransmitters in HB neurons

We analysed 11 HB projections, which included projections from P or MY to five different targets—TH, HY, SC, cerebellar nuclei and cerebellar cortex (CBX)—as well as the projection from P to MY. These projections were enriched in 20 cell clusters out of a total of 128 HB clusters. Notably, in both P and MY, neurons projecting to the CBX were the most distinct from other projection neurons (Fig. 5a).

The 20 projection-enriched clusters showed expression of six neurotransmitter transporter genes (Fig. 5b). Most of these clusters, such as the MY-to-CBX-enriched cluster 76, contain glutamatergic neurons expressing *Slc17a6*. Notably, *Slc17a7* (encoding VGLUT1) and *Slc17a6* (encoding VGLUT2) were co-expressed in cluster 0 neurons that were enriched for the P-to-CBX projection. These observations are consistent with those of previous studies that demonstrated the presence of VGLUT1 or VGLUT2 in climbing fibre (MY-to-CBX) terminals and both VGLUT1 and VGLUT2 in cerebellar mossy fibre (P-to-CBX) terminals using synaptic vesicle immunoisolation[27]. Moreover, different neurotransmitters were used in clusters enriched for the same projections. For instance, clusters 10, 11 and 27 were enriched for P-to-HY projections. Among them, cluster 10 is GABAergic, cluster 11 is glutamatergic, and cluster 27 is serotonergic, showing co-expression of *Slc6a4* (encoding SERT) and *Slc17a8* (encoding VGLUT3). Furthermore, several of these clusters also exhibited distinctive spatial distributions when mapped to the MERFISH data, such as clusters 0, 76, 10 and 27 (Fig. 5c). Together, these results underscore the extent of molecular, cellular and spatial specificity and diversity within HB projections.

We observed that neurons projecting to CBX from P or MY were distinct from other projections originating from the same regions. To investigate this further, we examined the molecular signatures that could differentiate CBX-projecting neurons from other projection neurons in P or MY. Analysis of gene-body DNA methylation identified genes that could distinguish the P-to-CBX cluster (0) from other projection-associated P clusters, or differentiate the MY-to-CBX cluster (76) from other projection-associated MY clusters (Fig. 5d). Notably, only 5 genes were common between the top 100 genes in the two sets, namely *Slit3*, *Phactr3*, *Pcbp3*, *Atp10a* and *Cdk14* (highlighted in Fig. 5d). *Slit3* encodes a repulsive axon guidance molecule[28,29], and *Phactr3* has been shown to be involved in regulating axonal morphology[30,31]. The five common genes might mediate functions that are shared between mossy fibres and climbing fibres that are both directed to CBX, whereas the larger numbers of genes that are not shared might be related to distinct functions of MY versus P and/or projections to cerebellar Purkinje cells versus granule cells, respectively. To understand how the DEGs in CBX-projecting neurons are regulated, we identified 223,839 hypo-DMRs in the HB that were associated with CBX-projecting neurons (Fig. 5e). These DMRs were further divided into subsets that were hypo-methylated in either P-to-CBX or MY-to-CBX, and only a limited number were hypo-methylated in both. Collectively, these findings suggest that the molecular mechanisms underlying CBX versus other projections in P and MY are largely distinct, but with some shared features at both the transcriptomic and epigenomic levels.

## AMY and MB neurotransmitters

We examined projections from the AMY to nine different targets, including the PFC, ENT, HIP, MOB, STR, TH, VTA, P and MY. These projections were enriched in 16 AMY clusters, with distinct sets of clusters enriched for neurons projecting to IT targets versus ET targets (Fig. 5f). The clusters enriched for IT projections were primarily glutamatergic and expressed *Slc17a7* and/or *Slc17a6* (Fig. 5g). By contrast, the clusters enriched for ET projections were divided between glutamatergic clusters that expressed *Slc17a6* and GABAergic clusters (Fig. 5g). Notably, the AMY-to-ENT projection was particularly distinct compared to other IT projections, exhibiting varied usage of vesicular glutamate transporters. Within the clusters enriched for AMY-to-ENT, *Slc17a7* was predominantly expressed in cluster 12, *Slc17a6* was the predominant transporter in clusters 24, 7 and 1, and clusters 31 and 64 expressed both *Slc17a7* and *Slc17a6*, suggesting a potential diversity in the physiology

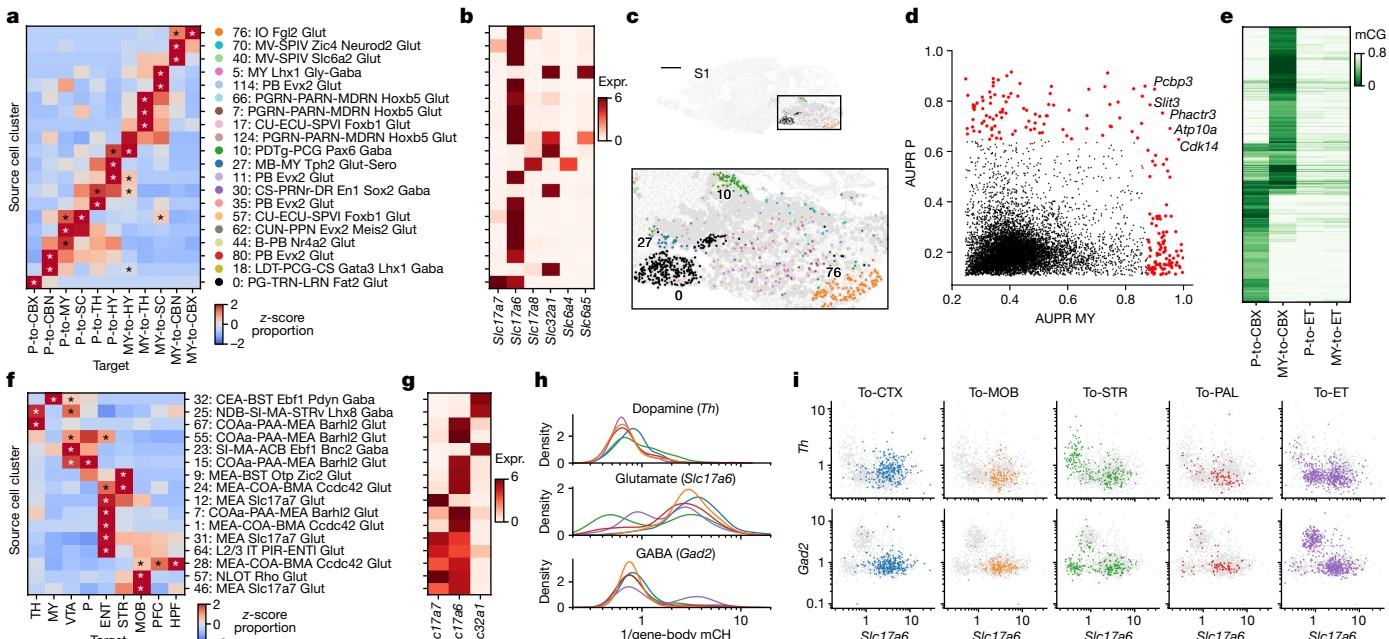

**Fig. 5 | Neurotransmitter usage in HB, AMY and VTA projection neurons.**
**a**,**b**, The proportion of each of the 11 assayed HB projections (**a**, z-score normalized across targets; the asterisk denotes enrichment with FDR < 0.01 (one-sided Fisher exact test, Benjamini–Hochberg procedure; Methods)) and the expression levels of six neurotransmitter transporter genes, encoding respectively VGLUT1, VGLUT2, VGLUT3, VGAT, SERT and GLYT2 (**b**), in each of the 20 HB projection-enriched clusters. **c**, Projection-enriched HB clusters mapped to the MERFISH slice S1. The colour palette for clusters is the same as in **a**. **d**, AUC of precision–recall (AUPR) of genes to distinguish the P-to-CBX cluster (0) and P-to-ET clusters (10, 11, 27, 30, 35, 44, 57, 62 and 80) versus the MY-to-CBX cluster (76) and MY-to-ET clusters (5, 7, 10, 11, 17, 57, 66 and 114) with gene-body mCH level in epi-retro-seq data. The genes with AUPR > 0.872 in P and AUPR > 0.647 in MY (>99th quantile) are coloured in red. Five genes selected in both P and MY are labelled. **e**, mCG levels of hypo-mCG DMRs (n = 22,3839) in

P and MY between the P-to-CBX clusters and P-to-ET clusters. **f**,**g**, The proportion of each of the 9 assayed AMY projections (**f**, z-score normalized across targets; the asterisk denotes enrichment with FDR < 0.01 (Fisher exact test, Benjamini–Hochberg procedure; Methods)) and the expression levels of neurotransmitter transporters *Slc17a7*, *Slc17a6* and *Slc32a1*, encoding respectively VGLUT1, VGLUT2 and VGAT (**g**), in each of the 16 projection-enriched clusters. **h**,**i**, The gene-body mCH levels of tyrosine hydroxylase (*Th*), *Gad2* and *Slc17a6* in VTA projection neurons, shown in density plots (**h**) or scatter plots (**i**). Colours represent VTA neurons projecting to different targets and the same palette is used in **h**,**i**. Note that the x axis in **h** and both axes in **i** are plotted as reciprocal mCH values (1/gene-body mCH), so low mCH is plotted to the right and top, indicating higher gene expression. ACA was not included in CTX (see Extended Data Fig. 9).

and function of AMY neurons projecting to the ENT. In summary, our results underscore the heterogeneity in neurotransmitters and their transporter utilization among AMY projection neurons.

The MB regions containing the VTA and substantia nigra (which we collectively refer to as VTA) exhibit some of the most notable and complex patterns of heterogeneous neurotransmitter usage between different projections. Our study analysed VTA neurons projecting to 16 different targets, including 6 cortical targets (PFC, MOp, SSp, ACA, RSP and PTLp), 6 other IT targets (MOB, ENT, PIR, AMY, STR and PAL) and 4 ET targets (TH, HY, SC and P). By integrating epi-retro-seq and unbiased snmC-seq data, as well as scRNA sequencing of VTA, we can distinguish between cell clusters with various combinations of the expected glutamate, GABA and dopamine transporters known to be expressed by VTA neurons[32–35] (Extended Data Fig. 9a,b).

To better examine the relationships between VTA neurons projecting to different targets and their use of neurotransmitters, we analysed the levels of mCH at specific marker genes, including tyrosine hydroxylase (*Th*) for dopaminergic neurons, *Gad2* for GABAergic neurons, and *Slc17a6* for glutamatergic neurons because previous studies showed that rodent VTA glutamatergic neurons mainly express *Slc17a6* but not *Slc17a7* or *Slc17a8* (Fig. 5h,i and Extended Data Fig. 9c,d; refs. 36,37). In general, VTA neurons that project to the cortex had lower levels of mCH at *Th* compared to subcortical projections (except for VTA-to-STR), suggesting a higher expression level of *Th* (Fig. 5h top; P values = 2.8 × 10⁻⁷ (CTX versus MOB), 3.0 × 10⁻⁵ (CTX versus PAL), 6.2 × 10⁻¹⁵ (CTX versus ET), two-sided Wilcoxon rank-sum tests). The CTX-projecting neurons also exhibited lower mCH levels at *Slc17a6*, indicating *Slc17a6*

expression (Fig. 5h middle). Therefore, these CTX-projecting VTA neurons are probably *Th*⁺ and *Slc17a6*⁺ and use both dopamine and glutamate (Fig. 5i and Extended Data Fig. 9c). In contrast to VTA-to-CTX neurons, the most prominent populations of VTA-to-STR neurons comprie two groups, *Th*⁺*Slc17a6*⁻ and *Th*⁻*Slc17a6*⁺, and there is a smaller proportion of neurons that are both *Th*⁺ and *Slc17a6*⁺ (Fig. 5i and Extended Data Fig. 9). (Note that the use of the "−" designation here indicates a relatively low expression level rather than a complete absence.) On the basis of their mCH levels, the ET-projecting neurons were generally divided into two subgroups: *Gad2*⁺ and *Slc17a6*⁺ (Fig. 5i). Among the ET-projecting VTA neurons, those projecting to TH and HY were more similar to each other than to those projecting to SC and P (Extended Data Fig. 9b). Notably, some of the SC- and P-projecting neurons were uniquely present in a VTA *Gad2*⁺ cluster that were absent in other projections (Extended Data Fig. 9b). Overall, our findings corroborate previous reports of diverse populations of VTA neurons that use single or combined neurotransmitters and highlight intricate patterns of distinct neurotransmitter usage among various projections.

## Summary

We have uploaded and made available data that inform potential users about the relationships between axonal projection status and DNA methylation at single-cell resolution for tens of thousands of neurons corresponding to hundreds of source-to-target combinations. We have provided quantitative measures of the discriminability of source neurons projecting to different targets for nearly 1,000 target-to-target

comparisons. We have further demonstrated how these data can be integrated with other single-cell data modalities, including scRNA-seq and MERFISH, to link the spatially resolved cell-type clusters to neural circuits. It is important to note that our experiments were designed to assess the methylation status of neurons projecting to relatively large targets that could be reliably injected and assessed for accuracy during dissections of fresh tissue, and from a large number of source regions that could be readily and reliably dissected. Integration with MERFISH data allowed for more precise anatomical localization of enriched clusters from these sources, but more focused studies using smaller retrograde tracer injections linked to smaller injection locations would be needed to identify possible differences between projection neurons at a finer resolution. More extensive details about the use of these data, their potential limitations and the analytic approaches we have taken can also be found in the Methods. The in-depth analyses provided here for both brain-wide comparisons of ET- versus IT-projecting neurons, and for the full sets of targets assayed for six of the assayed source regions (HY, TH, P, MY, AMY and VTA), exemplify the utility of the much larger dataset for further brain-wide and source- or target-focused analyses.

The observations and analysis presented here, both across the whole brain and for selected regions, provide new knowledge about the relationships between projection cell types and their epigenetics and gene expression. Overall, our data and analyses suggest that, as a general rule, the targets of projection neurons in any part of the brain can be predicted at levels above chance on the basis of knowledge of DNA methylation (for example, Extended Data Fig. 3). However, there is considerable diversity in the level of correlation between projection targets and methylation status. This diversity arises from differences in both source regions and targets. For example, cortical neurons projecting to ET versus IT targets can be readily identified for neurons from nearly any cortical area, and knowledge about correlations observed in one cortical area can be used to make predictions for another cortical area. Thalamic neurons projecting to ET versus IT targets can also be readily predicted, but knowledge from differential methylation of cortical neurons cannot be used to accurately predict the projection status of thalamic neurons. In contrast to cortical and thalamic sources, projections to ET versus IT targets cannot be predicted as reliably for neurons in some other sources such as PAL (Fig. 2b). It is also clear from our in-depth analyses of subsets of these data, including neurons in the HY and TH, that relationships between source locations, projection targets and methylation status of single neurons are complex. Although there are better than chance correlations between gene methylation and projection targets for neurons in all of the sources that we sampled, it is likely that the degree of correlation is shaped by a range of developmental events that affect both gene methylation and projection status through mechanisms that can work both independently and in concert. Future studies will be needed to better understand such developmental mechanisms in the context of the whole brain and to study which mechanisms are at work for each brain area and projection target.

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

## Methods

### Experimental animals

As described previously[2], all experimental procedures using live animals were approved by the Salk Institute Animal Care and Use Committee. The knock-in mouse line, R26R-CAG-loxp-stop-loxp-Sun1-sfGFP-Myc (INTACT) used in epi-retro-seq[2] was maintained on a C57BL/6J background. Adult male and female INTACT mice were used for the retrograde labelling experiments. Animals were housed in an Association for Assessment and Accreditation of Laboratory Animal Care-accredited facility at the Salk Institute. Lighting was controlled on a 12 h light/12 h dark cycle. Temperature was monitored and adjusted in accordance with Guide for the Care and Use of Laboratory Animals. Humidity was not controlled but monitored. As all air coming in is 100% fresh air (not recirculated), humidity in the animal facilities is approximately the same as the outside ambient air. San Diego averages 40–60% humidity year-round. Animals were 35–54 days old at the time of surgery for viral vector injections, were killed 13–17 days later, and were 50–70 days old on the day of dissection. C57BL/6J 'wild-type' mice aged 56–63 days were used for MERFISH experiments.

### Surgical procedures for viral vector injections

As described previously[2], to label neurons projecting to regions of interest, injections of rAAV-retro-Cre (produced by Salk Vector Core or Vigene, $2 \times 10^{12}$ to $1 \times 10^{13}$ viral genomes per millilitre, produced with capsid from Addgene plasmid No. 81070 packaging pAAV-EF1a-Cre from Addgene plasmid No. 55636) were made into both hemispheres of the INTACT mice. In summary, animals were anaesthetized with either ketamine–xylazine or isoflurane and placed in a stereotaxic frame. Pressure injections of 0.05 to 0.4 μl of AAV per injection site were made using glass micropipettes (tip diameters about 10–30 μm) targeted to stereotaxic coordinates corresponding to MOp, SSp, ACA, AUDp, RSP, PTLp, VISp, HPF, MOB, STR, PAL, TH, SC, VTA + substantia nigra, P, MY and CBX. To precisely target PFC, agranular insular cortex, ENT, PIR, AMY, HY and CBN, AAV was injected using iontophoresis to ensure confined viral infection. Iontophoretic injections (+5 μA, 7 s on/7 s off cycles for 5–10 min) were made with glass pipettes with a tip diameter of about 10 μm. For most of the desired target areas, injections were made at different depths, and/or at different anterior–posterior or medial–lateral coordinates to label neurons throughout the target area. More detailed injection coordinates and conditions are listed in Supplementary Table 1. At least two male and two female mice were injected for each desired target. No sample size calculation was carried out. We empirically determined to use two mice of the same sex for each injection to achieve minimum reproducibility. Animals used for injections into each brain area were selected at random.

### Brain dissection

Brain dissections were carried out as described previously[2]. In summary, approximately 2 weeks after the AAV-retro-Cre injection, brains were extracted from the 50- to 70-day-old INTACT mice, immediately submerged in ice-cold slicing buffer (2.5 mM KCl, 0.5 mM CaCl$_2$, 7 mM MgCl$_2$, 1.25 mM NaH$_2$PO4, 110 mM sucrose, 10 mM glucose and 25 mM NaHCO$_3$) that was bubbled with carbogen, and sliced into 0.6-mm coronal sections starting from the frontal pole. From each AAV-retro-Cre-injected brain, the slices were kept in the ice-cold dissection buffer from which selected brain regions (Fig. 1b) were manually dissected under a fluorescence dissecting microscope (Olympus SZX16), following the Allen Mouse Common Coordinate Framework, Reference Atlas, Version 3 (2015; Extended Data Fig. 1). The dissected brain tissues were transferred to prelabelled microcentrifuge tubes, immediately frozen in dry ice, and subsequently stored at −80° C. During the dissection, the injection site was visually inspected to verify the accuracy of the injection. Only brains with accurate injections were dissected for further analysis. Olympus cellSens dimension 1.8 was used for image acquisition.

### Nucleus preparation and single-nucleus isolation

Nucleus preparation and isolation were carried out as described previously[2]. In summary, for each dissected brain region, samples from two male and two female mice were pooled separately as biological replicates for nucleus preparation. Nuclei were prepared using a modified protocol as reported[38] and described[2] previously. Nucleus suspensions were then incubated with GFP antibody, Alexa Fluor 488 (Invitrogen, A-21311, 1:500 dilution) and anti-NeuN antibody (EMD Millipore MAB377) conjugated with Alexa Fluor 647 (Invitrogen A20173; 1:300 dilution). GFP$^+$NeuN$^+$ single nuclei were isolated using FANS on a BD Influx sorter or a BD Aria Fusion cell sorter with a 100-μm nozzle, and sorted into 384-well plates with digestion buffer for snmC-seq. BD Influx Software v1.2.0.142 was used to select cell populations. The collected plates were incubated at 50° C for 20 min and then stored at −20° C. FACS parameters were adjusted for each experiment on the basis of the density of labelled neurons such that a higher proportion of all labelled neurons would be recovered from regions with sparser labelling.

### snmC-seq library preparation

The bisulfite conversion and library preparation were carried out following the detailed snmC-seq protocol as previously described[15]. In brief, DNA samples from single nuclei were barcoded with random primers after the bisulfite conversion, pooled through two rounds of cleaning up with SPRI beads, and then added with adapters and PCR amplified to generate the libraries. Libraries were then pooled, cleaned up with SPRI beads, normalized and sequenced on Illumina Novaseq 6000 using the S4 flow cell 2 × 150 base-pair mode. Freedom EVOware v2.7 was used for library preparation, and Illumina MiSeq control software v3.1.0.13 and NovaSeq 6000 control software v.1.6.0 and Real-Time Analysis v3.4.4 were used for sequencing. Technicians doing nucleus preparations and snmC-seq analyses were blind to the injection sites used for each sample.

### Mapping and preprocessing

Epi-retro-seq data were mapped to the mm10 genome as described in our previous study[39]. The whole genome was parsed into 100 kb non-overlapping genomic bins (Chr. 1:0–100,000; Chr. 1: 100,000–200,000; and so on) using bedtools make-window, and for each single cell, we counted the methylated and total basecalls for all 100-kb bins using ALLCools generate-dataset. We also carried out the same counting on all gene bodies expanded 2 kb in both directions. The data are saved in Zarr format to allow chunk loading and on-disk computing[40]. To avoid the methylation differences being driven by the active and inactive X chromosomes, we used only the autosomal bins and genes in our analyses. The cell-by-bin and cell-by-gene posterior methylation levels were computed as previously described[39], which is the input for all downstream analyses.

### Quality control

In quality control (QC) step 1, the cells included in the analysis are required to have a median mCCC level of the experiment < 0.025; 500,000 < nonclonal reads < 10,000,000; and mCCC level < 0.05. In total, 56,843 cells from 703 experiments satisfied these requirements (Extended Data Fig. 2a,b).

In QC step 2, the potential doublets were removed as described in the next section, and 48,032 cells remained in the dataset (Extended Data Fig. 2c,d). The cell-type information and dissection information for these cells were used in our analysis, but further filters were applied to exclude non-neuronal cells as well as neurons whose projection targets are not confidently assigned.

In QC step 3, the experiments with fewer than 20 neurons were excluded to ensure the statistical power of projection analysis, resulting in 39,461 cells from 519 experiments left. The non-neuronal cells were

also removed from the dataset, after which 34,643 neurons remained. The cell-type classification method is described in the next section.

In QC step 4, the cortical cells from 286 experiments were further filtered to exclude the experiments with a high proportion of neurons of the cell types known not to project to the intended injection site (off-target clusters), using the same method as in our previous study[2]. Specifically, for each FANS run, we counted the number of neurons that were observed in known on-target cell types ($O_{on}$) and off-target cell types ($O_{off}$). Assuming that the proportions of contaminated cells in each subclass would be similar to those of a sample without projection-type enrichment, we compared the observed counts to the counts from unbiased snmC-seq data ($E_{on}$ and $E_{off}$) collected from the corresponding dissections in Extended Data Fig. 1. The fold enrichment was computed as $\frac{O_{on}E_{off}}{O_{off}E_{on}}$. A one-sided exact binomial test of goodness-of-fit was used to determine whether the enrichment of on-target cells was significant. The $P$ value was computed as $\Pr(X \geq O_{on}; n, p)$, in which $X \sim \text{binomial}(n, p)$, where $\sim$ represents distributed as, $n = O_{on} + O_{off}$ and $p = \frac{E_{on}}{E_{on} + E_{off}}$. For each ET target, we considered ET as on-target subclasses and IT+inhibitory neurons as off-target. The thresholds for fold enrichment and FDR (Benjamini–Hochberg procedure) were 8 and 0.001. For IT targets, we considered IT as on-target subclasses and layer 6 corticothalamic+inhibitory neurons as off-target. The thresholds for fold enrichment and FDR (Benjamini–Hochberg procedure) were 3 and 0.001. This eliminated 32 out of 286 sorting cases (Extended Data Fig. 2e).

The rationale of QC step 4 is to remove potential contamination in the dataset that might have resulted from inaccurate gating of GFP⁺NeuN⁺ cells and AAV-retro-Cre injection pipettes that passed through overlying source brain regions and directly labelled neurons at those sources rather than being taken up retrogradely from the intended target. Inaccurate gating of GFP⁺NeuN⁺ cells could be more common in the experiments of some weak projections, in which very few neurons were retrogradely labelled, resulting in small proportions of cells passing FANS gating criteria and subsequent inclusion of high proportions of cells accepted from the edges of FANS gates. Inaccurate labelling could be more common when targeting a deep structure in the brain (for example, TH or HY) and collecting cells from the superficial structures directly above the target (for example, cortex). Note that QC step 4 was carried out only for experiments on isocortical neurons, given that the on-target and off-target clusters were relatively clear in these areas. For subcortical projections, comprehensive prior knowledge of molecular cell types associated with projection is usually lacking, which makes the estimation of contamination using this method more challenging. The projections profiled in the subcortical structures are usually strong and do not involve overlaying of sources and targets, which would potentially lead to a lower noise level in those data. Nevertheless, it is worth noting that even after these QC steps, there are still expected to be some contaminated cells remaining in the dataset.

After all of the QC steps, 33,304 neurons from 487 experiments were used for analyses related to projection targets.

### Transfer of cell labels from one dataset to another with weighted $k$-nearest neighbours

This method is similar to the label transfer method in Seurat v3 (ref. 41), and implemented in our ALLCools python package. This is used in many analyses throughout the manuscript, including epi-retro-seq cell classification and doublet removal, and mapping of MERFISH cells and retro-seq cells into major dissection regions or RNA and mC co-clusters. The original Seurat method identified anchors between two datasets, and used the 100 nearest anchors for each cell in the unlabelled dataset to average the information from the labelled dataset. As the 100 anchors usually include cells from other clusters, especially for a cell in an underrepresented cluster, this method makes the label transfer of small clusters quite noisy. Instead of using the anchors between datasets to transfer the labels, we used the anchors only to integrate the datasets together, and directly find the neighbouring cells of the unlabelled dataset in the labelled dataset on the integrated space. As the larger dataset usually has more cells than the number of anchors, this method reduced the noise in the small clusters.

Assume we have two datasets in a co-embedding space, A with labels and B without labels. For each cell in B as a query cell, we first find its $k$-nearest neighbours in A with Euclidean distance, and denote its distances to the neighbours as a $k$-dimensional vector $\mathbf{d}$. $\mathbf{d}$ is then transformed to $\mathbf{w}$ as the weights for averaging the information from the neighbours through the following steps that are the same as in Seurat: $\mathbf{d}' = 1 - \frac{\mathbf{d}}{\max(\mathbf{d})}; \mathbf{d}'' = 1 - e^{-\frac{\mathbf{d}'}{2}}; \mathbf{w} = \frac{\mathbf{d}''}{\Sigma \mathbf{d}''}$. After the transformation, the closer neighbours have higher weights, and the weights of all neighbours sum up to 1. To transfer a categorical label from A to B, we used one-hot encoding to represent the label and the label vectors corresponding to the $k$ neighbours in A of the query cell ($k$-by-#categories, denoted as $L_{ref}$) were averaged with the weights $\mathbf{w}$. The resulting vector $\mathbf{L}_{qry} = \mathbf{w}L_{ref}$ represents the probability of the query cell belonging to each category. The category with the maximum probability is used as the final assignment.

### Cell classification and doublet removal

As described in our companion manuscript, the cell clustering of the unbiased dataset was carried out iteratively at four levels (L1–L4), which assigned the cells into 61 (L1), 411 (L2), 1,346 (L3) and 2,573 (L4) clusters, respectively. At each level, the highly variable 100-kb bins were selected, and principal component analysis (PCA) was used for dimension reduction. The significant principal components (PCs) from mCH and mCG were combined to carry out consensus clustering.

We first carried out doublet removal with the help of unbiased data. The 56,843 cells after QC step 2 are mapped to the 310,605 unbiased snmC-seq cells (including predicted doublet cells). We used the highly variable features selected in the unbiased data and the PCA model fit with the unbiased data to transform the epi-retro-seq to the same dimension reduction space as the unbiased data. Then we classified the epi-retro-seq cells into either 1 of the 61 L1 clusters or the predicted doublet clusters defined in the unbiased data. The classification was carried out with the $k$-nearest neighbour approach described above on the PCs combining mCH and mCG. The epi-retro-seq cells assigned to each non-doublet L1 cluster were analysed in the next iteration, using the highly variable features selected in the unbiased data for the cluster and the PCA model fit with the unbiased data for the cluster. All of the predicted doublet cells in the unbiased data were added in each L1 cluster in the L2 clustering to further exclude the potential doublets. After these two iterations, the cells predicted to be doublets were removed, with 48,032 epi-retro-seq cells remaining. These cells were mapped to the 301,626 unbiased snmC-seq cells (without predicted doublets) with the same feature selection and PCA methods through the four levels, so each epi-retro-seq cell is assigned to 1 cluster at each level. The 61 L1 clusters were annotated on the basis of their dissection source and marker genes. The cell clusters representing non-neuronal cells were removed from further analyses. The cells corresponding to the IT, ET, corticothalamic and cortical inhibitory clusters in the L1 cluster annotation were used for QC step 4 as described above.

### Quantification of projection neuron difference with AUROC

To test the similarity of two groups of cells based on DNA methylation, we trained logistic regression models to predict the group label of each cell. We compared the results using four different types of feature to predict the projection target of neurons from the same source. These include the posterior mCH level of 100-kb-bin and gene-body, and the dimension reduction results of the two matrices. A total of 50 PCs were used as dimension reduction, with unbiased snmC-seq to fit the PCA models and transform the epi-retro-seq data. We also used two methods to split the cells into training and testing sets. One used a random selection of half of the cells projecting to each target for training and the

other half for testing (computational replicates); the other was based on the sex of the mice from which the cells were collected (biological replicates). After the QC steps, we have 168 source–target combinations with data from both sexes and the other 57 with cells from only one sex. Therefore, all of the comparisons of 926 target pairs could be quantified with the computational replicates, but only 516 of them could be quantified with biological replicates. We noticed significant congruence of model performance between the different features and different train/test splits (Extended Data Fig. 3a–c). The performance when using 100-kb bins was very similar to that when using gene bodies (Extended Data Fig. 3a). The performance when using raw features was slightly better than that when using PCs (Extended Data Fig. 3b). The performance when using computational replicates was significantly better than that when using biological replicates (Extended Data Fig. 3c), which was expected given that the computational replicates dismissed the heterogeneity between biological replicates and made the predictions easier. Nevertheless, the computational replicates still provided strongly correlated results to biological replicates (Extended Data Fig. 3c), which allowed the comparison between different target pairs to evaluate their epigenomic differences.

All of the other results in the figures were computed using the computational replicates with gene-body mCH as features. The features were filtered on the basis of average read coverage across cells before the model training. We removed the 100-kb bins and genes with <500 average CH basecalls, resulting in 23,730 bins or 9,906 genes in the model. Sci-kit learn was used for model implementation. The area under the receiver operating characteristic (AUROC) from cross-validation was used to measure the performance of the model. The higher AUROC represented the better ability of the model to present the group label, which indicated that the two groups had larger mCH differences and were more distinguishable. For computational replicates, we carried out random sampling 50 times with different seeds, and used the average AUROC as the final result.

To test the predictability of projection targets with genes from different categories, we collected the genes from the following resources—neuropeptides and receptors: Table 1 in ref. 42 and Supplementary Fig. 16 in ref. 43; neurotransmitter receptors: Supplementary Fig. 15 in ref. 43; ion channels: Supplementary Fig. 14 in ref. 43 and the Guide to PHARMACOLOGY database (https://www.guidetopharmacology.org/DATA/targets_and_families.csv); neural projection development: Gene Ontology terms GO0031175 Neuron Projection Development and GO0050808 Synapse Organization; TFs: annotation from SCENIC+ (ref. 44). Only genes included in 9,906 genes with high CH coverage were analysed, and adding more lower-coverage genes to increase the size of gene sets did not improve the prediction performance.

Several reasons could contribute to a low prediction performance. Biological reasons would include the following. First, some neurons make projections to several targets simultaneously. These could result in the neurons being captured by several retrograde labelling experiments of different targets. It would be impossible to predict a single label with our pairwise models for this type of neuron. Second, some neurons project to different target regions but have tiny epigenetic differences. To systematically distinguish the first and second reasons, other anatomic and genetic validations are still needed.

Technical reasons would include the following. First, the contamination levels of some experiments might be relatively high, which make larger noise and hinder the models from capturing real projection differences. Second, the epigenetic differences between neurons projecting to different targets vary across replicates. Third, the sample sizes of some projections are small, which makes learning more challenging. Fourth, the models are not powerful enough to capture the complex differences between projections.

Elimination of contaminated FANS runs in QC step 4 decreased the potential influence by the first technical reason for cortical neurons as discussed in the "Quality control" section, although there are still contaminated cells included in the dataset. The improvement in labelling efficiency and specificity would help to better solve the molecular differences between projection types. In this study, male and female mice were treated as biological replicates after removing sex chromosomes. Although methylation patterns of autosomes are similar, differences between sexes or animals might still exist. The small differences in performance between data splitting methods (based on computation or biological replicates) might suggest a less notable effect contributed by the second technical reason in those samples. To evaluate the potential limitation of the fourth technical reason, more carefully curated models, and accordingly, more samples, would be required. Thus, given all of these factors, we are generally more confident in the distinguishable target pairs when training and testing sets were split on the basis of both computational and biological replicates. The interpretation of comparisons without biological replicates and the indistinguishable pairs would need to be more careful and are not involved in the major conclusions in this manuscript. Our study aims to provide a general view across many sources and targets. A more detailed understanding of specific projections would require larger-scale profiles on those specific projection types.

## Integration between snmC-seq, epi-retro-seq and scRNA-seq

snmC-seq and scRNA-seq data used in this study are comprehensive atlases of the whole mouse brain, so most of the cell types are expected to be present in both datasets. Therefore, the two datasets were integrated on the basis of a canonical correlation analysis (CCA) framework, which captures the shared variation between the two datasets[41]. Epi-retro-seq is a projection-enriched dataset that contains a subset of the cell types in the atlas, but the shared methylation modality with snmC-seq allowed it to be integrated with the comprehensive atlas with a reciprocal PCA framework. Both the epi-retro-seq and the scRNA-seq datasets were mapped to the dimension reduction space of the snmC-seq data to create a multi-modality atlas of each brain region group.

For each region group, we selected cells from the three datasets belonging to the dissection regions. The methylation cells in the L1 clusters corresponding to cerebellar neurons were excluded from the analysis of cerebral and brainstem regions. The RNA cells from the major classes of non-neuronal cells and immature neurons, and the subclasses of cerebellar neurons, were excluded from the analyses. The RNA cells from subclasses of medial mammillary nucleus (MM) and dorsal cochlear nucleus (DCO) were also excluded owing to the dissection differences between the two studies.

The gene expression levels of scRNA-seq cells were normalized by dividing the total unique molecular identifier (UMI) count of the cell and multiplying the average total UMI count of all cells, and then log transformed. The posterior gene-body mCH levels of snmC-seq and epi-retro-seq cells were used. The cluster-enriched genes (CEGs) were identified in each L4 cluster. We checked the variance of the mCH CEGs among the snmC-seq cells and scRNA-seq cells and used only the CEGs with mCH variance greater than 0.05 and expression variation greater than 0.005 for the analyses. The opposite of mCH levels was used for snmC-seq and epi-retro-seq data owing to the negative correlation between gene-body DNA methylation and gene expression. We fitted a PCA model with the snmC-seq cells and transformed the epi-retro-seq cells and scRNA-seq cells with the model. The PCs were normalized by the singular value of each dimension to avoid the embedding being driven by the first few PCs.

We adopted a similar framework to that of Seurat v3 (ref. 41) for data integration by first identifying the mutual nearest neighbours as anchors between datasets, and then aligning the datasets through the anchors.

To find anchors between snmC-seq and scRNA-seq, we first $z$-score scaled the mCH matrix and expression matrix of CEGs across cells, and the resulting matrices are represented as $X$ (mC cell-by-CEG) and $Y$

(RNA cell-by-CEG), respectively. CCA was used to find the shared low-dimensional embedding of the two datasets, solved by singular value decomposition of their dot product $USV^T = XY^T$. $U$ and $V$ were normalized by dividing the $L^2$ norm of each row, and were used to find five mutual nearest neighbours as anchors and score anchors using the same method as Seurat v3.

The original CCA framework of Seurat (v3) is difficult to scale up to millions of cells owing to the memory bottleneck, since the mC cell-by-RNA matrix was used as the input to CCA. To handle this limitation, we randomly selected 50,000 cells from each dataset ($X_{ref}$ and $Y_{ref}$) as a reference to fit the CCA and transformed the other cells ($X_{qry}$ and $Y_{qry}$) onto the same canonical correlation space. Specifically, the canonical correlation vectors (CCVs) of $X_{ref}$ and $Y_{ref}$ (denoted as $U_{ref}$ and $V_{ref}$) were computed by $U_{ref}SV_{ref}^T = X_{ref}Y_{ref}^T$, for which $U_{ref}^TU_{ref} = I$ and $V_{ref}^TV_{ref} = I$. Then the CCV of $X_{qry}$ and $Y_{qry}$ (denoted as $U_{qry}$ and $V_{qry}$) were computed by $U_{qry} = X_{qry}(Y_{ref}^TV_{ref})/S$ and $V_{qry} = Y_{qry}(X_{ref}^TU_{ref})/S$. The embeddings from the reference and query cells were concatenated for anchor identification.

To find anchors between snmC-seq and epi-retro-seq, we used the snmC-seq data to fit a PCA model and use the model to transform epi-retro-seq cells to the same space and find the five nearest snmC-seq cells for each epi-retro-seq cell. Reciprocally, we fit another PCA model with the epi-retro-seq cells and transform the snmC-seq cells and find five nearest epi-retro-seq cells for each snmC-seq cell. The mutual nearest neighbours between the two datasets were used as anchors and scored using the same method as Seurat v3.

The PCs derived from the previous step were then integrated together using the same method as Seurat v3 through these anchors. This integration step projects the PCs of epi-retro-seq and scRNA-seq (query) to the PCs of the snmC-seq (reference) while keeping the PCs of the reference dataset unchanged. The resulting PCs from the three datasets were used for $t$-SNE visualization and $k$-nearest neighbour ($k = 25$) graph construction with Euclidean distance. The joint clustering was carried out with the Leiden algorithm on the graph using a resolution of 1.0.

The quality of the integration analysis was evaluated from two aspects. First, we visualized the different modalities in the co-embedding space (Extended Data Fig. 4 left). The local neighbour-hoods of the co-embedding usually contain cells from all modalities, suggesting a good mixture between the three datasets after integration. Second, we computed the proportion of cells in each mC cluster (Extended Data Fig. 4 middle) or RNA cluster (Extended Data Fig. 4 right) assigned to each cluster defined on the co-embedding space (co-cluster). As we used the highest granularity of clustering from individual modalities (original cluster), the co-clusters were usually larger than the original clusters. We therefore used the proportion of original clusters rather than the proportion of co-clusters, to demonstrate that almost all original clusters are included in one co-cluster with low ambiguity. The strongest signals align on the diagonals suggesting that the co-embedding preserved the cluster structures that were originally present within each modality. Further evidence of integration quality was suggested by the downstream analyses, for which highly consistent cell-type specificity of marker-gene expression and gene-body mCH were observed (Figs. 3f,g and 4e,f and Extended Data Fig. 9a).

### Cluster associated with projection

For neurons projecting to each target within one source, we computed the proportion of these neurons in each joint Leiden cluster. The clusters with >5% of the cells were considered as associated with the projection. The clusters associated with at least one projection are shown in the heatmaps of Figs. 3–5 and Extended Data Fig. 5. The values in the heatmaps represent the proportion of projection neurons in each cluster, $z$-scored within each cluster across the projection targets. We used the Fisher exact test to quantify whether one projection has a similar proportion of cells in each cluster compared with all other projections. One-sided tests were used to select odds ratios greater than 1 (projection cells enriched in one cluster compared with other projections). FDR values < 0.01 are labelled in the heatmaps in Figs. 3–5 and Extended Data Fig. 5. We also used the Fisher exact test to quantify whether the male and female samples have a similar proportion of cells in each cluster, when the samples of both sexes passed the QC described above. Two-sided tests were carried out and FDR values < 0.01 are labelled in the heatmaps in Extended Data Fig. 5.

In general, there are two intuitive ways to quantify the enrichment of projection neurons in a cluster. One is to directly find the clusters with a high absolute proportion of epi-retro-seq neurons projecting to a target. The other is to find clusters captured at a significantly higher frequency in the projection-enriched data relative to the unbiased data. The two methods each have their advantages and shortcomings. For example, the contaminated cells from inaccurate labelling or gating are likely to have a similar distribution across clusters to unbiased profiling. So a comparison using unbiased data as a control might help exclude the contaminated clusters better. However, if most of the neurons from a projection type are in the clusters that are originally abundant cell types in the source, by comparing with unbiased data, we would miss the predominant clusters making the projection.

In this manuscript, we used the absolute proportions but not the relative ones to the unbiased data owing to the different profiling strategies between the two datasets. Although the epi-retro-seq samples and unbiased snmC-seq samples were dissected in the same way, we pooled the different dissections into the 32 different sources to carry out FANS and sequencing for epi-retro-seq, so that the proportion of cells from different dissection regions of the same source is likely to follow their proportions in the mouse brain. However, the unbiased snmC-seq profiled all of the dissection regions separately and sequenced the same number of cells in each dissection, which manually amplified the proportion of cells from the smaller or sparser dissection regions relative to the larger or denser ones, and limited the power to estimate the real proportion of neurons in each cluster from the sources.

As we pooled the two mice of the same sex before sequencing, we do not have biological replicates to study the different distribution across clusters of projection neurons from different sexes. For the identification of projection-enriched clusters and their differences between sexes, we used Fisher exact tests to find the proportion difference that treats each single cell as an independent sample but does not consider the consistency between biological replicates. This could lead to false discovery, so we consider the projection-enriched clusters with no significant sexual proportion differences as more confident ones. We also tested general linear models with binomial dependent variables and use biological replicates as a random effect. However, probably owing to the number of replicates being too small (1 or 2), the limited power for accessing random effects resulted in low detection power and the test was highly biased to identifying small clusters. Therefore, our dataset provides a general resource across the whole brain suggesting projection-associated cell types, and more biological replicates are needed to validate the patterns and investigate the sexual differences.

It is also worth noting that our integration and clustering strategies did not consider the projection target labels of neurons. Therefore, the granularity and boundary of unbiased clusters could be different from the actual projection-associated cell types. For example, some clusters may have only a small fraction enriched for certain projection neurons, whereas some clusters may have very similar projection enrichment patterns but still split into several in our analyses. Improved methods for clustering including the information of both molecular feature embedding and projection labels could further expand our understanding of the association between molecular cell types and neural projections.

### Classification of MERFISH cells into major brain regions and cell clusters

The MERFISH experiments were conducted as described in ref. 17, including the gene panel design, tissue preparation, imaging, data

processing and annotation. The dataset includes two sagittal slices (S1 and S2, with S1 being more lateral and S2 being more medial) and 14 coronal slices (C2, C4, C6, C8, C10, C12 and C14, roughly corresponding to slices 2, 4, 6, 8, 10, 12 and 14 in Extended Data Fig. 1, with two replicates for each slice, represented as R1 and R2). The same naming of slices was used throughout this manuscript (Figs. 3e, 4d and 5c and Extended Data Fig. 6).

The MERFISH cells were classified into subclasses and brain region groups by integration with scRNA-seq data. The 489 autosomal genes that overlapped between scRNA-seq and MERFISH datasets were used. We fitted a PCA model with the scRNA-seq cells and transformed the MERFISH cells with the model. The PCs were normalized by the singular value of each dimension. The cell-by-gene matrices were $z$-score normalized across cells within each dataset, and CCA was used to find anchors between the two datasets. We used 50,000 cells to fit the CCA and transformed the other cells as described above. The transformed PCs of MERFISH cells were then aligned to the PCs of scRNA-seq cells to derive a co-embedding between the two datasets. This co-embedding was used for label transfer of cell subclasses from scRNA-seq data to MERFISH data, considering 25 neighbouring scRNA-seq cells for each MERFISH cell.

The cells classified as non-neuronal and immature neuronal subclasses were excluded owing to lack of regional specificity, and the rest of cells from the two datasets were integrated again with the procedures described above to transfer the label of 14 brain region groups from the scRNA-seq neurons to the MERFISH neurons. The initial label assignment is noisy. Therefore, a smoothing step was carried out to refine the region group assignment. Specifically, for each MERFISH cell $i$, we found its 25 neighbours on the same slice (denoted as $Ns_i$) based on the spatial coordinates, and used $Ds_i$ to represent the corresponding distances between $i$ and its $j$th neighbour $Ns_{i,j}$. Similarly, we found the 25 scRNA-seq neighbours for each MERFISH cell $i$ based on the integration, and used $Dr_i$ to represent the distances. The distance matrices were transformed as described in the label transfer section, and the final spatial labels were transferred from the 25 RNA neighbours of each of the 25 spatial neighbours (625 scRNA-seq cells in total) to 1 MERFISH cell. The weight between the MERFISH cell $i$ and the $k$th scRNA-seq neighbour of its $j$th spatial neighbour was computed as $D''_{i,25j+k} = Ds''_{i,j}Dr''_{Ns_{i,j},k}$. $D''$ is row normalized and used as weights for label transfer as described in previous sections.

We note that this could also be achieved through registration of MERFISH DAPI images to the common coordinate framework. However, our companion works demonstrated that the procedure is relatively challenging and it is important to use cell types with known locations as landmarks to refine the registration.

Finally, the neurons from each region group were selected and integrated with the scRNA-seq cells from the same region group, using the same procedure as described above. The mC−RNA co-cluster labels were transferred from the scRNA-seq cell to the MERFISH cells. The MERFISH cells assigned to the MM and DCO subclasses in the last step were also excluded as those clusters were not included in the co-cluster analysis as described in the previous section.

### Comparison with Act-seq

The 10x scRNA-seq data in ref. 7 were downloaded from Mendeley Data (https://data.mendeley.com/datasets/ypx3sw2f7c/3), and are referred to as Act-seq in this section and Extended Data Fig. 7. The dataset contains 168,877 cells in total, among which 78,476 were labelled as neurons and were used to integrate to the unbiased scRNA-seq data for hypothalamic neurons, using the 5,314 CEGs of 1,891 L3 scRNA-seq clusters. PCA was fitted with scRNA-seq data and the Act-seq data were transformed. The CCA framework was used to find anchors between the two datasets, and the transformed PCs of Act-seq were aligned to the scRNA-seq PCs. We used the label transfer method described in the previous section to transfer the mC−RNA co-cluster labels from

scRNA-seq cells to the Act-seq cells, considering five neighbouring scRNA-seq cells for each Act-seq cell. The Act-seq cells with $Fos$ expression level > 0 were considered to be $Fos^+$ cells, and the proportions of $Fos^+$ cells were compared between control and each behaviour using one-sided Fisher exact tests. The $Fos$ expression levels were compared with one-sided Wilcoxon rank-sum tests.

Only the 23,345 Act-seq cells from 16 ventromedial HY (VMH) neuron clusters were considered as ventrolateral VMH cells (VMHvl) and were used for the behaviour-association studies in ref. 7. However, almost none of them correspond to projection-associated clusters in our data (Extended Data Fig. 7a–c). We further compared our projection-associated clusters with all of the neuron clusters profiled in ref. 7 and note that five clusters have corresponding clusters in the Act-seq data (Extended Data Fig. 7d, in red). Among them, clusters 4 and 64 showed weak but significant increases in proportions of $Fos^+$ cells labelled during certain behaviours (Extended Data Fig. 7f).

The generally weak associations between projection-associated and behaviour-associated clusters are probably due to the small overlap between the brain regions profiled in the two datasets, particularly the under representation of VMHvl neurons in epi-retro-seq data. Additionally, because there were far fewer cells profiled in epi-retro-seq versus Act-seq, the granularity of clusters used for projection association and behaviour association is different; this difference is particularly pronounced in VMHvl for which >10 times more cells were used for the behaviour-association study (Extended Data Fig. 7c–e). Therefore, further increasing the size of datasets to achieve higher granularity of cell typing in specific regions of interest could facilitate further association between molecular types with projections and behaviours. Our study aimed at a comprehensive view of a large number of projections across the whole brain and focused on targets that do not seem to receive strong input from VMH. This apparently limited the data overlap between these and limited the ability to make direct comparisons between studies.

### Comparison with retro-seq

The scRNA-seq data in ref. 8 were downloaded from the Gene Expression Omnibus with the identifier GSE133912. The retro-seq data were integrated to the unbiased scRNA-seq data for thalamic neurons, using the 5,404 CEGs of 1,128 L3 scRNA-seq clusters. PCA was fitted with scRNA-seq data and retro-seq data were transformed. The CCA framework was used to find anchors between the two datasets, and the transformed PCs of retro-seq were aligned to the scRNA-seq PCs. To compare the distribution of retro-seq cells and epi-retro-seq cells across thalamic cell clusters, we used the label transfer method described in the previous section to transfer the mC−RNA co-cluster label and the joint $t$-SNE coordinates from scRNA-seq cells to the retro-seq cells, considering five neighbouring scRNA-seq cells for each retro-seq cell.

### DEGs

The gene expression level of each single cell was normalized by the total UMI count of the cell and log transformed. We carried out pairwise comparisons between clusters associated with projection neurons. For each cluster pair, the $P$ values were derived with two-sided Wilcoxon rank-sum tests, and the fold change is computed as the ratio between the average expression level across cells in the two clusters. The genes with absolute value of $\log_2$ fold change greater than 1 and FDR (Benjamini−Hochberg procedure) values smaller than 0.01 were considered as differentially expressed. The DEGs from all cluster pairs were merged to generate the heatmaps in Figs. 3 and 4. Only the top 100 DEGs ranked by FDRs were used if there were more than 100 DEGs identified between a pair of clusters.

### DMRs and association with genes

The unbiased snmC-seq cells from each mC−RNA co-cluster were merged to generate pseudobulk methylation profiles. The epi-retro-seq

cells were not used owing to the different genome backgrounds of the mice to avoid confounding results. DMRs were identified within each region group between clusters using ALLCools. We then calculated the PCC between DMR mCG and gene mCH fraction. We shuffled the DMRs and genes within each sample to calculate the null PCC and estimate FDR. We filtered DMR–target edges with FDR < 0.01.

### TF motif enrichment

We used an ensemble motif database from SCENIC+ (ref. 44), which contained 49,504 motif position weight matrices (PWMs) collected from 29 sources. Redundant motifs (highly similar PWMs) were combined into 8,045 motif clusters through clustering based on PWM distances calculated by TOMTOM[45] by the SCENIC+ authors. Each motif cluster was annotated with one or more mouse TF genes. To calculate motif occurrence on DMRs, we used the Cluster-Buster[46] implementation in SCENIC+, which scanned motifs in the same cluster together with hidden Markov models.

Within each region group, we assign hypo-DMRs to each cluster if the mCG level of a DMR in the cluster is below the 10th quantile of all DMRs from the region group and below the 10th quantile of the mCG level of this DMR in all clusters from the region group. To carry out motif enrichment analysis, we used the recovery-curve-based cisTarget algorithm[44]. In brief, the cisTarget algorithm performed motif enrichment on the hypo-DMRs of each cluster by calculating the area under the recovery curve (AUC) for each motif, which is further normalized on the basis of all other motifs in the collection to calculate a normalized enrichment score. We used the cutoff AUC > 0.01 and normalized enrichment score > 3 to select enriched motifs. The TF-encoding genes shown in Figs. 3i and 4i were additionally required to have expression level > 0 and normalized mCH level < 1 in at least one cluster that its motif enriched in, to select the TFs that are likely to be expressed among a family of TFs showing the same motif enrichment scores.

### GRN analysis

GRNs were constructed in each major brain region, with genes (TF and non-TF) and DMRs as nodes and three types of edge. The GRNs were summarized into many triplet combinations, each of which contains a TF, a DMR, a target gene (including TFs) and edges between each other. In this analysis, we include only the TFs whose motif is significantly enriched in hypo-DMRs of any cluster in the brain region (as described in the "TF motif enrichment" section). We use only a union set of genes with absolute $\log_2$ fold change > 1 and FDR < 0.01 (described in the section "DEGs") in any pairwise comparisons between clusters. The weights of edges were computed on the basis of the PCC between nodes across clusters, and only the edges showing significant correlation in the shuffle tests (described in the "DMRs and association with genes" section) were kept. Before computation of the PCC, all data were quantile normalized within each cluster (across features). The edges include the following three categories: TF–DMR edges, connected if the PCC between TF expression and mCG level of DMRs is significant (FDR < 0.01) and the TF has a binding site in the DMR predicted by Cluster-Buster; TF–gene edges, connected if the PCC between TF expression and target gene expression is significant (FDR < 0.01); and DMR–gene edges, connected if the PCC between mCG level of DMRs and target gene expression is significant and the distance between gene transcription start site and the DMR is within 1 megabase.

Note that because the GRNs were constructed based only on correlations between data modalities and the binding motifs, they inevitably capture indirect and false-positive relationships. Perturbation experiments would be necessary to validate the connections between *cis* or *trans* regulators and target genes.

### Reporting summary

Further information on research design is available in the Nature Portfolio Reporting Summary linked to this article.

### Data availability

Raw and processed data are available at the Gene Expression Omnibus under accession code GSE230782. Processed data can be explored on our web portal: http://neomorph.salk.edu/epiretro. Other datasets used in this study include scRNA-seq (https://portal.brain-map.org/atlases-and-data/bkp/abc-atlas), snmC-seq and MERFISH (https://mousebrain.salk.edu/download), Act-seq (https://doi.org/10.17632/ypx3sw2f7c.3) and retro-seq (GSE133912). The mm10 genome was downloaded from https://hgdownload.soe.ucsc.edu/goldenPath/mm10/bigZips/.

### Code availability

The code for all of the analyses can be found at https://github.com/zhoujt1994/EpiRetroSeq2023.git.

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

**Acknowledgements** We are grateful to M. Nunn for help with management of the project. This work is supported by National Institute of Mental Health grant U19MH114831 to E.M.C. and J.R.E. under the BRAIN Initiative of the National Institutes of Health (NIH) and by National Eye Institute grant R01EY022577 to E.M.C. The Flow Cytometry Core Facility of the Salk Institute (Research Resource Identifier: SCR_014839) is supported by funding from NIH–National Cancer Institute Cancer Center Support Grant P30 014195 and Shared Instrumentation Grant S10-OD023689 (Aria Fusion cell sorter). J.R.E. is an investigator of the Howard Hughes Medical Institute. The 10x scRNA-seq was financed by the grant U19MH114830 to H.Z. under the BRAIN Initiative of the NIH.

**Author contributions** Contribution to research design: E.M.C., Z.Z., M.M.B., J.R.E., J.Z., X.J. and K.-F.L. Contribution to data collection: Z.Z., Y.P., A.B., A.R., W.N.L., E.W., C.-T.L., P.A.M., A.A., J.L.A.S., N.C., M.L., L.B., C.F., Z.Y., K.A.S., B.T., J.A., M.A.K., C.V., J.R.N., R.G.C., N.S.P., M.V., M.R., M.J., T.I., J.O., N.E., J.L., S. Cho, J.R., H.-H.H., A.P.-D., B.D., J.B.S., C.O., H.Z. and M.M.B. Contribution to data analysis: J.Z., Z.Z., E.M.C., M.W., H.L., Q.Z., Z.Y. and H.Z. Contribution to data portal: Z.P., J.Z., W.D. and S. Chen. Contribution to data archive and infrastructure: E.A.M., J.Z., Z.Z., E.A.M., Y.P., A.B. and A.R. Contribution to research coordination: Z.Z., E.M.C., M.M.B., J.R.E., J.Z., Y.P., X.J., E.W., C.-T.L., E.A.M. and K.-F.L. Contribution to writing manuscript: J.Z., Z.Z., E.M.C. and J.R.E.

**Competing interests** J.R.E. serves on the scientific advisory board of Zymo Research Inc.

**Additional information**
**Correspondence and requests for materials** should be addressed to Joseph R. Ecker or Edward M. Callaway.

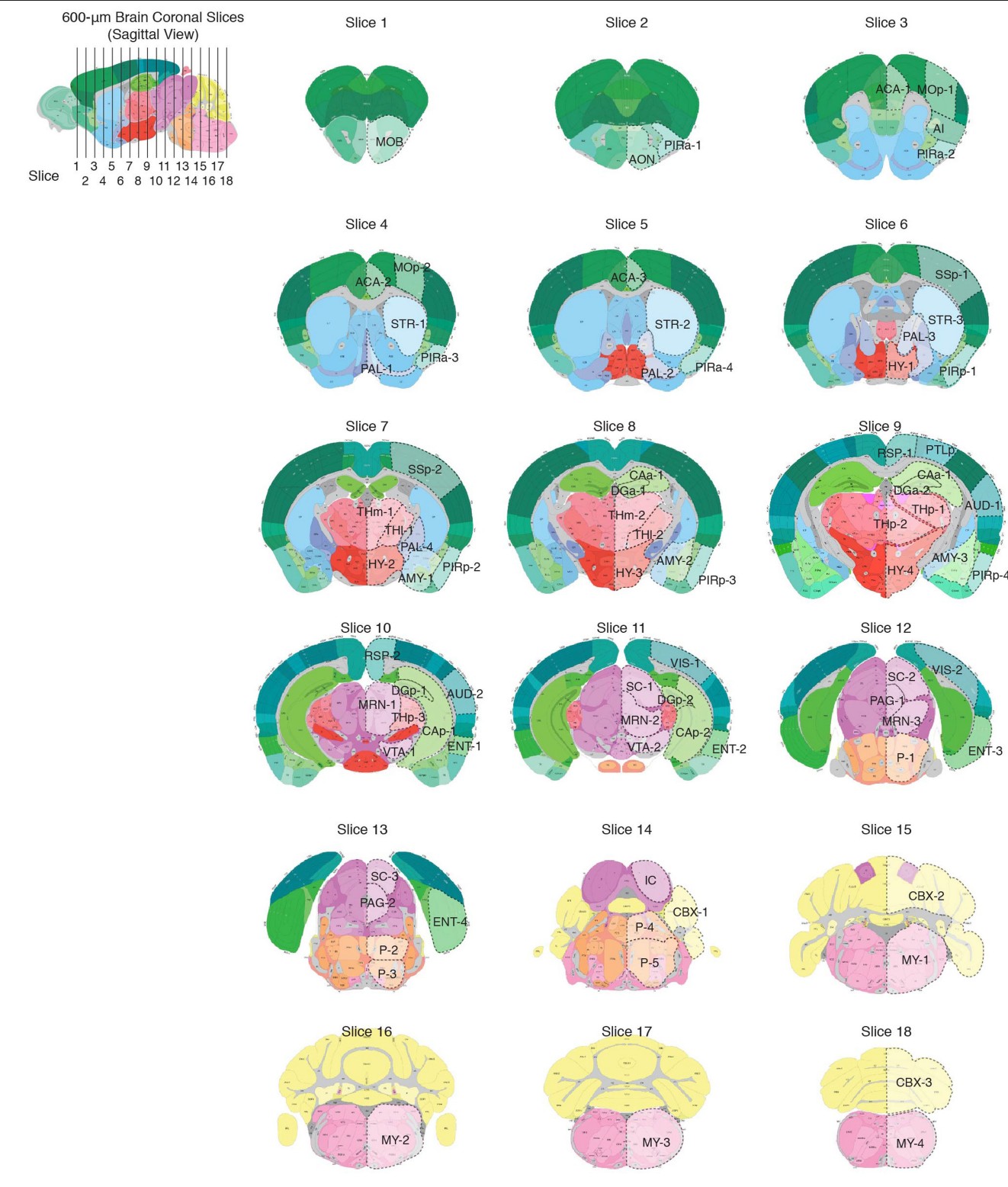

**Extended Data Fig. 1 | The dissection map of source regions across the mouse brain.** The posterior views of dissected slices are shown. The slices correspond to Allen Mouse Common Coordinate Framework (CCF), Reference Atlas, Version 3 (2020)[13], level 21 - 27 (slice 1), 27 - 33 (slice 2), 33 - 39 (slice 3), 39 - 45 (slice 4), 45 - 51 (slice 5), 51 - 57 (slice 6), 57 - 63 (slice 7), 63 - 69 (slice 8), 69 - 75 (slice 9), 75 - 81 (slice 10), 81 - 87 (slice 11), 87 - 93 (slice 12), 93 - 99 (slice 13), 99 - 105 (slice 14), 105 - 111 (slice 15), 111 - 117 (slice 16), 117 - 123 (slice 17), and 123 - 129 (slice 18), respectively. Regions dissected from each slice are indicated by dotted lines and are annotated. Allen Brain Reference Atlas, http://www.atlas.brain-map.org, © 2017 Allen Institute for Brain Science.

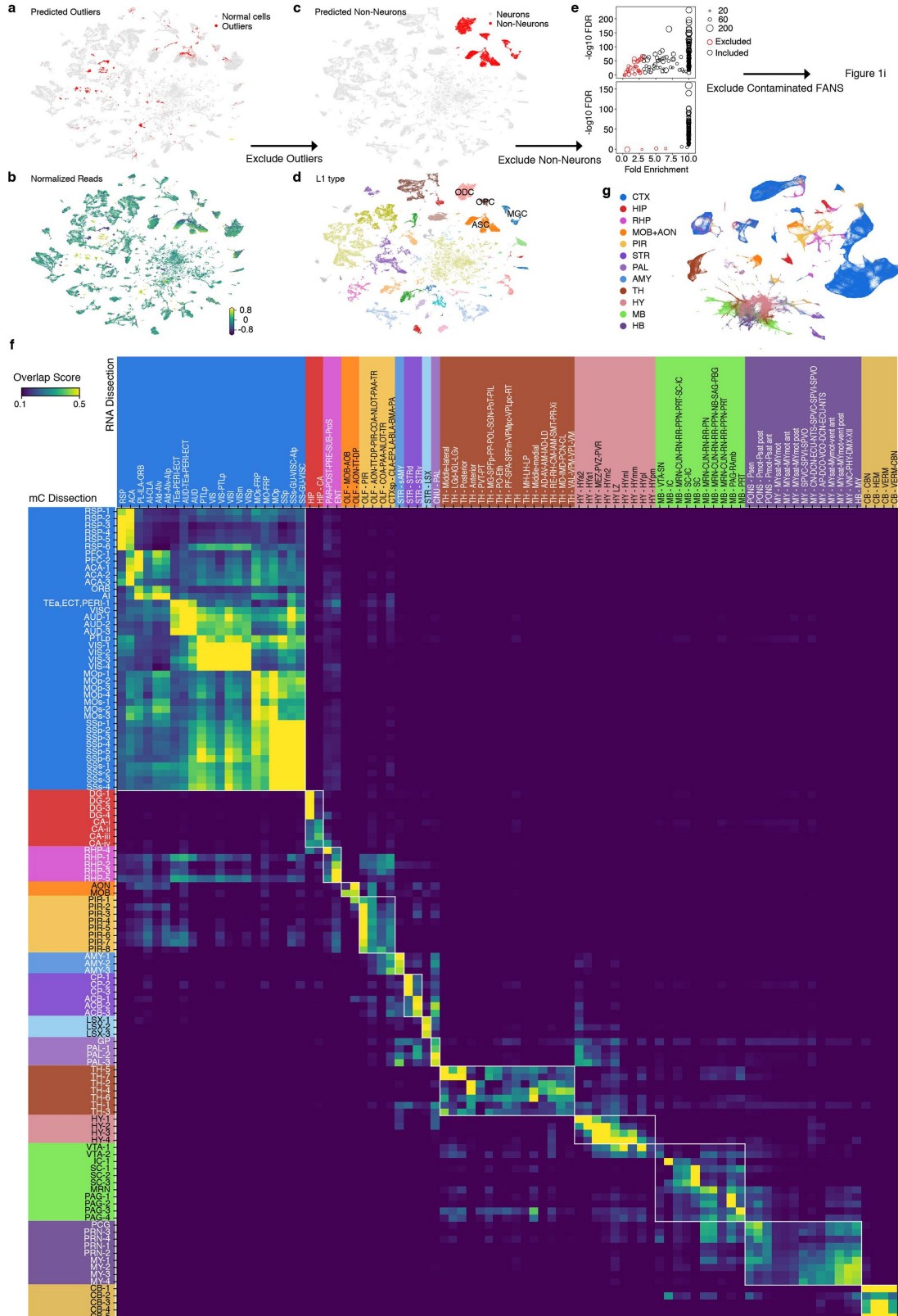

**Extended Data Fig. 2 | Quality control workflow and region group assignment. a, b,** Joint t-SNE of Epi-Retro-Seq cells (n = 56,843) and unbiased snmC-seq cells (n = 310,605) after basic QC (Methods, QC Step 1) colored by the predicted outliers (**a**) or the total number of reads normalized per sequencing plate (**b**). **c, d,** Joint t-SNE of Epi-Retro-Seq cells (n = 48,032) and unbiased snmC-seq cells (n = 301,626) after removing outlier clusters (Methods, QC Step 2) colored by neuronal vs. non-neuronal cells (**c**) or their assigned L1 type (**d**). **e,** the on-target vs. off-target fold enrichment (x-axis) and -log10 FDR (y-axis) of

IT (n = 186, top) or ET (n = 100, bottom) FANS experiments. The size of the circle is proportional to the number of neurons captured in the experiment. **f,** The overlap scores between 115 snmC-seq dissections and 87 scRNA-seq dissections. Each region group is colored differently on the x and y axes and squared in the heatmap. **g,** Joint t-SNE of Epi-Retro-Seq (n = 35,743), snmC-seq (n = 266,740), and scRNA-seq (n = 2,434,472) neurons colored by region groups. (**f**) and (**g**) share the same color palette.

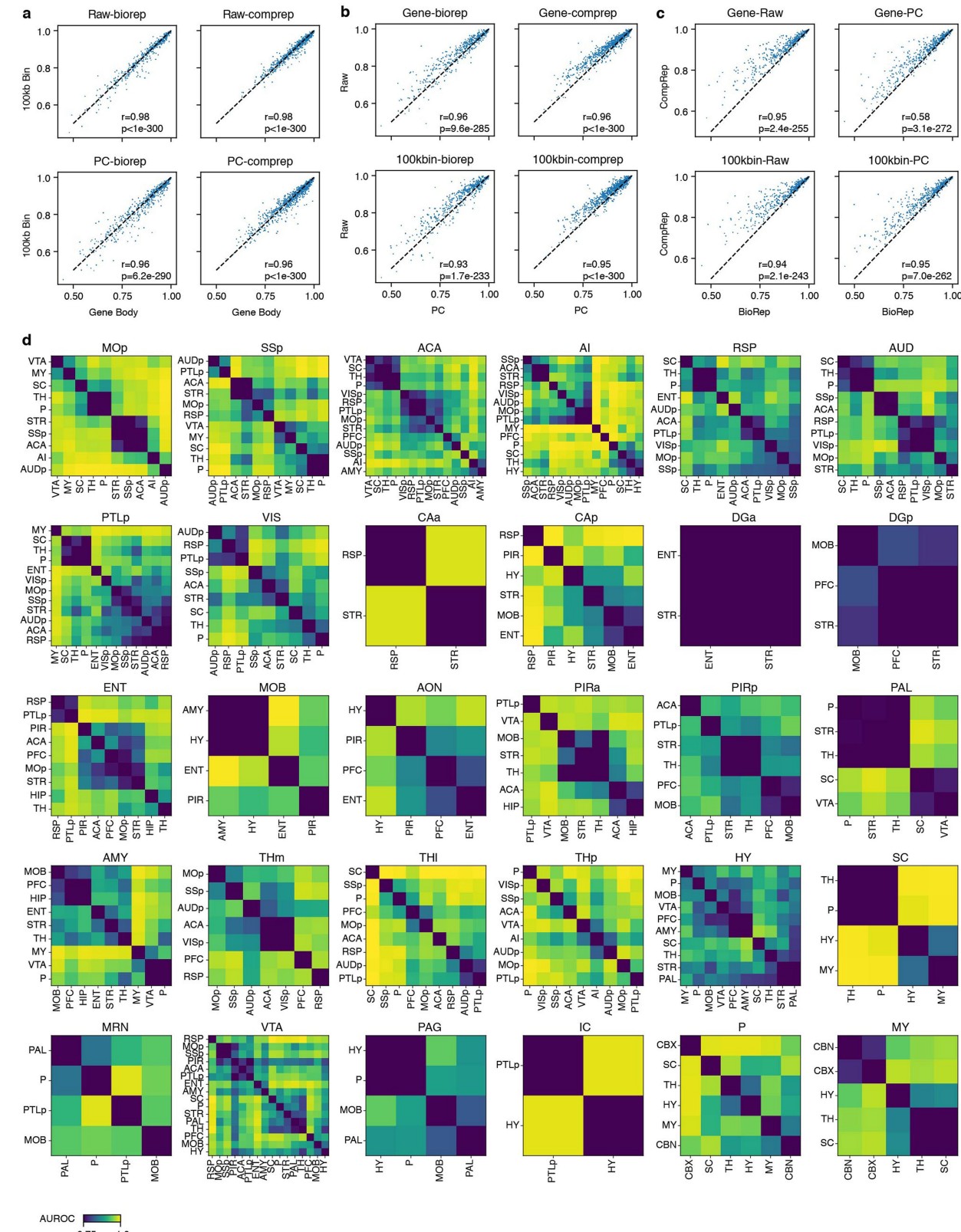

**Extended Data Fig. 3** | See next page for caption.

**Extended Data Fig. 3 | Quantification of target discriminability for all 926 target pairs from all source regions. a**, Comparisons of AUROCs from models trained with gene features (x-axis) or 100 kb bin features (y-axis) using posterior mCH level (top) or mCH principal components (bottom) when splitting the training and testing data according to biological replicates (left) or computational replicates (right). **b**, Comparisons of AUROCs from models trained with mCH principal components (x-axis) or posterior mCH level (y-axis) using gene (top) or 100 kb bin (bottom) features when split the training and testing data according to biological replicates (left) or computational replicates (right). **c**, Comparisons of AUROCs from models when splitting the training and testing data according to biological replicates (x-axis) or computational replicates (y-axis) using gene (top) or 100 kb bin (bottom) features and posterior mCH level (left) or mCH principal components (right). For **a-c**, each dot represents a pairwise comparison between the neurons projecting from the same source to two different targets. The plots involving biological replicates have 516 data points each while the others have 926 data points each. Pearson Correlation Coefficient (PCC, r) and *P* value (permutation test) are labeled in each panel. **d**, The AUROC between neurons projecting from each of the 30 sources to all possible pairs of targets that have been profiled for the source. STR and CBX are not included since we only profiled one target for these sources.

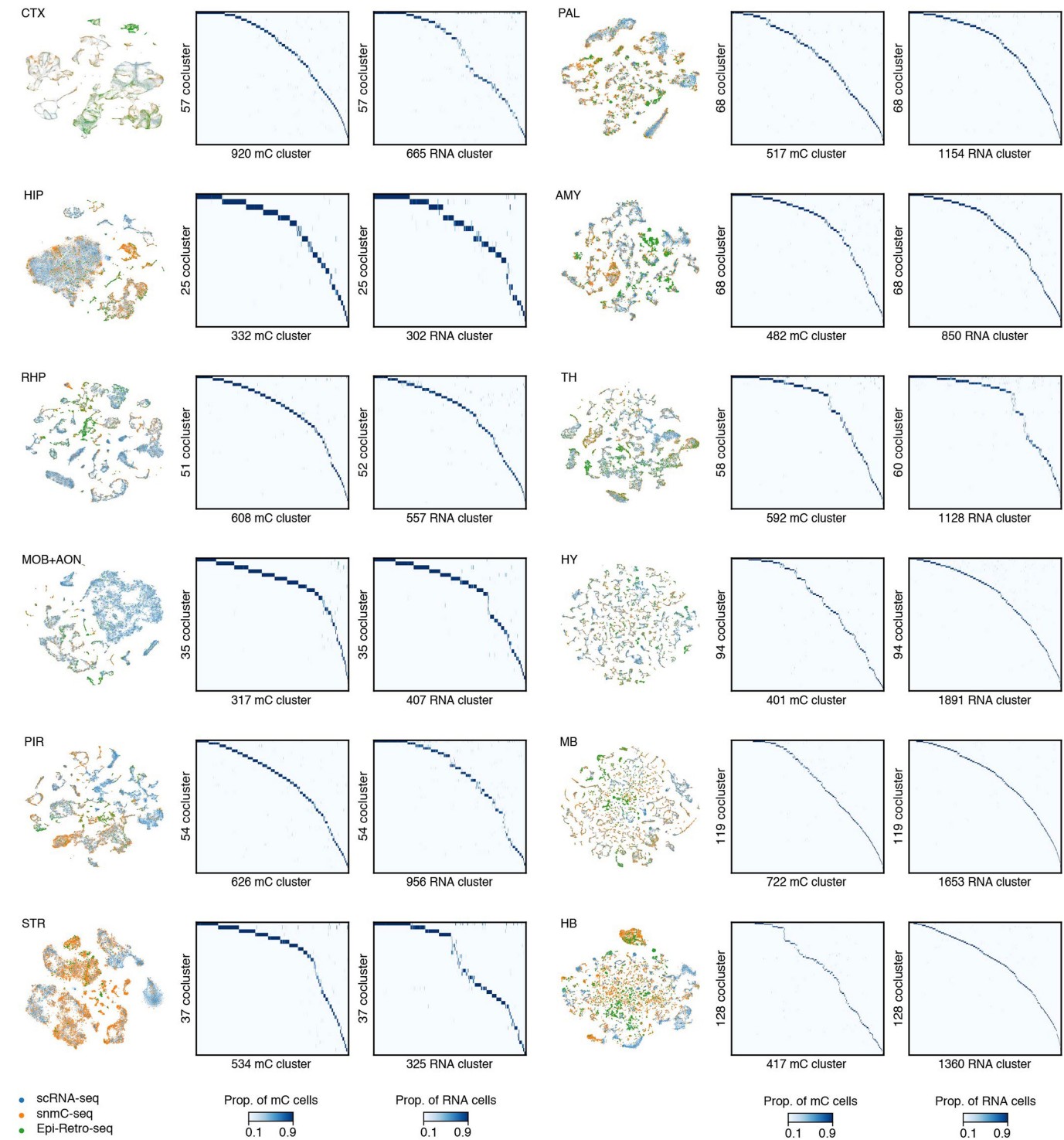

**Extended Data Fig. 4 | Co-clustering of Epi-Retro-Seq, unbiased snmC-seq and scRNA-seq data for 12 brain regions.** For each brain region group, the joint t-SNE colored by data modality (left) and the proportion of cells from each snmC-seq L4 cluster (middle, column) or scRNA-seq L3 cluster (right, column) within each co-cluster (middle and right, row). Numbers of co-clusters in the rows are sometimes different for middle and right columns because only the co-clusters (rows) with more than one value > 10% across the columns are shown. See Methods for further details.

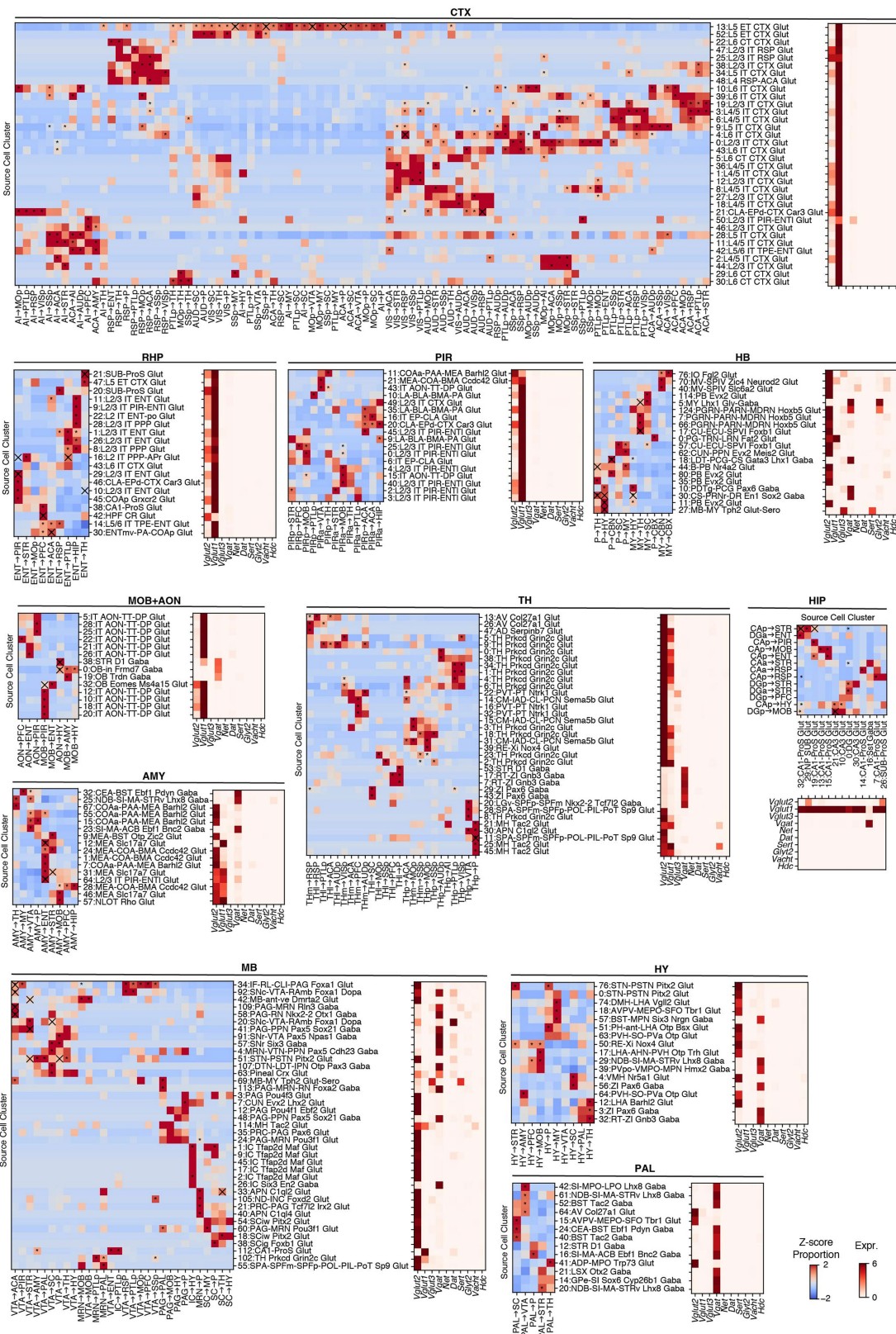

**Extended Data Fig. 5** | See next page for caption.

**Extended Data Fig. 5 | Projection-enriched cell clusters and their neurotransmitter usage for all brain regions.** Joint clustering analysis of Epi-Retro-Seq, unbiased snmC-seq and scRNA-seq was performed on each of the major brain region groups, including CTX, RHP, PIR, HB, MOB + AON, AMY, TH, HIP, MB, HY, and PAL (STR not included because there is only one target), to characterize neuronal cell clusters that were enriched for Epi-Retro-Seq projections. The normalized proportion of each projection in each cluster was visualized in the heatmaps (left) for each of the brain region groups. "*" denotes FDR < 0.01 for testing if the proportion of cells in a cluster among those projecting to a target is greater than the proportion of cells in this cluster among those projecting to all other targets (one-sided Fisher exact test; Benjamini-Hochberg procedure). "X" denotes FDR < 0.01 for testing if the proportion of cells in a cluster among those projecting to a target is different between male and female samples (two-sided Fisher exact test; Benjamini-Hochberg procedure). In addition, the expression levels of 10 marker genes for neurotransmitter usage in each cluster are visualized in the heatmap (right) for each brain region group. These genes included *Slc17a7* (*Vglut1*), *Slc17a6* (*Vglut2*), and *Slc17a8* (*Vglut3*) for glutamatergic neurons, *Slc32a1* (*Vgat*) for GABAergic neurons, *Slc6a2* (*Net*) for noradrenergic neurons, *Slc6a3* (*Dat*) for dopaminergic neurons, *Slc6a4* (*Sert*) for serotonergic neurons, *Slc6a5* (*Glyt2*) for glycinergic neurons, *Slc18a3* (*Vacht*) for cholinergic neurons, and histidine decarboxylase (*Hdc*) for histaminergic neurons.

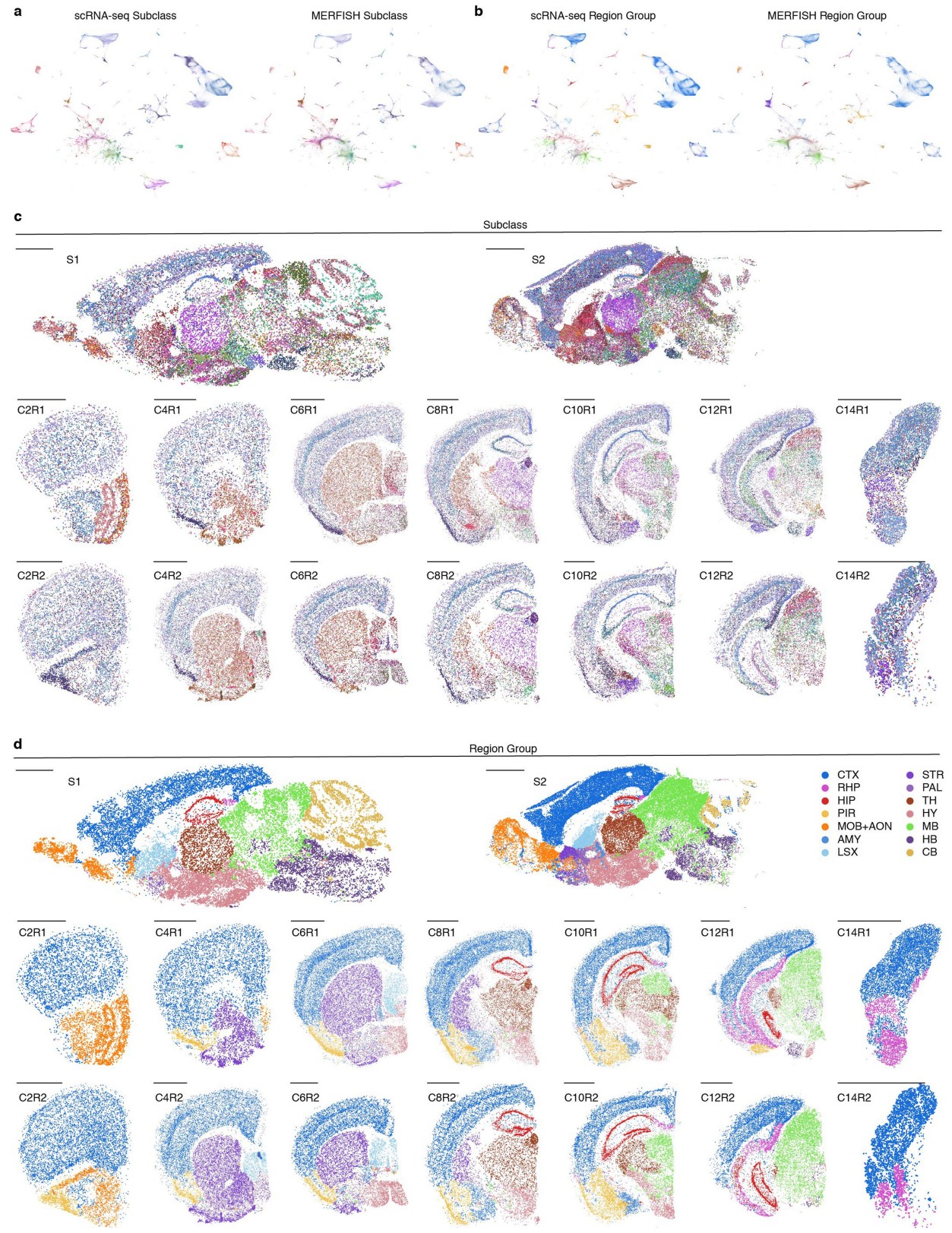

**Extended Data Fig. 6 | Joint clustering and annotation of MERFISH and scRNA-seq data. a**, **b**, Joint t-SNE of scRNA-seq neurons (n = 2,619,158, left) and MERFISH neurons (n = 329,282, right) colored by cell subclass (**a**) or region group (**b**). The labels for scRNA-seq cells are based on Yao et al.[18] and the labels for MERFISH cells are predicted through integration. **c**, **d**, The MERFISH slices colored by cell subclass (**c**) or region group (**d**). Scale bars represent 15 mm.

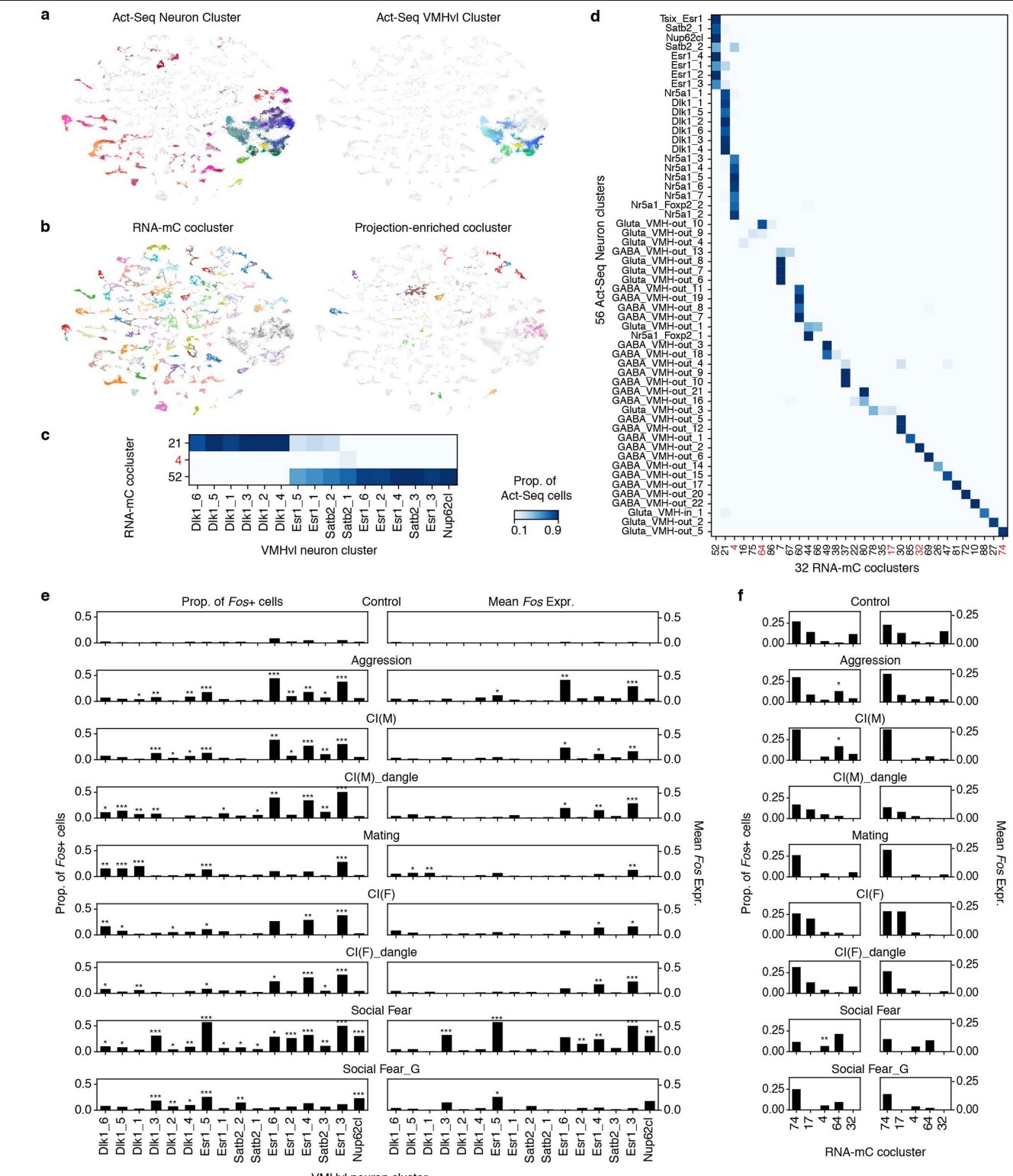

**Extended Data Fig. 7 | Integration and comparisons of hypothalamic Epi-Retro-Seq, Act-seq, and scRNAseq data. a**, **b**, Joint t-SNE of Act-Seq neurons[7] (n = 78,476, **a**) and scRNA-seq neurons (n = 148,840, **b**). In (**a**), all Act-Seq neurons (left) or only VMHvl neurons (right) are colored by Act-Seq neuron cluster (left) or VMH cluster (right). In (**b**), all scRNA-seq neurons (left) or only neurons in projection associated clusters (right) are colored by the co-cluster label. **c**, **d**, The proportion of neurons from each of the Act-Seq VMHvl clusters (**c**) or all neuron clusters (**d**) classified as neurons of each co-cluster. Only the co-clusters with value > 0.1 in at least one Act-Seq cluster are shown. The projection-associated co-clusters are labeled in red. **e**, **f**, Proportion of *Fos*+ "behavior activated" cells (left) or average *Fos* expression (right) of each VMHvl cluster (**e**) or each co-cluster (**f**) in control and different behavior experiments. Only the co-clusters labeled red in d are shown in (**f**). *, **, and *** represent FDR < 0.1, 0.01, and 0.001, respectively.

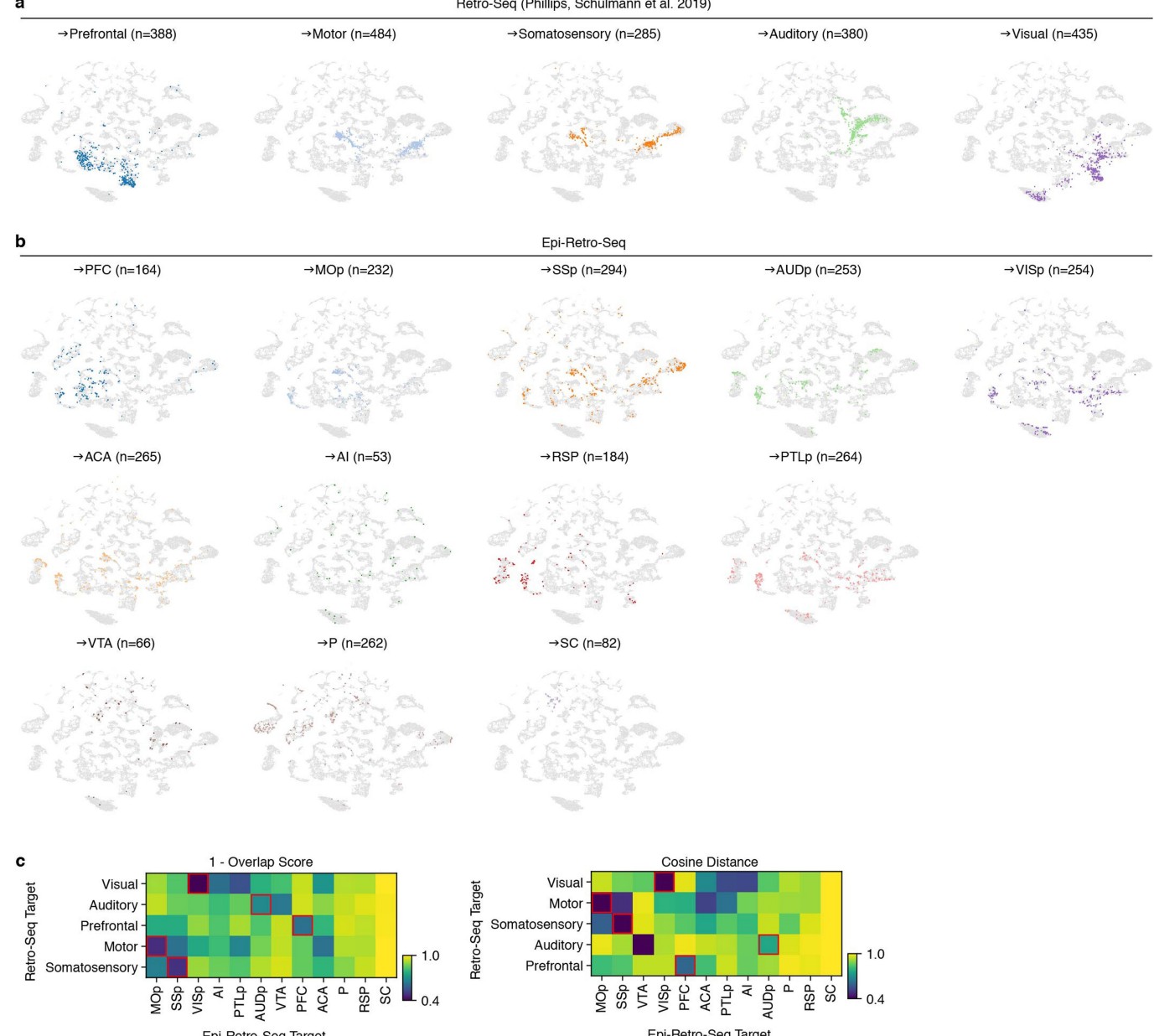

**Extended Data Fig. 8 | Comparison of thalamic Retro-Seq and Epi-Retro-Seq data. a**, t-SNEs for visualization of the Retro-Seq data from thalamic neurons projecting to prefrontal, motor, somatosensory, auditory, and visual cortices[8] that were mapped onto the joint-clustering analysis of Epi-Retro-Seq, unbiased snmC-seq and scRNA-seq in TH. **b**, The t-SNEs for visualization of the Epi-Retro-Seq data for thalamic neurons projecting to 12 different targets that were mapped onto the same t-SNE space. **c**, The overlap score and cosine distance were calculated for each pairwise comparison of Retro-Seq and Epi-Retro-Seq projections and were visualized in the heatmaps, respectively.

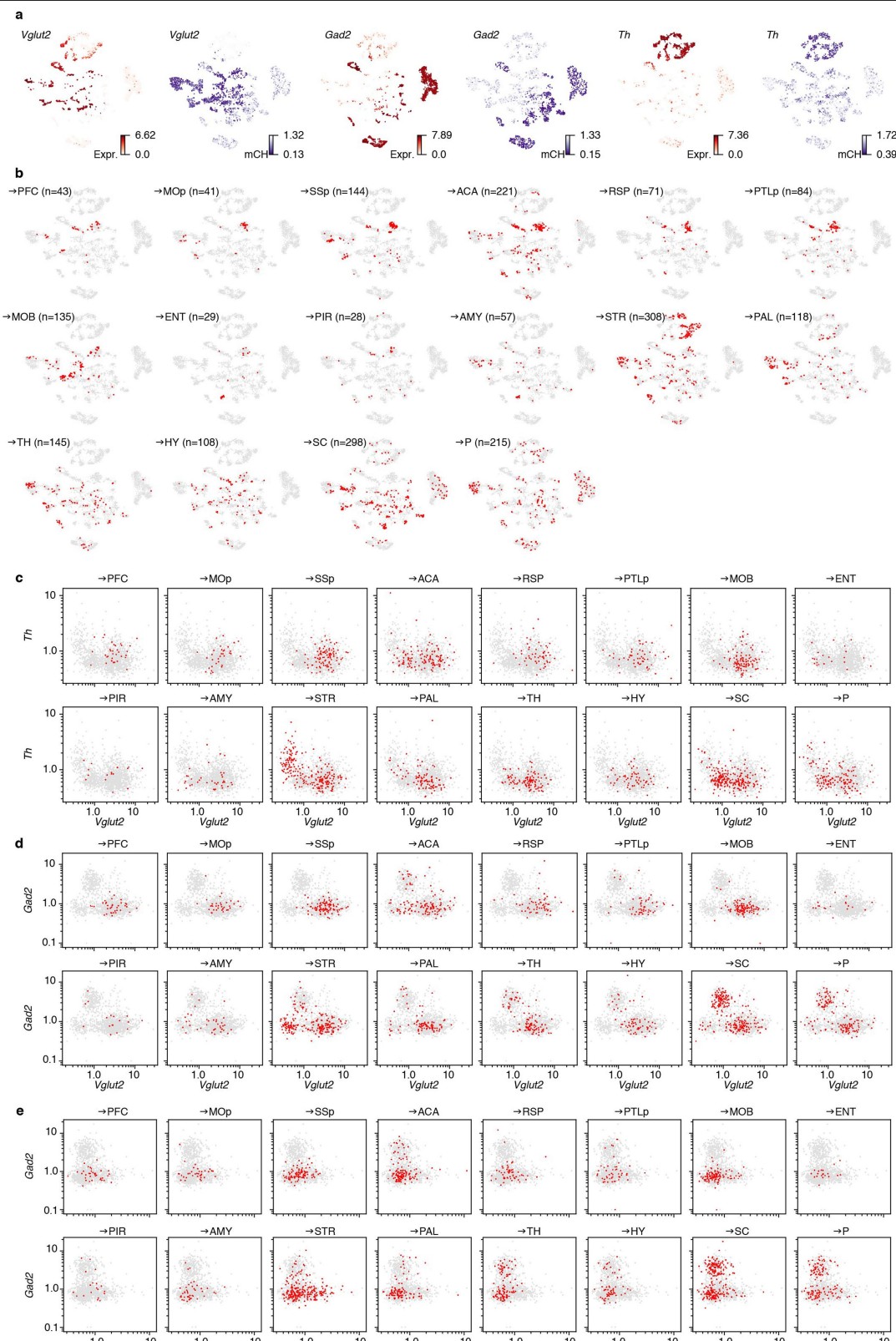

**Extended Data Fig. 9 | The neurotransmitter usage of VTA projection neurons. a**, Joint t-SNE of Epi-Retro-Seq, unbiased snmC-seq and scRNA-seq of VTA neurons colored by the gene expression levels (red) and gene-body mCH levels (purple) for *Vglut2* (left), *Gad2* (middle), and *Th* (right), marker genes for glutamatergic, GABAergic, and dopaminergic neurons, respectively. **b**, The distribution of VTA neurons projecting to each of the 16 targets on the same t-SNE. **c-e**, The gene-body mCH levels of *Th* versus *Vglut2* (**c**), *Gad2* versus *Vglut2* (**d**), and *Gad2* versus *Th* (**e**) for VTA neurons projecting to each of the 16 targets, are visualized in scatter plots. Note that, because low mCH levels indicate high gene expression, the axes in **c-e** are plotted as the reciprocal mCH values (1/gene body mCH), so low mCH is plotted to the right/up and high to the left/down.

# Reporting Summary

## Statistics

For all statistical analyses, confirm that the following items are present in the figure legend, table legend, main text, or Methods section.

| n/a | Confirmed | |
|---|---|---|
| ☐ | ☒ | The exact sample size (*n*) for each experimental group/condition, given as a discrete number and unit of measurement |
| ☐ | ☒ | A statement on whether measurements were taken from distinct samples or whether the same sample was measured repeatedly |
| ☐ | ☒ | The statistical test(s) used AND whether they are one- or two-sided *Only common tests should be described solely by name; describe more complex techniques in the Methods section.* |
| ☒ | ☐ | A description of all covariates tested |
| ☐ | ☒ | A description of any assumptions or corrections, such as tests of normality and adjustment for multiple comparisons |
| ☐ | ☒ | A full description of the statistical parameters including central tendency (e.g. means) or other basic estimates (e.g. regression coefficient) AND variation (e.g. standard deviation) or associated estimates of uncertainty (e.g. confidence intervals) |
| ☐ | ☒ | For null hypothesis testing, the test statistic (e.g. *F*, *t*, *r*) with confidence intervals, effect sizes, degrees of freedom and *P* value noted *Give P values as exact values whenever suitable.* |
| ☒ | ☐ | For Bayesian analysis, information on the choice of priors and Markov chain Monte Carlo settings |
| ☒ | ☐ | For hierarchical and complex designs, identification of the appropriate level for tests and full reporting of outcomes |
| ☐ | ☒ | Estimates of effect sizes (e.g. Cohen's *d*, Pearson's *r*), indicating how they were calculated |

*Our web collection on statistics for biologists contains articles on many of the points above.*

## Software and code

Policy information about availability of computer code

| Data collection | BD Influx Sortware v1.2.0.142 (flow cytometry), Freedom EVOware v2.7 (library preparation), Illumina MiSeq control software v3.1.0.13 and NovaSeq 6000 control software v1.6.0/RTA v3.4.4 (sequencing), Olympus cellSens Dimension 1.8 (image acquisition) |
|---|---|
| Data analysis | System: python=3.7.12, skypilot==1.0.0.dev0; Mapping: yap=1.5.11, cutadapt=2.10, bismark=0.20.0, bowtie2=2.3.5, picard=2.18, samtools=1.9; Analysis: allcools=1.0.17, anndata=0.8.0, scanpy=1.9.1, zarr=2.12.0, bedtools=2.30.0, scikit-learn=1.0.2, h5py=3.7.0, hdf5=1.12.2, htslib=1.16, matplotlib-base=3.5.2, harmonypy=0.0.9, qnorm=0.8.1, wmb=0.1.36; Other code are available on https://github.com/zhoujt1994/EpiRetroSeq2023.git |

For manuscripts utilizing custom algorithms or software that are central to the research but not yet described in published literature, software must be made available to editors and reviewers. We strongly encourage code deposition in a community repository (e.g. GitHub). See the Nature Portfolio guidelines for submitting code & software for further information.

## Data

Policy information about availability of data
All manuscripts must include a data availability statement. This statement should provide the following information, where applicable:
- Accession codes, unique identifiers, or web links for publicly available datasets
- A description of any restrictions on data availability
- For clinical datasets or third party data, please ensure that the statement adheres to our policy

Raw and processed data are also available at GEO under accession code GSE230782. Processed data can be explored on our web portal: http://neomorph.salk.edu/epiretro. Other datasets used in this study include scRNA-seq (https://portal.brain-map.org/atlases-and-data/bkp/abc-atlas), snmC-seq and MERFISH (https://mousebrain.salk.edu/download), Act-Seq (DOI 10.17632/ypx3sw2f7c.3), and Retro-Seq (GSE133912). The mm10 genome was downloaded from https://hgdownload.soe.ucsc.edu/goldenPath/mm10/bigZips/.

## Research involving human participants, their data, or biological material

Policy information about studies with human participants or human data. See also policy information about sex, gender (identity/presentation), and sexual orientation and race, ethnicity and racism.

| | |
|---|---|
| Reporting on sex and gender | N/A |
| Reporting on race, ethnicity, or other socially relevant groupings | N/A |
| Population characteristics | N/A |
| Recruitment | N/A |
| Ethics oversight | N/A |

Note that full information on the approval of the study protocol must also be provided in the manuscript.

# Field-specific reporting

Please select the one below that is the best fit for your research. If you are not sure, read the appropriate sections before making your selection.

☒ Life sciences  ☐ Behavioural & social sciences  ☐ Ecological, evolutionary & environmental sciences

For a reference copy of the document with all sections, see nature.com/documents/nr-reporting-summary-flat.pdf

# Life sciences study design

All studies must disclose on these points even when the disclosure is negative.

| | |
|---|---|
| Sample size | No sample size calculation was performed. Number of cells from each experiment after QC were reported in Supplementary Table 3. The sample size allowed us to obtain high coverage methylomes for each projection, and confidently identify projection neuron enriched clusters. |
| Data exclusions | We imaged the live tissue and closely inspected every injection site to ensure that the injection location was as intended and off-target injections were eliminated.<br>Poor quality nuclei were excluded from downstream analyses through four steps of quality controls (QCs) described in Methods. |
| Replication | At least 2 male and 2 female mice were injected with AAV-retro-Cre for each projection target. Male and female samples were pooled separately for nuclei preparation. Nuclei collected from the male and female pool were used as biological replicates in the downstream analyses. The consistency of projection enriched clusters between biological replicates are reported in Extended Data Fig. 5. The comparison of results from biological replicates and computational replicates are shown in Extended Data Fig. 3 and discussed in "Quantification of projection neuron difference with AUROC" in Methods. |
| Randomization | Animals used for injections into each brain area were selected at random. FACS sorted cells selected for sequencing were randomized during the sorting process. |
| Blinding | Technicians doing nuclei preps and snmC-seq analyses were blind to the injection sites used for each sample. |

# Reporting for specific materials, systems and methods

We require information from authors about some types of materials, experimental systems and methods used in many studies. Here, indicate whether each material, system or method listed is relevant to your study. If you are not sure if a list item applies to your research, read the appropriate section before selecting a response.

## Materials & experimental systems

| n/a | Involved in the study |
|---|---|
| ☐ | ☒ Antibodies |
| ☒ | ☐ Eukaryotic cell lines |
| ☒ | ☐ Palaeontology and archaeology |
| ☐ | ☒ Animals and other organisms |
| ☒ | ☐ Clinical data |
| ☒ | ☐ Dual use research of concern |
| ☒ | ☐ Plants |

## Methods

| n/a | Involved in the study |
|---|---|
| ☒ | ☐ ChIP-seq |
| ☐ | ☒ Flow cytometry |
| ☒ | ☐ MRI-based neuroimaging |

## Antibodies

| | |
|---|---|
| Antibodies used | anti-GFP antibody, dilution: 1:500, Alexa Fluor 488 (Invitrogen, A-21311)<br>anti-NeuN antibody, clone A60, dilution: 1:300,  EMD Millipore MAB377 conjugated with Alexa Fluor 647 (Invitrogen A20173) |
| Validation | Anti-NeuN antibodies have been previously published for use in immunohistochemistry  and  flow cytometry experiments (PMID: 23828890, 26087164). Anti-GFP antibody has been validated in Kim et al. Neuron 2020 (PMID: 32396852). |

## Animals and other research organisms

Policy information about studies involving animals; ARRIVE guidelines recommended for reporting animal research, and Sex and Gender in Research

| | |
|---|---|
| Laboratory animals | The knock-in mouse line, R26R-CAG-loxp-stop-loxp-Sun1-sfGFP-Myc (INTACT) used in Epi-Retro-Seq was maintained on a C57BL/6J background. Adult male and female INTACT mice were used for the retrograde labeling experiments. Animals were housed in an AAALAC accredited facility at the Salk Institute. Lighting was controlled on a 12 hour light/12 hour dark cycle. Temperature was monitored and adjusted in accordance with Guide for the Care and Use of Laboratory Animals. Humidity was not controlled but monitored. Because all air coming in is 100% fresh air (not re-circulated), humidity in the animal facilities is approximately the same as the outside ambient air. San Diego averages 40-60% humidity year-round. Animals were 35-54 days old at the time of surgery for viral vector injections, were sacrificed 13-17 days later, and were 50-70 days old on the day of dissection.<br>56-63 day old, C57BL/6J "wild-type" mice were used for MERFISH experiments. |
| Wild animals | N/A |
| Reporting on sex | The sample sizes for different sexes are reported in Supplementary Table 3. The consistency of projection enriched clusters between sexes are reported in Extended Data Fig. 5. Different sexes are used to split training and testing sets and the performances were compared with random split in Extended Data Fig. 3. |
| Field-collected samples | The study did not involve samples collected from the field. |
| Ethics oversight | All experimental procedures using live animals were approved by the Salk Institute Animal Care and Use Committee. |

Note that full information on the approval of the study protocol must also be provided in the manuscript.

## Plants

| | |
|---|---|
| Seed stocks | N/A |
| Novel plant genotypes | N/A |
| Authentication | N/A |

# Flow Cytometry

## Plots

Confirm that:

☒ The axis labels state the marker and fluorochrome used (e.g. CD4-FITC).

☒ The axis scales are clearly visible. Include numbers along axes only for bottom left plot of group (a 'group' is an analysis of identical markers).

☒ All plots are contour plots with outliers or pseudocolor plots.

☒ A numerical value for number of cells or percentage (with statistics) is provided.

## Methodology

**Sample preparation**

Manually dissected mouse brain samples were snap-frozen on dry ice and stored at -80 °C.  Prior to nuclei preparation, for each projection, samples from 2 males and 2 females were pooled separately as biological replicates. The frozen brain tissues were transferred to a pre-chiled 2-mL dounce homogenizer with 1 mL ice-cold NIM buffer (0.25M sucrose, 25mM KCl, 5mM MgCl2, 10mM Tris-HCl (pH7.4), 1mM DTT (Sigma 646563), 10μl of protease inhibitor (Sigma P8340)), with 0.1% Triton X-100 and 5μM Hoechst 33342 (Invitrogen H3570), and gently homogenized on ice with the pre-chilled pestle 10-15 times. The homogenate was transferred to pre-chilled microcentrifuge tubes and centrifuged at 1000 rcf for 8 min at 4 °C to pellet the nuclei. The pellet was resuspended in 1 mL ice-cold NIM buffer, and again centrifuged at 1000 rcf for 8 min at 4 °C. The pellet was then resuspended in 450 μL of ice-cold NSB buffer (0.25M sucrose, 5mM MgCl2, 10mM Tris-HCl (pH7.4), 1mM DTT, 9ul of Protease inhibitor), and filtered through 40μM cell strainer. The filtered nuclei suspension was incubated on ice for at least 30 minutes with 50μl of nuclease-free BSA for at least 10 minutes, then incubated with GFP antibody, Alexa Fluor 488 (Invitrogen, A-21311) and anti-NeuN antibody (EMD Millipore MAB377) conjugated with Alexa Fluor 647 (Invitrogen A20173). GFP+/NeuN+ single nuclei were isolated using fluorescence-activated nuclei sorting (FANS) on a BD Influx sorter with 100μm nozzle, and sorted into 384-well plates preloaded with 2μl of digestion buffer for snmC-seq215 (20mL digestion buffer consists of 10mL M-digestion buffer (2×, Zymo D5021-9), 1ml Proteinase K (20mg, Zymo D3001-2-20), 9mL water, and 10μL unmethylated lambda DNA (100pg/μL, Promega, D1521)). The collected plates were incubated at 50 °C for 20 minutes then stored at -20 °C.

**Instrument**

BD Influx

**Software**

BD Influx Sortware v1.2.0.142

**Cell population abundance**

We sorted NeuN-positive and GFP-positive nuclei.

**Gating strategy**

Intact nuclei were first discriminated from debris by virtue of their bright DNA labeling (Hoechst Height signal) followed by light scattering profiles (Forward Scatter (FSC) Height vs Side Scatter (SSC) Height). Events with high Pulse Width measurements for FSC and SSC were then excluded as aggregates. Next, NeuN-AlexaFluor 647 labelled neuronal nuclei were selected ("*670/30 640" Height) from which GFP positive nuclei were sorted ("*530/40 488" Height).

☒ Tick this box to confirm that a figure exemplifying the gating strategy is provided in the Supplementary Information.

