## [Peer Review File · Nature]

Manuscript Title: Brain-wide Correspondence Between Neuronal Epigenomics and Long-Distance Projections

Reviewer Comments & Author Rebuttals

Reviewer Reports on the Initial Version:

Referees' comments:

Referee #1:

Remarks to the Author:

Linking the genetic/epigenetic signature of a neuron to its projection pattern is an important goal in Neuroscience. Developing methods to achieve this are important and Epi-retro- RNASeq is one such method. Zhou and colleagues deploy this method to study cell-type specific projections to 24 distinct target regions of the murine CNS. They aim to expand on previous work, and address 1. whether neuronal projection is correlated with genetic signature across many brain regions, and 2. are there different principles in linking projection status to epigenetics for different brain areas. A substantial strength of this study is the integration of three datasets (epi-retroseq, scRNAseq, and snmseq) plus spatial transcriptomics to define molecular and anatomical correlations. While a plethora of valuable data and sophisticated analyses are presented, I am not yet persuaded that the main questions posed are satisfactorily addressed, as currently presented. Furthermore, the paper is written in a dense style, partly due to space restrictions I'm sure, but it makes for a laborious read. Below is a point by point critique.

Main points.

1. Epi retro seq was chosen because it allows "identification of gene regulatory elements and prediction of gene expression in the same neuron". But a more direct way to address the stated questions would be to use transcriptomic signatures, which they do only using adjacent datasets. Furthermore, the paper does not seem to go into much depth on the stated goal of identifying gene regulatory elements, and the validation of at least some exemplary findings on regulatory elements is lacking.
2. The total number of neurons analyzed from 225 target-source pairs is ~ 33,000, which seems low, considering the breadth of regions analyzed. This is likely due to the lower throughput of epi-retro-seq. Related, the location of the retro-AAV is crucial to the interpretation of the data, yet it was not validated or quantified. Furthermore, it would be informative to present the number of GFP+ nuclei recovered for each brain region, either as raw number and/or density per sorted cells. This data might inform the reader on the density of axon projections in each target area and help understand the variability observed in the dataset. Finally, the approach is not unbiased, as dissections of source regions are based on "locations with known projections to the target".
3. Fig 1d,f require better labels.
4. Fig 3 highlights the utility of the overall approach, showing the different hypothalamic cell clusters that project to distinct targets. The surprising contrast with Kim et al 2019 is appreciated.
5. Lateral Hypothalamus is known to send GABA and VGlut2+ projections to the VTA; only one of these appears robustly in the analysis? How can one account for this discrepancy?
6. Fig 3 and 4 maps CREs for various DE genes, and identifies putative TFs that bind these regulatory regions; however, no example(s) of functional validation is provided, a weakness, considering the stated goals of the paper.

7. The DA findings are not particularly novel, and some conclusions on dopamine neurons do not align with published literature. For instance, the authors state that neurons from the SNc or VTA projecting to the striatum do not express Vglut2 (671-673). It is likely this is due to sampling bias. Vglut2 is expressed in some VTA and SNc DA subsets that project to various parts of the striatum, now reproduced in several studies. Also, in Fig S9b, it does not seem like the Dopa cluster is represented in any of the cortical projection datasets.

8. For at least a few key regions presented, validation of at least one new biological finding obtained from the dataset, using orthogonal approaches (as an example, projection mapping in a Cre strain, or retrograde tracer + markers) , should be performed and shown to build confidence in the underlying datasets/processing thereof.

9. For each region/fig, the conclusions are not succinctly summarized. My suggestion is that for each Fig/brain region, the novel biological findings should be explicitly stated. Additionally, a supplementary table should be provided that includes the clusters from each figure and the cognate projections, how that compares to the literature (with citations). Where no literature is available, this could be highlighted as a novel finding.

10. For each region, the number of total clusters is substantially larger than the number of projection identified clusters (eg 17 out of 94 hypothalamic clusters). What might account for this?

11. The limitations of this epi retro seq approach/analyses also warrants a paragraph in the main text.

12. The summary provided is more a technical summary, which seems redundant; I would prefer a summary addressing the stated goals (points 1 and 2 listed above)

Minor points

1. Fig 3b., two clusters, 76 and 0, seem to have the same name. Please clarify. These seem to delineate paraSTN and STN in accordance with previous schemes (Wallen Mackenzie, Comm Biol, 2020).

2. In Fig 4b, it is unclear why some cell types have the same name. For instance, what distinguishes cluster 29 from 43.

3. Fig 3 panel h, i.....I don't see cluster 76.

4. The use and meaning of AUROC is poorly explained.

5. The sentence (lines 264-265) is not clear.

6. For the data provided on Fig 2f, significance should be mentioned in the legend..

7. It is very difficult for the reader to orient him/herself in the hypothalamus in figure 3e. The small schematics are helpful, but too small and too pale to be valuable.

8. The sentence (lines 401-404) is somewhat speculative.

9. Fig 3a and 3b will benefit from having the color code mentioned in the same figure.

10. The methods to define the genomic bins should be explained better (mapping and preprocessing section). Also, can the authors better explain

11. Please clarify panel 2f

Referee #2:

Remarks to the Author:

This manuscript describes the large-scale analysis and data collection performed from Zhou et al. using Epi-Retro-Seq to characterize DNA methylation patterns of neuronal projections within 24 different target mouse brain areas, which comprised of 32 processed source brain regions. The authors are commended for performing these in-depth analyses of mouse brain neuronal circuitry and validating their Epi-Retro-seq datasets with unbiased samples of snMC-seq, single cell RNA-Seq from 87 microdissected brain regions and spatial transcriptomics from MERFISH data. This manuscript aims to demonstrate the validity of these analyzed datasets to answer whether neurons from distinct source regions that project to similar brain targets have similar epigenomic and transcriptomic signatures. Moreover, they aim to determine how distinguishable neurons from the same source that project to the distinct target regions

are from one another using DNA methylation and transcriptomics.

The quality control approach appears robust and rigorous. We applaud the authors for leveraging the use of previously classified data to their advantage to make comparisons within published datasets throughout this manuscript to determine the validity of their own data. Often there are degrees of overlapping or shared patterns from comparing clustering between datasets, demonstrating their identified projection cell types from a particular source region are accurately representative of that projection neuron type. However, it is also noted throughout when these comparisons detect unshared clusters between two sequencing datasets. MERFISH data is also implemented to map the spatial organization of projection cell types, which is ultimately found to vary depending on the projection and source. The authors use the expression patterns of DEGs, DNA methylation pattern and identify transcription factors within motif enrichment in differentially methylation regions to determine how similar gene regulation is amongst distinct projection neurons within a particular brain region. Lastly, the authors examine the use of neurotransmitters in different projections by examining expression levels of 9 canonical neurotransmitter transporter genes in each projection enriched cluster. The authors find that neurotransmitter usage varies across projection enriched clusters, with more than one neurotransmitter transporter gene significantly expressed within a large proportion of projection enriched clusters. One example of these analyzed datasets was the hindbrain, where neurotransmitters within MY \diamond CBX and P \diamond CBX projection neurons were detected that were consistent with previously published datasets. New neurotransmitters were also revealed from these datasets with several clusters showing distinctive spatial distributions.

1) In the methods the authors state that "42-49 day old adult male and female INTACT mice were used for the retrograde labeling experiments." However, this age is considered the adolescent stage. Please clarify that and provide more specific details on the age that surgeries are performed and the age that tissue is collected. Further, please clarify the age used for "Adult C57BL/6J "wild-type" mice were used for MERFISH experiments."

2) Males and females are used in these datasets however no sex comparisons are made throughout this manuscript. Presumably this is due to using n= 2 per sex. It is understood that one subject alone produces a great amount of data through this single-cell approach. However, including only 2 replicates per sex may not adequately capture some of the key sources of variance including within-subject variance. Further, some of the brain regions examined are known to be sexually dimorphic. It is recommended that the authors address this by describing these limitations. The authors could also provide initial comparisons of differences in enriched projection clusters between sexes, with the understanding that future studies are needed to adequately capture sex differences.

3) It appears that only the total number of neurons used for Epi-Retro-Seq (n=35,743) and single- cell RNA-Seq (n=2,434,472). Please list each number of nuclei isolated from each source + target combination to provide context for the readers about the number of cells analyzed/projection in a Supplemental Methods table. Do the authors observe any consistency with the amount of neurons isolated from each projection to the number of target neurons known to project to particular sources in other published work?

4) When comparing the expression levels of neurotransmitters in each of the projection-enriched clusters, are there any common transcription factor motifs or other DEGs in enriched-projection clusters that are highly expressing the same transporter genes? In essence, are common TFs or DEGs in for instance all GABAergic cells that project to the VTA for example?

Author Rebuttals to Initial Comments:

RS2.1 Whole Brain Response to Reviews:

We thank the reviewers and editors for their very helpful and constructive suggestions for improving our

manuscript. We respond to the specific comments of each reviewer in the context of their text, below. We note here that in addition to responding to the reviewer suggestions we have generated a data browser that will allow users to access our entire data set and conduct analyses related to specific questions that might be of interest to them. Related to this we have added the following text to the manuscript: “In subsequent figures we focus on a subset of all possible analyses of our very large dataset in order to highlight the utility of the data and to provide examples of interest. In order to facilitate further analyses of the complete data set, we have developed a data browser that incorporates functions to allow each of the types of analyses that we highlight below to be conducted for any given source brain region and/or projection target that might be of particular interest (<http://neomorph.salk.edu/epiretro>).”

Referees' comments:

Referee #1 (Remarks to the Author):

Linking the genetic/epigenetic signature of a neuron to its projection pattern is an important goal in Neuroscience. Developing methods to achieve this are important and Epi-retro-Seq is one such method. Zhou and colleagues deploy this method to study cell-type specific projections to 24 distinct target regions of the murine CNS. They aim to expand on previous work, and address 1. whether neuronal projection is correlated with genetic signature across many brain regions, and 2. are there different principles in linking projection status to epigenetics for different brain areas. A substantial strength of this study is the integration of three datasets (epi-retroseq, scRNAseq, and snmseq) plus spatial transcriptomics to define molecular and anatomical correlations. While a plethora of valuable data and sophisticated analyses are presented, I am not yet persuaded that the main questions posed are satisfactorily addressed, as currently presented. Furthermore, the paper is written in a dense style, partly due to space restrictions I'm sure, but it makes for a laborious read. Below is a point by point critique.

We thank the reviewer for recognizing the importance and potential impact of this work. We hope that the revisions that we have made in response to more specific suggestions below help to make the main points of the manuscript more clear. We hope that it can be appreciated that a study that addresses details across the entire brain is difficult to condense into a small number of take-home messages. Instead we have attempted to illustrate how these data can be used and to highlight a few brain areas of interest.

We believe that we have addressed both of the main questions listed above, although, because the relationships between projection targets and gene expression are complex, the answers to the questions are not as clear cut as they might have been. We hope that our revisions have made this more clear. With respect to “whether neuronal projection is correlated with genetic signature across many brain regions”, we have quantified the relationships between projection targets and molecularly defined cell clusters for every one of the 225 target source combinations and provided those measures in Extended Data Fig. 5. We also used AUROC to quantify the distinguishability between projection types based on epigenomic signatures, which emphasized the link between molecular type and projections. These results are provided in Extended Data Fig. 3 and we now also added Supplementary Table 4 with the numbers. The results presenting those relationships

across the entire brain also provide clear insight into the extent to which “there are different principles linking projection status to epigenetics for different brain areas.” Moreover, we specifically addressed this question in Figure 2, providing a quantitative measure for the selected example of projections to cortical versus subcortical targets for different brain areas. We have added a more extensive Summary section at the end of the manuscript that discusses these findings within the broader context of the whole manuscript.

Note that because of the complexity of these relationships, we have worded the questions stated at the beginning of the Results in a way that avoids suggesting that there will be a simple answer. “Overarching questions that can be addressed by this large data set include: how distinct are neurons from a given source that project to different targets? And are neurons in different sources that project to the same target combinations more or less distinguishable?” We believe that the new Summary section provides a better synthesis of these findings.

Main points.

1. Epi retro seq was chosen because it allows “identification of gene regulatory elements and prediction of gene expression in the same neuron”. But a more direct way to address the stated questions would be to use transcriptomic signatures, which they do only using adjacent datasets. Furthermore, the paper does not seem to go into much depth on the stated goal of identifying gene regulatory elements, and the validation of at least some exemplary findings on regulatory elements is lacking.

We appreciate the reviewer’s inquiry into the “stated questions”, which we interpret to refer to the “identification of gene regulatory elements and prediction of gene expression in the same neuron”. While transcriptomic analyses, such as snRNA-seq, indeed yield valuable insights into gene expression patterns; they do not provide direct information on gene regulatory elements. Specifically, transcriptomics primarily illuminates the transcriptional landscape of gene exons, which represents a minor fraction of the genome (less than 2%). It does not provide information on the epigenetic states of the genes or the expansive intronic regions encompassing regulatory DNA elements that regulate gene expression in neurons. Conversely, the utility of single neuron methylation sequencing to identify gene regulatory elements has been well documented in previous studies. We anticipate that the data we present here would align with the established utility observed in prior research, sustaining a consistent level of reliability and insight.

We also interpret the reviewer’s comment about validation of regulatory elements as a reference to the possibility that cell type specific enhancers might be predicted. It is important to note that, while such epigenomics data are extremely valuable, no method has been developed that allows unambiguous prediction about the ability of any putative enhancer to drive cell-type-selective gene expression. Instead it is expected that better algorithms for using these data will be developed in the future. For example, 2 publications from the Allen Institute group (Graybuck et al. 2021; Mich et al. 2021) demonstrate both the utility of such data and difficulties in making strong predictions. Some predicted enhancers are specific but many others are not. In response to this comment we compared our methylation data from L5-ET versus L5-IT cells to the enhancers that Graybuck et al. tested for targeting these cell types. We found that the enhancer regions that they predicted might target L5-ET cells generally lacked methylation in our L5-ET cells, but these enhancers did not always drive selective expression.

If the “stated questions” that the reviewer refers to is the relationship between neuronal projections and transcriptomic clusters, we would like to point out that there is a direct inverse relationship between gene body methylation and gene expression. While the relationship is not quite as direct as with RNAseq data it is practically as good and with the added advantage of obtaining epigenetic information.

Although it's challenging to finemap the enhancers that specifically drive gene expression in given cell types. We have added analyses of gene regulatory networks (GRN) combining information between cis and trans regulators based on their correlation across cell clusters. These correlations can potentially pinpoint gene regulation events with higher confidence. Our results show that the analysis captures known links between TF and target genes as well as new regulators of neuronal genes in specific cell types. “We then constructed a GRN in thalamus, which consists of 10.9M TF-DMR-Target triplet combinations, involving 469 TFs, 375k DMRs, and 13.3k target genes. These networks captured regulatory relationships reported in previous studies. For example, *Rora* has been identified as an essential factor for thalamocortical axon branching (Vitalis et al. 2018), and transcriptome analysis suggested that *Sema7a*, another essential regulator of thalamic cortical circuit maturation (Carcea et al. 2014), could be a potential target of *Rora*. In our data, *Rora* motifs are enriched in many clusters that are enriched for neurons projecting to cortical targets. Similar expression patterns were observed for *Rora* and *Sema7a*, as both of them are also highly expressed in the cortical projection enriched clusters. 43 DMRs that potentially mediate this regulation were identified at the flanking region of *Sema7a* (TSS±1Mb). Those DMRs contain the RORA motif and their mCG levels are anticorrelated with the expression of *Rora* and *Sema7a* (Fig. 4k). Additionally, our study also suggests new regulatory relationships in thalamus. *Pou4f1* has its binding motif enriched in DMRs hypo-methylated in clusters 25 and 45 that make projections to Pons. The network suggests that prepatter TFs *Irx1/2* (Bosse et al. 1997), are candidate downstream targets of *Pou4f1*, which is also specifically expressed in the same two clusters (Fig. 4l).”

2. The total number of neurons analyzed from 225 target-source pairs is ~ 33,000, which seems low, considering the breadth of regions analyzed. This is likely due to the lower throughput of epi-retro-seq.

Related, the location of the retro-AAV is crucial to the interpretation of the data, yet it was not validated or quantified.

Furthermore, it would be informative to present the number of GFP+ nuclei recovered for each brain region, either as raw number and/or density per sorted cells. This data might inform the reader on the density of axon projections in each target area and help understand the variability observed in the dataset.

Finally, the approach is not unbiased, as dissections of source regions are based on “locations with known projections to the target”.

These comments touch on numerous issues which we will attempt to organize according to the decisions that were made at the outset of the project and which informed our experimental design. These decisions involved a number of trade-offs with respect to reliability, repeatability, throughput, and data modalities. It should be appreciated that there are important technical limitations that prevent experiments combining retrograde tracing with single-cell omics from generating data that simultaneously addresses all of the questions that one might want to answer. Our experiments also build on a very large, decades-old prior literature that provides quantitative information about projection cell types across the mouse brain. We therefore did not intend for our dataset to replace

that knowledge, but rather to provide additional insight that can only be provided by single-cell omics analyses. Finally, we designed our experiments to be complementary to other ongoing experiments that were simultaneously being conducted as part of the BICCN network and the analysis which we have conducted and present here illustrate the value of maintaining consistent methodology across the different data generation platforms that our group employed. We illustrate this for integration of our projection-labeled neurons with unbiased data and with MERFISH data throughout the manuscript.

Regarding the validation of injection locations, the practical considerations relate to the overarching choice to maintain a consistent brain dissection scheme across all experiments in our group. In all experiments it is necessary to dissect fresh brain tissue in order to assign assayed single neurons to an approximate brain location. We had the same one person dissect as consistently as possible each of the hundreds of mouse brains that were used. Because every possible target region comes from a slice that is then dissected and contains other brain regions that connect to the target region, it is not possible to conduct high-resolution post hoc analysis of injection sites in fixed tissue. Instead we imaged the live tissue and closely inspected every injection site to ensure that the injection location was as intended and off-target injections were eliminated. We have added new text to the Methods section to describe this procedure. Further, our injection locations were chosen to target the centers of relatively large regions that are well isolated from surrounding regions. We did not attempt for example, to target injections to subregions of the thalamus or hypothalamus.

Note that inclusion of the numbers of sorted GFP+ nuclei would not (and was not intended to) provide information about the density of axon projections. There are many reasons that retrograde labeling followed by cellular dissociation and FACS (as was required for our workflow) would generate numbers that would not reliably indicate projection strength. First, projection targets vary widely in size relative to the size of AAV-retro injections. As a result the expected completeness of labeling varies between different targets. Second, in order to optimize the quality of epigenetic data, the FACS parameters used for various source/target combinations resulted in collection of different proportions of labeled nuclei. For example, when very few neurons were present we increased the flow rates so that we could sample from a larger overall volume and therefore collect enough cells to have a meaningful sample size. Thus, for projections with more labeled cells we must then choose between using the same flow rate which results in many cells passing through undetected while cells are being sorted into wells in the plate, or slowing down the flow rate, which results in much of the volume not being sorted within the time that the sample remains viable. For these reasons, the numbers of FACS sorted cells cannot be used as an indication of projection strength.

The purpose of our experiments was not to reveal the patterns of connections across the whole brain, but rather to link methylation status to projection targets for known connections. We therefore only dissected and analyzed neurons in brain regions known to project to each injected target region. While these samples might be considered “biased” in the sense that they will not allow for discovery of unknown projection cells, they do allow for statistically meaningful and quantitative comparisons between the methylation status of neurons from a particular source projecting to different targets. Furthermore, these enriched (“biased”) samples can then be compared to unbiased samples (companion papers and as illustrated in figures) if quantitative estimates of cell type frequency are desired. The target regions were selected based on both practical considerations and on expected scientific value.

The above comments are related to a later comment from this reviewer (#11 below) asking for discussion of limitations of the approach in the main text. With respect to manuscript revisions, we therefore address these issues in that context. We have added text to the Summary section discussing limitations imposed by our experimental design and the high-throughput nature of these brain-wide studies.

3. Fig 1d,f require better labels.

Figure 1d and f are intended to be an icon to illustrate the analyses we can perform with our dataset, not to illustrate specific results. Therefore, we only put the title and brief labels for each panel, rather than the full legends and axes tick labels.

4. Fig 3 highlights the utility of the overall approach, showing the different hypothalamic cell clusters that project to distinct targets. The surprising contrast with Kim et al 2019 is appreciated.

We thank the reviewer for this comment.

5. Lateral Hypothalamus is known to send GABA and VGlut2+ projections to the VTA; only one of these appears robustly in the analysis? How can one account for this discrepancy?

We appreciate the reviewer's comment on the GABA and VGlut2+ projections from the hypothalamus (HY) to the VTA, which we interpret as referring to the cell clusters enriched for the HY→VTA projection. We want to emphasize that the color in the heatmap represents the relative proportion of neurons in the cluster compared with the other projections. So a blue color does not mean that there are no neurons in the cluster, but the proportion of neurons is lower compared with the other projections. Additionally, the statistical significance is also related to the sample size, where a higher proportion but smaller cell number could represent an enrichment with lower confidence. In the HY→VTA example, Figure 3 shows a strong red color for this projection in only one cluster that is VGlut-expressing. We do identify cells in clusters that express GABA, but they are just not significantly enriched compared to the other projections (see Figure below). Related to this, the number of sampled hypothalamic cells projecting to VTA is small (n=35), reducing statistical power. Note, however, that the number of AMY-projecting cells is similar and we are able to detect more enriched clusters. We have added text to the manuscript to point out that the enrichment analysis is able to reliably identify existing projections but can be subject to false negatives. "It should be noted that in addition to clusters that are identified as being enriched in projection neurons, there are invariably neurons in other clusters without statistically significant enrichment. The absence of statistically significant enrichment should not be interpreted as an absence of projections from neurons belonging to a particular cluster."

6. Fig 3 and 4 maps CREs for various DE genes, and identifies putative TFs that bind these regulatory regions; however, no example(s) of functional validation is provided, a weakness, considering the stated goals of the paper.

Please see response to point #1 above, about identification of cell type specific enhancers. We also decided to make it more clear that we are referring to “candidate” CREs that have not been tested or functionally validated. We have therefore replaced “CREs” with candidate CREs (cCREs).

7. The DA findings are not particularly novel, and some conclusions on dopamine neurons do not align with published literature. For instance, the authors state that neurons from the SNc or VTA projecting to the striatum do not express Vglut2 (671-673). It is likely this is due to sampling bias. Vglut2 is expressed in some VTA and SNc DA subsets that project to various parts of the striatum, now reproduced in several studies. Also, in Fig S9b, it does not seem like the Dopa cluster is represented in any of the cortical projection datasets.

We thank the reviewer for pointing this out. While the data are in alignment with the published literature, our description of the results in the text did not accurately reflect the data. We have revised the text to correct this issue. The striatum projecting cells are highly enriched in a Dopa cluster that does not express Vglut2. However, there are other striatum projecting cells expressing both Vglut2 and Dopa (See figure below, where the boxes represent Vglut2+ Dopa+ cells). There is no specific threshold on DNA methylation to determine whether the gene is expressed or not, so the bottom of the boxes were determined roughly by comparing with the cortical projections. We modified the text to be more clear that it is not an absence of Vglut2 in the STR projecting cells. “In contrast to VTA→CTX neurons, VTA→STR neurons include a prominent population that is Th+ Vglut2- and there is a smaller proportion of neurons that are both Th+ and Vglut2+ (Fig. 5i). The most prominent populations of VTA→STR neurons compose two groups: Th+ Vglut2- and Th- Vglut2+ (Fig. 5i). (Note that the use of the “-” designation here indicates relatively low expression rather than a complete absence.)”

Again in Extended Data Fig. 9, although the largest Dopa cluster only showed up in the STR projection dataset, there are smaller clusters with both Vglut2 and Dopa that project to the cortex (See figure below, indicated by the arrows). The RNA data did not extensively sampled VTA, so those clusters look smaller in the integration space when plotting only the RNA data.

- For at least a few key regions presented, validation of at least one new biological finding obtained from the dataset, using orthogonal approaches (as an example, projection mapping in a Cre strain, or retrograde tracer + markers) , should be performed and shown to build confidence in the underlying datasets/processing thereof.

While we appreciate that incorporating further experiments could add to the novelty of the descriptions in this manuscript, we do not feel that this is necessary to build confidence. Instead, post-hoc comparisons to known relationships between projection targets and gene expression should be just as valuable. We show such comparisons for both cortical neurons projecting to ET vs. IT targets and for thalamic neurons. Furthermore, reviewer #2 explicitly points this out as a strength of our study. "One example of these analyzed datasets was the hindbrain, where neurotransmitters within MY→CBX and P→CBX projection neurons were detected that were

consistent with previously published datasets. New neurotransmitters were also revealed from these datasets with several clusters showing distinctive spatial distributions.”

9. For each region/fig, the conclusions are not succinctly summarized. My suggestion is that for each Fig/brain region, the novel biological findings should be explicitly stated. Additionally, a supplementary table should be provided that includes the clusters from each figure and the cognate projections, how that compares to the literature (with citations). Where no literature is available, this could be highlighted as a novel finding.

We have added text within each section to better contextualize and/or summarize the findings. We also added Supplementary Table 7 to summarize the cluster projection correspondence in hypothalamus and thalamus. We have now provided access to a data browser (see above) that allows visualization of any related data such as MERFISH labeling. Note that existing literature discusses cell types mostly with respect to neurotransmitter expression. Our study provides a map of projection neurons related to a much larger number of marker genes and cCREs. Thus, these different data sets and modalities are not readily subject to direct comparisons.

10. For each region, the number of total clusters is substantially larger than the number of projection identified clusters (eg 17 out of 94 hypothalamic clusters). What might account for this?

This result is expected for two main reasons. First, nearly every region contains cell types that only connect locally within the dissected region so would never be labeled retrogradely from a distant injection. Second, We have not made retrograde injections into every structure that receives input from each region, so it is expected that cell types projecting to distant targets that we have not sampled will not be included. (This is apparent in the comparison between the data in this manuscript and Kim et al., 2019, which sampled different HY projection targets.) Finally, the numbers of clusters indicated in some types of analyses (such as 17 of 94 from hypothalamus) include only those in which there was significant enrichment. Therefore, there are additional clusters that had some label but our sample size was not large enough to definitively identify those as being enriched for a particular projection.

We have added the following text to the manuscript: “It should be noted that in addition to clusters that are identified as being enriched in projection neurons, there are invariably neurons in other clusters without statistically significant enrichment. The absence of statistically significant enrichment should not be interpreted as an absence of projections from neurons belonging to a particular cluster.”

11. The limitations of this epi retro seq approach/analyses also warrants a paragraph in the main text.

We have added the following text to the paragraph in the discussion to include limitations that influence experimental design.

“It is important to note that our experiments were designed to assess the methylation status of neurons projecting to relatively large targets that could be reliably injected and assessed for accuracy during dissections of fresh tissue, and from a large number of source regions that could be readily and reliably dissected. Integration with MERFISH data allowed for more precise

anatomical localization of enriched clusters from these sources, but more focused studies using smaller retrograde tracer injections linked to smaller injection locations would be needed to identify possible differences between projection neurons at a finer resolution.”

Due to space restrictions, we discuss further detailed limitations of our analyses in Methods, when describing the corresponding approaches, including the paragraphs in “Quality control”, “Quantification of projection neuron difference with AUROC”, “Cluster associated with projection”, “Classification of MERFISH cells into major brain regions and cell clusters”, “Comparison with Act-Seq”, and “Gene regulatory network (GRN) analysis”.

12. The summary provided is more a technical summary, which seems redundant; I would prefer a summary addressing the stated goals (points 1 and 2 listed above)

A final paragraph has been added that is a less technical summary and focuses on the scientific findings about relationships between gene methylation and projection targets.

Minor points

1. Fig 3b., two clusters, 76 and 0, seem to have the same name. Please clarify. These seem to delineate paraSTN and STN in accordance with previous schemes (Wallen Mackenzie, Comm Biol, 2020).

We have added the following text to the manuscript in order to clarify this issue: “We annotated each co-cluster based on the 299 neuronal cell subclasses identified in scRNA-seq across the whole brain (Yao et al. 2023) (Methods). Note that this annotation does not give a unique name to each of the finest cluster divisions. Therefore clusters noted by different identifying cluster numbers (e.g. 0 to 94) may share the same cluster name (e.g. clusters 0 and 76 are both annotated as STN-PSTN Pitx2 Glut).” See further below.

2. In Fig 4b, it is unclear why some cell types have the same name. For instance, what distinguishes cluster 29 from 43.

We thank the reviewer for the comment and apologize for the confusion. The cluster name was assigned based on the annotation in Yao et al 2023, where our collaborators from the Allen Institute for Brain Science used their single-cell RNA-seq data together with MERFISH data to annotate 5204 cell clusters into 306 cell subclasses, each of which is given a descriptive name according to the spatial location and gene expression. In our analyses, to give a more detailed map of projection neuron enriched clusters, we clustered the cells from each major brain structure using a cluster resolution that is finer than the cell subclasses. The clusters we defined also have differentially expressed/methylated genes to support their separation, and could also locate differentially in the brain as shown in the associated MERFISH data. However, to build the link with our companion papers, we used the same naming as in Yao et al 2023, and it is expected that each cell subclass is split into more than one cluster in our study. As the reviewer notes here, readers can obtain more detail about clusters than just their annotated name by reading our figures and by using our data browser. We have also described the strategies for assigning names to our clusters in the method section. “Each of the Leiden coclusters was annotated using the name of one of the 299 neuronal

subclasses identified in our companion paper. Specifically, we used the subclass which the largest proportion of RNA cells in the cocluster are annotated as.”

3. Fig 3 panel h, i.....I don't see cluster 76.

This is because cluster 76 is a rare cluster with <30 cells in the unbiased methylation dataset. Therefore we do not have enough power to confidently identify DMRs in the cluster. Because of this, cluster 76 was used only in the DEG and DMG analyses but not used in DMR and TF analyses. We have edited the figure legend to clarify this.

4. The use and meaning of AUROC is poorly explained.

We have added the following sentence (red text) to describe the AUROC.

“To provide a resource that can be used to address the distinguishability of neurons with different projection targets, we quantified which projection types are epigenetically more different than the others by computing area under the curve of receiver operating characteristic (AUROC) for each of the target pairs from every source region (926 pairwise comparisons in total; **Fig. 1d and Extended Data Fig. 3**). Specifically, a linear model was trained to distinguish single neurons projecting to two different targets based on their DNA methylation profile, and the cross-validation AUROC was used to quantify the performance of the model, which in turn reflected the differences between neurons projecting to a pair of targets (**Methods**).”

5. The sentence (lines 264-265) is not clear.

We apologize for the confusion and decided to remove the sentence.

~~“The ET vs. IT differences described above group together various more specific targets, which are nevertheless distinct structures.”~~

6. For the data provided on Fig 2f, significance should be mentioned in the legend.

Due to the large number of meaningful comparisons (n=34), we have now added the P-values in Supplementary Table 6.

7. It is very difficult for the reader to orient him/herself in the hypothalamus in figure 3e. The small schematics are helpful, but too small and too pale to be valuable.

We have reorganized Figure 3 and 4 to make the schematics larger.

8. The sentence (lines 401-404) is somewhat speculative.

We agree that the statement is somewhat speculative. That we are speculating should be clear from the wording of the statement. Since the reviewer has also asked us to add more discussion of the implications of our findings in other parts of the manuscript we have kept this statement. For reference the statement reads: “Our observations across the full spatial extent of hypothalamus and a large number of projection targets reveal strong correlations between clusters and projection targets, suggesting that cell types defined by their projections and genetics/epigenetics are also likely to make distinct contributions to hypothalamic function and related behaviors.”

9. Fig 3a and 3b will benefit from having the color code mentioned in the same figure.

There is no relationship between the color codes used in Fig 3a versus 3b. And in Figure 3a the colors are simply assigned randomly to each cluster so there is not any particular code. In the legend, we stated the color codes in Figure 3b-e are shared. While in figure 3a, there are 76 clusters so many of the colors are reused and randomly assigned.

10. The methods to define the genomic bins should be explained better (mapping and preprocessing section). Also, can the authors better explain

We have added the following sentence to describe the binning of the genome.

“The whole genome was parsed into 100kb non-overlapping genomic bins (chr1:0-100,000; chr1:100,000-200,000; etc.) using bedtools make-window and”

11. Please clarify panel 2f

Because we provided considerable detail about this figure in the original text and legend, we are having difficulty determining what needs to be clarified. We have added some more detail in both the text and the figure legend that we hope does clarify any confusion. But we would welcome further comments that might allow us to better understand any sources of confusion.

Referee #2 (Remarks to the Author):

This manuscript describes the large-scale analysis and data collection performed from Zhou et al. using Epi-Retro-Seq to characterize DNA methylation patterns of neuronal projections within 24 different target mouse brain areas, which comprised of 32 processed source brain regions. The authors are commended for performing these in-depth analyses of mouse brain neuronal circuitry and validating their Epi-Retro-seq datasets with unbiased samples of snMC-seq, single cell RNA-Seq from 87 microdissected brain regions and spatial transcriptomics from MERFISH data. This manuscript aims to demonstrate the validity of these analyzed datasets to answer whether neurons from distinct source regions that project to similar brain targets have similar epigenomic and transcriptomic signatures. Moreover, they aim to determine how distinguishable neurons from the same source that project to the distinct target regions are from one another using DNA methylation and transcriptomics.

The quality control approach appears robust and rigorous. We applaud the authors for leveraging the use of previously classified data to their advantage to make comparisons within published datasets throughout this manuscript to determine the validity of their own data. Often there are degrees of overlapping or shared patterns from comparing clustering between datasets, demonstrating their identified projection cell types from a particular source region are accurately representative of that projection neuron type. However, it is also noted throughout when these comparisons detect unshared clusters between two sequencing datasets. MERFISH data is also implemented to map the spatial organization of projection cell types, which is ultimately found to vary depending on the projection and source. The authors use the expression patterns of DEGS, DNA methylation pattern and identify transcription factors within motif enrichment in differentially methylation regions to determine how similar gene regulation is amongst distinct projection neurons

within a particular brain region. Lastly, the authors examine the use of neurotransmitters in different projections by examining expression levels of 9 canonical neurotransmitter transporter genes in each projection enriched cluster. The authors find that neurotransmitter usage varies across projection enriched clusters, with more than one neurotransmitter transporter gene significantly expressed within a large proportion of projection enriched clusters. One example of these analyzed datasets was the hindbrain, where neurotransmitters within MY \diamond CBX and P \diamond CBX projection neurons were detected that were consistent with previously published datasets. New neurotransmitters were also revealed from these datasets with several clusters showing distinctive spatial distributions.

We thank the reviewer for this excellent summary.

- 1) In the methods the authors state that “42-49 day old adult male and female INTACT mice were used for the retrograde labeling experiments.” However, this age is considered the adolescent stage. Please clarify that and provide more specific details on the age that surgeries are performed and the age that tissue is collected. Further, please clarify the age used for “Adult C57BL/6J “wild-type” mice were used for MERFISH experiments.”

We have added/modified text in the methods section which now reads:

“Animals were 35-54 days old at the time of surgery for viral vector injections, were sacrificed 13-17 days later, and were 50-70 days old on the day of dissection. 56-63 day old, C57BL/6J “wild-type” mice were used for MERFISH experiments.”

- 2) Males and females are used in these datasets however no sex comparisons are made throughout this manuscript. Presumably this is due to using n= 2 per sex. It is understood that one subject alone produces a great amount of data through this single-cell approach. However, including only 2 replicates per sex may not adequately capture some of the key sources of variance including within-subject variance. Further, some of the brain regions examined are known to be sexually dimorphic. It is recommended that the authors address this by describing these limitations. The authors could also provide initial comparisons of differences in enriched projection clusters between sexes, with the understanding that future studies are needed to adequately capture sex differences.

We have modified Extended Data Fig. 5 to provide information on whether the distribution of projection cells across clusters are consistent between sexes or not. The revised figure also now also highlights statistical significance of all comparisons. This can be referenced along with Supplementary Table 3 (see below) to provide insight into the relationships between sample sizes and statistical significance. We also added to the Method section the potential limitation of small sample size as the reviewer points out here.

- 3) It appears that only the total number of neurons used for Epi-Retro-Seq (n=35,743) and single-cell RNA-Seq (n=2,434,472). Please list each number of nuclei isolated from each source + target combination to provide context for the readers about the number of cells analyzed/projection in a Supplemental Methods table. Do the authors observe any consistency with the amount of neurons isolated from each projection to the number of target neurons known to project to particular sources in other published work?

We thank the reviewer for this suggestion. We have now added this information as Supplementary Table 3. There are two reasons that these numbers could be of potential interest. First, knowing the sample size can help the reader to understand the statistical power that was available for various analyses. The second reason suggested by the reviewer is that these numbers might provide insight into the relative strength of projection from a source to different targets. We would like to point out that the numbers provided should not be used for that purpose. There are many reasons that retrograde labeling followed by cellular dissociation and FACS (as was required for our workflow) would generate numbers that would not reliably indicate projection strength. First, projection targets vary widely in size relative to the size of AAV-retro injections. As a result the expected completeness of labeling varies between different targets. Second, in order to optimize the quality of epigenetic data, the FACS parameters used for various source/target combinations resulted in collection of different proportions of labeled nuclei. For example, when very few neurons were present we increased the flow rates so that we could sample from a larger overall volume and therefore collect enough cells to have a meaningful sample size. Thus, for projections with more labeled cells we must then choose between using the same flow rate which results in many cells passing through undetected while cells are being sorted into wells in the plate, or slowing down the flow rate, which results in much of the volume not being sorted within the time that the sample remains viable. For these reasons, the numbers of FACS sorted cells cannot be used as an indication of projection strength.

This is an interesting point to look at the reciprocal connections between brain regions. However, in our study, we collected a fixed number of cells for each projection, so we are not able to accurately quantify the proportion of neurons projecting from different targets. This approach helped us to enrich for rare projections and enabled us to study the molecular features of them, while at the cost of losing information about the proportions of projection cells in each region.

- 4) When comparing the expression levels of neurotransmitters in each of the projection-enriched clusters, are there any common transcription factor motifs or other DEGs in enriched-projection clusters that are highly expressing the same transporter genes? In essence, are common TFs or DEGs in for instance all GABAergic cells that project to the VTA for example?

If we are interpreting the reviewers' question correctly, this can be answered by referring to the cluster names in Figure 4b and then looking at the plots of the DEGs, DMRs and enriched TFs for those named clusters in Figures 4e-4h. Figure 4b shows that the GABAergic clusters with enriched projections are 29 and 43 → SC, 7 and 11 → P, and 20 → VTA. The data for these clusters can be compared in Figures 4e-h.

Reference

- Bosse, Antje, Armin Zülch, May-Britt Becker, Miguel Torres, José Luis Gómez-Skarmeta, Juan Modolell, and Peter Gruss. 1997. "Identification of the Vertebrate Iroquois Homeobox Gene Family with Overlapping Expression during Early Development of the Nervous System." *Mechanisms of Development* 69 (1): 169–81.
- Carcea, Ioana, Shekhar B. Patil, Alfred J. Robison, Roxana Mesias, Molly M. Huntsman, Robert C. Froemke, Joseph D. Buxbaum, George W. Huntley, and Deanna L. Benson. 2014. "Maturation of Cortical Circuits Requires Semaphorin 7A." *Proceedings of the National Academy of Sciences of the United States of America* 111 (38): 13978–83.
- Graybuck, Lucas T., Tanya L. Daigle, Adriana E. Sedeño-Cortés, Miranda Walker, Brian Kalmbach, Garreck
- H. Lenz, Elyse Morin, et al. 2021. "Enhancer Viruses for Combinatorial Cell-Subclass-Specific Labeling." *Neuron* 109 (9): 1449–64.e13.
- Mich, John K., Lucas T. Graybuck, Erik E. Hess, Joseph T. Mahoney, Yoshiko Kojima, Yi Ding, Saroja Somasundaram, et al. 2021. "Functional Enhancer Elements Drive Subclass-Selective Expression from Mouse to Primate Neocortex." *Cell Reports* 34 (13): 108754.
- Vitalis, Tania, Luce Dauphinot, Pierre Gressens, Marie-Claude Potier, Jean Mariani, and Patricia Gaspar. 2018. "ROR α Coordinates Thalamic and Cortical Maturation to Instruct Barrel Cortex Development." *Cerebral Cortex* 28 (11): 3994–4007.
- Yao, Zizhen, Cindy T. J. van Velthoven, Michael Kunst, Meng Zhang, Delissa McMillen, Changkyu Lee, Won Jung, et al. 2023. "A High-Resolution Transcriptomic and Spatial Atlas of Cell Types in the Whole Mouse Brain." *bioRxiv*. <https://doi.org/10.1101/2023.03.06.531121>.

Reviewer Reports on the First Revision:

Referees' comments:

Referee #1 (Remarks to the Author):

In the revised manuscript by Zhou et al, the authors have made a substantial effort to respond to all critiques carefully and thoroughly. In all cases, better descriptions, clarifications or cautions have been appropriately added. The one shortcoming is the lack of functional validation requested for CREs, and my overall modest enthusiasm for the epigenomic findings – what new major insight did they provide that the transcriptomic analysis did not? However, with the GRN analysis provided (even though these are only bioinformatic analyses), and in general, considering the large scope of the work, on aggregate I think my original request may exceed the expectations from a single paper. The browser is also a very useful addition, which will hopefully allow readers to access these datasets. Overall, this paper represents a next step in connecting emerging molecularly defined cell types, to their projections, an important goal in neuroscience.

Referee #2 (Remarks to the Author):

The authors have addressed all concerns. This study provides an important contribution to the knowledge of transcriptome and epigenome signatures across cell types in the brain.